# INFORMATION BOTTLENECK FOR ACTIVE FEATURE ACQUISITION

## ABSTRACT

Traditional supervised learning typically assumes that all features are available simultaneously during deployment. However, this assumption does not hold in many real-world scenarios, such as medicine, where information is acquired sequentially based on an evolving understanding of a specific patient's condition. Active Feature Acquisition aims to address this problem by dynamically selecting which feature to measure based on the current observations, independently for each test instance. Current approaches either use Reinforcement Learning, which suffers from training difficulties; or greedily maximize the conditional mutual information of the label and unobserved features, which inherently makes myopic acquisitions. To address these shortcomings, we introduce a novel method using information bottleneck. Via stochastic encodings, we make acquisitions by reasoning about the features across many possible unobserved realizations in a regularized latent space. Extensive evaluation on a large range of synthetic and real datasets demonstrates that our approach reliably outperforms a diverse set of baselines.

## 1 INTRODUCTION

The standard supervised learning paradigm is to learn a predictive model using a training dataset of features and labels, such that the model can make accurate predictions on unseen test inputs. A fundamental assumption is that, at test time, all features are jointly available, however, this assumption does not always hold. Consider the example of a doctor diagnosing a patient (Kachuee et al., 2019b;a). Initially, there is little to no information available and, while there are many tests that could be conducted, the doctor will choose which ones to carry out based on their current understanding of the specific patient's condition. For instance, if a patient has pain in their leg, and the doctor suspects a fracture, a leg X-ray might be prioritized. Active Feature Acquisition (AFA)[1] is an inference time task, where the features are not assumed to be all available at once. Instead, on an instance-wise basis, a model sequentially acquires features based on the observations to best aid long-term prediction. A common approach is to use Reinforcement Learning (RL) (Rückstieß et al., 2013; Shim et al., 2018), since this is a natural solution to a sequential decision making problem. However, RL suffers from training difficulties such as sparse reward, exploration vs exploitation, and the deadly-triad (Henderson et al., 2018; Erion et al., 2022; Van Hasselt et al., 2018). An alternative approach is to select features that *greedily* maximize the conditional mutual information (CMI) (Chen et al., 2015a;b). This has a significant drawback: CMI does not capture the effects of unobserved features that can be acquired at a later stage due to marginalizing these out. This prevents CMI from selecting features that are independent of the label but highly informative of which feature to acquire next, resulting in myopic decision making that optimizes for immediate predictive power. Additionally, we argue that CMI is not even guaranteed to be the best short-term objective to make decisive predictions. Since maximizing CMI is equivalent to minimizing entropy and this can be achieved by making unlikely classes even more unlikely, rather than selecting features that distinguish between more probable outcomes. We explore the drawbacks of CMI in more detail in Section 4.

Motivated by the shortcomings of RL and CMI maximization, we introduce a novel AFA approach using Information Bottleneck (IB) (Tishby et al., 2000). We call our approach Information Bottleneck for Feature Acquisition (IBFA) and it departs from existing methods in several key ways. First, we shift the acquisition problem from reasoning in a complex feature space to a latent space. This is

---

[1]This problem has also been referred to as Dynamic Feature Selection in the literature.

regularized with IB such that decisions are made using label-relevant information only and not feature-level noise. Second, we use *stochastic* encoders, allowing us to acquire features by considering their effect across a diverse range of possible latent realizations. By removing marginalization over unobserved features the resulting acquisitions are non-greedy by design. Third, our acquisition objective places more focus on labels with higher predicted likelihood, leading to acquisitions that help to disambiguate between the most likely classes. Finally, to avoid the difficulties posed by RL, we do not train our model to make acquisitions directly. Instead we train with a predictive loss and make acquisitions by maximizing a custom objective in a suitably regularized latent space. Our contributions are as follows: (1) We re-examine the CMI objective and provide theoretical reasoning and concrete examples of its sub-optimality. (2) We introduce IBFA, our novel AFA approach motivated by the limitations of RL and CMI maximization. (3) We evaluate IBFA on multiple synthetic and real-world datasets, including cancer classification tasks. Comparing against various AFA baselines, we see that IBFA consistently outperforms these methods. Extensive ablations further demonstrate each novel design choice is required for the best performance.

## 2 RELATED WORK

### 2.1 ACTIVE FEATURE ACQUISITION

**Reinforcement Learning.** The most common AFA approach is to frame the problem as a Markov Decision Process and train a policy network with RL to decide which feature to acquire next (Dulac-Arnold et al., 2011; Rückstieß et al., 2013; Shim et al., 2018; Janisch et al., 2019; Mnih et al., 2014; Kachuee et al., 2019a). The RL approach readily extends to a temporal setting where features and labels can change over time (Kossen et al., 2023; Yin et al., 2020). Whilst a natural solution to AFA, RL suffers from training difficulties, and various advances in the RL field have been applied to account for this. For example, using generative models to augment datasets (Zannone et al., 2019), providing mutual information as additional input to the policy (Li & Oliva, 2021), using gradient information in the training process (Ghosh & Lan, 2023), and reward shaping (Peng et al., 2018).

**Conditional Mutual Information Maximization.** Conditional Mutual information tells us how much we can learn about one variable by measuring a second, whilst already knowing a third. Greedy CMI maximization is a common AFA approach, due to its grounding in information theory, however (as we demonstrate in Section 4), it inherently makes short-term acquisitions and is prone to making acquisitions that do not distinguish between likely labels. Among existing approaches, networks can be trained to directly predict CMI (Gadgil et al., 2024), or policy networks can be specially trained to maximize CMI without ever calculating it (Chattopadhyay et al., 2023; Covert et al., 2023). Generative models are a second way to estimate CMI by taking Monte Carlo estimates over conditional distributions, (Chattopadhyay et al., 2022; Rangrej & Clark, 2021; Early et al., 2016). This approach suffers from associated generative modeling challenges, producing poor estimates of CMI, thus adding to the limitations. Improved performance can be achieved with advances in generative modeling (Peis et al., 2022; He et al., 2022; Li et al., 2020; Li & Oliva, 2020).

**Alternative Solutions.** Sensitivity-based solutions make selections based on how sensitive the label is to a given feature (Kachuee et al., 2017; 2018). However, since missing values are filled with zero and measuring a feature is discontinuous, the gradient does not reliably represent the true sensitivity. Imitation learning has been applied (Valancius et al., 2023; He et al., 2016), however, this requires access to an oracle or to construct one. Prior to deep learning, decision trees were used, with features acquired at each branch of a tree if unobserved (Xu et al., 2012; 2013; Kusner et al., 2014; Trapeznikov & Saligrama, 2013; Xu et al., 2014). This has also been generalized to ensembles (Nan et al., 2015; 2016).

### 2.2 INFORMATION BOTTLENECK

Information bottleneck (IB) is a technique that aims to compress a feature vector to a new representation, so that as much label information is preserved while removing unnecessary feature information (Tishby et al., 2000; Alemi et al., 2017). Existing applications of IB include improving adversarial robustness (Zhang et al., 2022; Wang et al., 2021; Kuang et al., 2024), integrating data from multiple views (Lee & Van der Schaar, 2021; Wang et al., 2019; Federici et al., 2020), and recently imputation (Choi & Lee, 2023). IB has only recently been applied to standard feature selection. These methods

work by either scoring a feature with the optimized IB objective using only one feature (Pan et al., 2023), or by using a stochastic gate to drop features before the encoder, and optimizing both the encoder and gate with the IB objective (Zhang et al., 2023). We instead use IB to regularize the latent space in which we will be conducting AFA, rather than scoring features to find a fixed global subset. To our best knowledge, we are the first to apply IB in the context of AFA.

## 3 ACTIVE FEATURE ACQUISITION

**Problem Setup.** In standard $C$-way classification, we have a $d$-dimensional feature vector given by the random variable $X \in \mathcal{X}$ with realization $\mathbf{x} = (x_1, x_2, \ldots, x_d)$, and a label given by $Y \in [C]$ with realization $y$. Ordinarily, we assume all features are observed; however, more generally, we wish to allow arbitrary feature subsets as valid inputs. Therefore, let $*$ represent a missing feature value and $\mathcal{X} = \prod_{i=1}^{d} (\mathcal{X}_i \cup \{*\})$. We denote an input with feature subset $S \subseteq [d]$, as $\mathbf{x}_S$, where $x_{S,i} = x_i$ if $i \in S$, and $x_{S,i} = *$ if $i \notin S$. Given a training set $\mathcal{D}_{\text{Train}} = \{(\mathbf{x}_S, y)_n\}_{n=1}^{N}$, the AFA task is to train a model that takes a test instance with arbitrary observations $\mathbf{x}_O$, and iteratively acquires new features to improve predictive power. The model's long-term goal is to acquire a sequence of features $S^*$ to maximize its confidence in the prediction whilst minimizing the number of acquired features:

$$S^* = \underset{S \in [d] \setminus O}{\arg\max} \left( \max_{c \in [C]} p_{\text{Model}}(Y = c | \mathbf{x}_{O \cup S}) - \lambda |S| \right) \quad \text{subject to} \quad |S| \leq B$$

Where $\lambda$ balances how much we optimize for a confident prediction compared to acquiring as few features as possible, and $B$ is a given feature budget. These parameters are highly domain dependent, for example, in medicine, where the stakes are high, we have large $B$ and low $\lambda$, there is a high tolerance for acquiring features if we can make confident predictions.

**Acquisition in Practice.** The standard approach to AFA is to construct an acquisition objective function $R : \mathcal{X} \times [d] \to \mathbb{R}$, that scores each feature, and to select the feature that maximizes this: $i^* = \arg\max_{i \in [d] \setminus O} R(\mathbf{x}_O, i)$. The objective is defined by the method. As discussed in the Related Work, the two main approaches are CMI maximization and RL. CMI methods use the CMI to score features, telling us how much measuring $X_i$ will reduce the the entropy of $Y$ conditioned on $\mathbf{x}_O$: $R_{\text{CMI}}(\mathbf{x}_O, i) = I(X_i; Y | \mathbf{x}_O) = D_{\text{KL}}(p(X_i, Y | \mathbf{x}_O) || p(X_i | \mathbf{x}_O) p(Y | \mathbf{x}_O))$. RL methods use the output of a policy or Q network, trained directly on the sequential feature acquisition problem: $R_{\text{RL}}(\mathbf{x}_O, i) = Q_\theta(\mathbf{x}_O)_i$. Following this we update our observed feature set to be $O \cup i^*$, that is, we use the new observed vector as input and repeat the acquisition process.

## 4 UNDERSTANDING THE LIMITATIONS OF CMI MAXIMIZATION

Here we more closely examine the shortcomings of greedy CMI maximization for AFA, to gain understanding into *why* CMI maximization can be sub-optimal and how this can be addressed. Whilst grounded in theory and extensively applied, it suffers from two drawbacks previously alluded to.

First, greedy CMI maximization makes myopic acquisitions, which in some scenarios is guaranteed to be sub-optimal. We prove this with an example. Consider a feature vector with $d+1$ features, the first $d$ of which are binary, and the last taking an integer value from 1 to $d$: $X \in \{0, 1\}^d \times [d]$. The final feature acts as an indicator, informing us which of the other $d$ features gives the label, $y = x_{x_{d+1}}$. The optimal strategy is to first choose the indicator then its designated feature, 2 acquisitions. However, to arrive at the *same* prediction, the expected number of acquisitions by greedily maximizing CMI is $3 - \frac{1}{d}$. We prove this in Appendix G; here, we provide theoretical insight into why CMI fails on this task, motivating our solution. CMI fails because possible future observations are not considered in the present decision since they are marginalized out, $p(x_i, y | \mathbf{x}_O) = \int p(x_j, x_i, y | \mathbf{x}_O) dx_j$. Each acquisition is made like there are no subsequent acquisitions and therefore the indicator is not chosen first. This is not specific to CMI, but any scoring that marginalizes out unobserved features.

**Proposition 1.** *Any acquisition objective that uses the marginal $p(x_i, y)$ to score feature $i$ will not select the indicator first.*

The proof is straightforward: With no other features, the indicator and label are independent, so the marginal is given by $p(x_{d+1}, y) = p(x_{d+1})p(y)$. It is therefore impossible to measure its effect

on the label without considering possible values of other features, regardless of how the effect is measured. RL methods do not suffer from this, since during training different scenarios are seen and the effects distilled into the parameters. Building on this, adjusting the CMI objective to include possible values of unobserved features can solve the indicator problem under greedy maximization.

**Proposition 2.** *Greedy maximization of $\int I(X_i; Y|\mathbf{x}_O, \mathbf{x}_U)p(\mathbf{x}_U|\mathbf{x}_O)d\mathbf{x}_U$ is an optimal strategy for the indicator problem, where $\mathbf{x}_U$ are unobserved features excluding $i$, $U = \mathbf{x}_{[d]\setminus(O\cup i)}$.*

We prove this in Appendix G. Note we will *not* use this as our acquisition objective, since this is intractable. The key takeaway from these two propositions is that considering possible values of other unobserved features is *necessary* for optimality and, if the objective is chosen well, *sufficient*.

The second drawback of CMI is that, even as a short-term objective, it is not guaranteed to be the best objective for identifying the most likely class. CMI maximization is equivalent to minimizing entropy, and to show why this is not guaranteed to be optimal, consider two distributions over 3 classes and their entropies: $H([0.5, 0.5, 0.0]) = 0.693$ and $H([0.7, 0.15, 0.15]) = 0.819$. The first distribution has lower entropy, but the second is more favorable for making a prediction. It is possible to maximize CMI by making low probabilities lower, rather than distinguishing between possible answers. We provide a detailed example in Appendix H. The insight is that reducing the entropy is not always equivalent to making a decisive prediction, therefore an effective acquisition objective will place more focus on the most likely labels.

## 5 METHOD: INFORMATION BOTTLENECK FOR FEATURE ACQUISITION

To address the limitations of RL and CMI for AFA, we propose a novel method, called **I**nformation **B**ottleneck for **F**eature **A**cquisition (**IBFA**). We provide a block diagram of our method in Figure 1, showing both how the model makes predictions and calculates the acquisition objective. In short, IBFA uses an encoder-predictor architecture with intermediate latent variable $Z \in \mathcal{Z}$, predictions are given by $p_{\theta,\phi}(y|\mathbf{x}_S) = \int p_\phi(y|\mathbf{z})p_\theta(\mathbf{z}|\mathbf{x}_S)d\mathbf{z}$. The key novelty is in how we use and adapt this architecture to construct an effective acquisition objective. Our acquisition objective is:

$$R(\mathbf{x}_O, i) = \sum_{c\in[C]} p_{\theta,\phi}(Y = c|\mathbf{x}_O) \int p_\theta(\mathbf{z}|\mathbf{x}_O)r(c, \mathbf{z}, i)d\mathbf{z} \tag{1}$$

for a given function $r : [C] \times \mathcal{Z} \times [d] \to \mathbb{R}_{\geq 0}$. We formally describe $r$ in Section 5.2. At a high level, it can be viewed as calculating how much we expect measuring feature $i$ to change the predicted probability of class $c$ in the context of a sampled latent vector $\mathbf{z}$. For now it is more important to understand the objective as a whole. We break down each technical detail below, giving the motivation based on the failure cases of CMI and RL.

**Training with Predictive Loss.** To avoid the difficulites associated with training an acquisition objective with RL, we train using a novel predictive loss (given in Section 5.3) with IB regularization (Tishby et al., 2000). Our objective is *explicitly* defined in equation 1.

**Acquisition via the Latent Space.** The label can be a highly non-linear function of the features, and training an AFA model to make decisions directly in the feature space can be an equally complex task. We sidestep this difficulty by writing the acquisition objective as an expectation in a highly regularized latent space. This is why $r$ takes $\mathbf{z}$ as an explicit input, and not $\mathbf{x}_O$, under sufficient IB regularization, $\mathbf{z}$ contains only label relevant information and no noise associated with the features.

**Stochastic Encodings.** As demonstrated in Section 4, taking into account possible values of other unobserved features is necessary for optimality. Therefore we take an expectation of $r(c, \mathbf{z}, i)$, over the latent distribution $p_\theta(\mathbf{z}|\mathbf{x}_O)$. This way, future possible latent realizations are taken into account in the current decision. To sample the full diversity of the latent space, we take multiple samples during acquisition, we empirically verify the importance of this in Section 6.

**Weighting by Predictions.** Finally, as demonstrated in Section 4, CMI maximization can be achieved by focusing on reducing the likelihood of classes with already low probabilities. To overcome this, $r$ takes $c$ as input, measuring how much observing feature $i$ would affect the predicted probability of class $c$. And then an expectation is taken using the current predictions $p_{\theta,\phi}(Y = c|\mathbf{x}_O)$ so that our acquisition objective places more focus on the classes with higher predicted likelihood. We demonstrate the impact of this idea empirically in ablations in Appendix C.

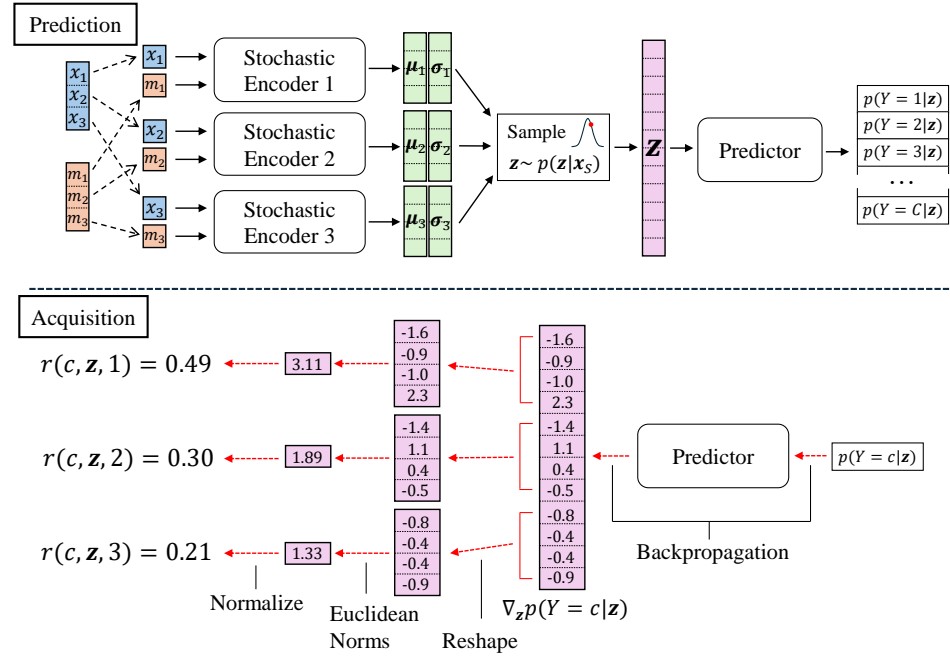

Figure 1: **Block diagram of IBFA.** Illustrated using 3 features and 4 latent components per feature. The presence or absence of a feature value is indicated with a mask vector $\mathbf{m}$. Prediction and acquisition scoring with one latent sample is given with example numerical values given for acquisition.

## 5.1 ARCHITECTURE

**Encoder.** A crucial element of our method is the ability to take decisions made in the latent space and easily translate these to the feature space. With fully connected, non-linear encoders this is a non-trivial task. To overcome this barrier we propose to factorize the latent distribution such that each feature is individually responsible for $l$ latent components

$$p_\theta(\mathbf{z}|\mathbf{x}) = \prod_{i=1}^{d} p_{\theta_i}(\mathbf{z}_{\mathcal{G}_i}|x_i).$$

Here $\mathcal{G}_i$ selects the latent components that feature $i$ is responsible for encoding. For example, if $l = 3$ then $\mathcal{G}_1 = \{1, 2, 3\}, \mathcal{G}_2 = \{4, 5, 6\}$ etc. This allows us to define $r(c, \mathbf{z}, i)$, such that it only calculates the sensitivities of the output with respect to latent samples, and then we can trivially link the most important latent components to the feature that encodes them. We achieve this factorization by having an encoder for each feature (see Figure 1). Each encoder is an MLP, $f_{\theta_i}^e : \mathcal{X}_i \times \{0, 1\} \to \mathbb{R}^l \times \mathbb{R}^l_{>0}$ with parameters $\theta_i$. They take as input a feature value and a binary mask indicating missingness, and output a mean and diagonal standard deviation of a normal distribution.

**Predictor.** We make predictions on individual latent samples with a predictor network given by an MLP, $f_\phi^p : \mathbb{R}^{ld} \to \Delta_C$ with parameters $\phi$, that predicts a probability distribution over $C$ classes.

## 5.2 SCORING FUNCTION

To calculate $r(c, \mathbf{z}, i)$, we propose using the gradients of the predicted probability with respect to the latent sample, since they are scalable, available via backpropagation, and they tell us how sensitive $p_\phi(Y = c|\mathbf{z})$ is to a latent sample. Additionally, since the distribution of $\mathbf{z}$ is factorized such that feature $i$ is only responsible for $\mathbf{z}_{\mathcal{G}_i}$, it is trivial to score features. Our proposed scoring is given by

$$r(c, \mathbf{z}, i) = \frac{||\mathbf{g}_{\mathcal{G}_i}||_2}{\sum_{j=1}^{d} ||\mathbf{g}_{\mathcal{G}_j}||_2}, \qquad \text{where } \mathbf{g} = \nabla_{\mathbf{z}} p_\phi(Y = c|\mathbf{z}). \tag{2}$$

The reasoning behind this form is as follows: the gradient vector points in the direction that locally $p_\phi(Y = c|\mathbf{z})$ is most sensitive to. We calculate the feature scores by considering the length of the gradient in a feature's associated latent components, telling us how sensitive the prediction is to those specific components. Finally, we normalize scores to sum to one to treat each latent sample equally, removing the effect of the overall gradient length. For a worked example see Figure 1 or Appendix J.

## 5.3 TRAINING

We train the above networks in a supervised fashion: training only to make predictions with appropriate latent space regularization that makes it conducive to acquisition. By training in a supervised manner, we avoid potential issues associated with RL.

**Information Bottleneck.** We use IB (Tishby et al., 2000; Alemi et al., 2017) to regularize the latent space, whose criterion for an arbitrary model with parameters $\theta$ is

$$\max_\theta I_\theta(Z; Y) - \beta I_\theta(Z; X).$$

This seeks to find a stochastic encoding of $\mathbf{x}$ to $\mathbf{z}$ that maintains maximum information about the label whilst simultaneously removing irrelevant information about the features. This is a natural choice for our application, since we reason about the acquisition in the latent space with stochastic encodings, we want to use representations that only contain information relevant to predicting the label. The standard approach to Deep IB is the Variational IB (VIB) Loss (Alemi et al., 2017), for a single subsampled point this is:

$$L_{\text{VIB}} = \mathop{\mathbb{E}}_{p_\theta(\mathbf{z}|\mathbf{x}_S)} \left[ - \log(p_\phi(y|\mathbf{z})) \right] + \beta D_{\text{KL}}(p_\theta(Z|\mathbf{x}_S)||p(Z)),$$

this is averaged over all points in the batch. The first term corresponds to $I_\phi(Z; Y)$[2] The second term corresponds to $I_\theta(Z; X)$, $\mathcal{N}(0, 1)$ is used as $p(Z)$ since it gives a closed form solution.

**Custom Loss for AFA.** The VIB loss is intended for prediction tasks only, and not AFA. Therefore we *adapt* the loss for AFA by: (1) moving the expectation over $p_\theta(\mathbf{z}|\mathbf{x}_S)$ inside the logarithm and (2) taking multiple samples giving our custom loss for a single subsampled train point

$$L = - \log \left( \mathop{\mathbb{E}}_{p_\theta(\mathbf{z}|\mathbf{x}_S)} \left[ p_\phi(y|\mathbf{z}) \right] \right) + \beta D_{\text{KL}}(p_\theta(Z|\mathbf{x}_S)||p(Z)). \tag{3}$$

The change is subtle but important. If we were to train taking the mean outside the logarithm or only using one sample, all samples from $p_\theta(\mathbf{z}|\mathbf{x})$ must *individually* produce good predictions. In particular, this affects the case where we have very few features, all samples from $p_\theta(\mathbf{z}|\mathbf{x})$ produce high uncertainty predictions. This does not affect the *predictive* power of the model, but the acquisitions suffer, during acquisition if all samples produce the same prediction then there is no diversity across latent samples, and the acquisition relies on this to make long-term acquisitions (we empirically verify this in Section 6).

**Additional Regularization.** The proposed change to the loss function is crucial to encourage diversity across latent samples. However, there is an alternative theoretical justification for the change. Within the IB framework, the change can be framed as adding further regularization to the latent space that makes it more conducive to acquisitions. Whilst we have explained why CMI is not an *optimal* acquisition objective, it still provides a useful foundation to consider how the latent space can be further regularized. The CMI objective can first be rewritten.

**Theorem 1.** *The CMI objective can be written as the equivalent minimization in the latent space*

$$\arg\min_i \mathop{\mathbb{E}}_{p_{\theta,\phi}(x_i|\mathbf{x}_O)} I_{\theta,\phi}(Z; Y|x_i, \mathbf{x}_O).$$

We prove this equivalence in Appendix E. An acquisition that maximizes CMI is one where the information between the label and latent variable is minimized, this is unexpected but the key is conditioning on features. To provide intuition, consider a latent space with disparate regions, within each region the prediction is the same. A good acquisition is one where the latent distribution $p_\theta(\mathbf{z}|\mathbf{x}_O)$ shrinks to contain only one of these regions. No matter where we move within $p(\mathbf{z}|x_i, \mathbf{x}_O)$,

---

[2]There is in fact an $H(Y)$ term missing which is not affected by the optimization so is disregarded.

the prediction is the same, and the label and latent variable are independent *conditioned on the features*. We desire a latent space where acquisitions like this are possible and regular. To encourage this property across all possible feature subsets we therefore want to additionally minimize $I_{\theta,\phi}(Z; Y|X_S)$. The caveat is that $I_\phi(Z; Y)$ must still be maximized so that individual latent samples make decisive predictions, since a trivial way to minimize $I_{\theta,\phi}(Z; Y|X_S)$ on its own is to undesirably predict a uniform distribution for any $\mathbf{z}$. Note this is *not* carrying out CMI maximization as an acquisition objective, but shaping the latent space during training, to make acquisition with our objective more effective. This desired regularization term is in fact in our custom loss.

**Theorem 2.** *The loss given in equation 3 is equivalent to* $-I_\phi(Z; Y) + \beta I_\theta(Z; X_S) + I_{\theta,\phi}(Z; Y|X_S)$.

We prove this result in Appendix F. This gives us the required IB objective with the additional desired regularization term $I_{\theta,\phi}(Z; Y|X_S)$ derived from analyzing Theorem 1.

## 6 EXPERIMENTS

Here we evaluate IBFA against various deep AFA baselines. We consider a range of synthetic, image, tabular, and medical datasets. For reproducibility, we provide full experimental details in Appendix K, including hyperparameter choices and training procedures, and full dataset details in Appendix I.

**Baselines.** We consider four different state-of-the-art baselines: Opportunistic Learning as an RL baseline (Kachuee et al., 2019a), GDFS (Covert et al., 2023) and DIME (Gadgil et al., 2024) as greedy CMI maximization methods, and EDDI (Ma et al., 2019) as a generative model for CMI maximization. We also use two vanilla baselines: a VAE (Kingma & Welling, 2013), which has a separate predictive and generative model to estimate the CMI, and an MLP to determine a fixed *global* ordering of features. Further details about all baselines are given in Appendix J.

### 6.1 SYNTHETIC DATASETS

We begin by constructing three synthetic classification tasks (denoted Syn 1-3) based on the synthetic experiments used by Yoon et al. (2019), where we know the optimal instance-wise feature ordering. These are binary classification tasks with 11 normally distributed features. Three logits, $\ell$ are calculated from the first ten features, defined as:

$$\ell_1 = 4x_1 x_2, \qquad \ell_2 = \sum_{i=3}^{6} 1.2x_i^2 - 4.2, \qquad \ell_3 = -10\sin(0.2x_7) + |x_8| + x_9 + e^{-x_{10}}$$

The binary label is sampled with $p(Y = 1) = (1 + e^\ell)^{-1}$. Syn 1 uses $\ell_1$ if $x_{11} < 0$ and $\ell_2$ otherwise. Syn 2 uses $\ell_1$ if $x_{11} < 0$ and $\ell_3$ otherwise. Syn 3 uses $\ell_2$ if $x_{11} < 0$ and $\ell_3$ otherwise. In all cases $x_{11}$ determines which features are important to the prediction, so the optimal strategy is to acquire $x_{11}$ first and then to acquire the relevant features. Table 1 shows how many features each model acquires until all features relevant to a particular instance (including $x_{11}$) are selected. IBFA achieves this in the fewest acquisitions and is close to optimal in all three datasets. Estimating CMI using generative models (EDDI and VAE) performs worse than the fixed ordering, showing that inaccurate estimation of CMI worsens the issues already associated with its greedy maximization. EDDI, in particular, consistently performs poorly across all experiments, since it is only trained to indirectly predict $y$ from $\mathbf{x}_S$ and thus subsequently inaccurately estimates CMI and $p(y|\mathbf{x}_S)$.

Table 1: Number of acquisitions to acquire the correct features on the synthetic datasets, the lower the better. We provide the mean and one standard error.

| Model | Syn 1 | Syn 2 | Syn 3 |
|---|---|---|---|
| DIME | $4.079 \pm 0.057$ | $4.581 \pm 0.194$ | $5.667 \pm 0.034$ |
| EDDI | $9.183 \pm 0.187$ | $9.208 \pm 0.371$ | $9.789 \pm 0.167$ |
| Fixed MLP | $6.009 \pm 0.000$ | $5.996 \pm 0.000$ | $7.999 \pm 0.000$ |
| GDFS | $4.568 \pm 0.195$ | $4.484 \pm 0.142$ | $5.587 \pm 0.179$ |
| Opportunistic RL | $4.203 \pm 0.034$ | $4.846 \pm 0.020$ | $5.856 \pm 0.063$ |
| VAE | $6.593 \pm 0.085$ | $6.659 \pm 0.131$ | $7.895 \pm 0.057$ |
| IBFA (ours) | $\mathbf{4.017 \pm 0.003}$ | $\mathbf{4.098 \pm 0.007}$ | $\mathbf{5.081 \pm 0.021}$ |

We investigate which features are acquired by the best four models for Syn 3 (Figure 2). IBFA consistently chooses $x_{11}$ first and then continues to make optimal acquisitions, almost achieving the best possible performance of 5 (Table 1). In contrast, DIME acquires $x_7$ first, since this has the highest mutual information initially, despite not being the best for long-term acquisitions. Therefore, when $x_{11} < 0$, DIME does not start acquiring features 3-6 until acquisition 3. GDFS performs similarly, since it is also trained to maximize CMI. Opportunistic RL tends to make noisy acquisitions, as seen by the red trajectories, demonstrating how it suffers from training difficulties. See Appendix A for equivalent diagrams and analysis for Syn 1 and Syn 2.

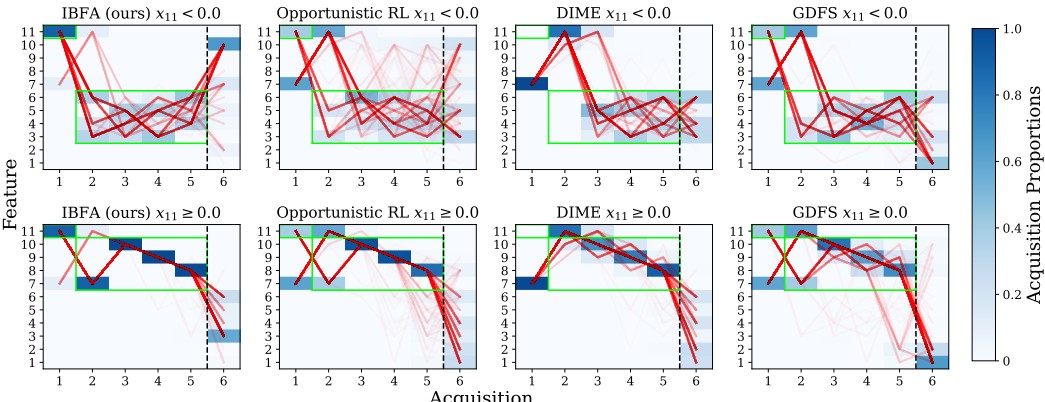

Figure 2: Acquisition heat maps and trajectories for Syn 3. Individual trajectories are plotted in red, with the acquisition proportions at each step as a heat map. Green boxes show the optimal strategy, while the vertical black line denotes the minimum number of features required (5).

**Ablations.** To provide further insight into why IBFA performs well, we conduct ablations on the synthetic datasets in Table 2. We investigate the impact of removing IB so the loss reduces to negative log-likelihood with no latent space regularization; using only a single latent sample during training so the loss reduces to the standard variational IB loss without the additional $I_{\theta,\phi}(Z; Y|X_S)$ term; using only one latent sample during acquisition so we do not sample the full diversity of the latent space; and using a deterministic encoder with no IB or sampling in the acquisition. Removing any of the novel components significantly impacts the model's performance. We examine acquisition heat maps in Appendix B to better understand the performance differences, for completeness we also carry out sensitivity analyses on $\beta$, number of train samples and number of acquisition samples.

Table 2: Ablation for number of acquisitions to acquire the correct features on the synthetic datasets, the lower the better. We provide the mean with one standard error.

| Model | Syn 1 | Syn 2 | Syn 3 |
|---|---|---|---|
| No IB | $4.529 \pm 0.074$ | $4.571 \pm 0.095$ | $5.719 \pm 0.094$ |
| 1 Train Sample | $4.420 \pm 0.156$ | $4.714 \pm 0.141$ | $5.187 \pm 0.095$ |
| 1 Acquisition Sample | $4.679 \pm 0.025$ | $4.868 \pm 0.027$ | $5.690 \pm 0.024$ |
| Deterministic Encoder | $4.910 \pm 0.105$ | $4.679 \pm 0.239$ | $5.523 \pm 0.110$ |
| IBFA (full) | $\mathbf{4.017 \pm 0.003}$ | $\mathbf{4.098 \pm 0.007}$ | $\mathbf{5.081 \pm 0.021}$ |

## 6.2 Datasets with Unknown Feature Orderings

Here, we consider multiple synthetic and real-world datasets where the correct feature ordering is not known a priori. To evaluate, we start with zero features and calculate the evaluation metric at every step during acquisition. For binary classification tasks the metric is AUROC, for multi-class it is accuracy. We report the average metric during acquisition in Table 3 and we plot the curves for IBFA, DIME, GDFS, Opportunistic RL and the fixed MLP ordering in Figure 3.

**Cube.** We start with the Cube Synthetic Dataset (Rückstieß et al., 2013; Shim et al., 2018; Zannone et al., 2019). The task is eight-way classification with 20 features. The feature vector is normally distributed around the corners of a cube, with the cube occupying three different dimensions for each

class. Irrelevant features are normally distributed around the center. IBFA has the highest average accuracy, and consistently maintains the highest acquisition curve. All active methods outperform the fixed ordering, except EDDI which suffers from the lack of an inbuilt predictive model.

Table 3: Average evaluation metrics during acquisition. Higher values are better, we report the mean and standard error.

| Model | Cube | Bank Marketing | California Housing | MiniBooNE |
|---|---|---|---|---|
| DIME | $0.901 \pm 0.001$ | $0.905 \pm 0.002$ | $0.661 \pm 0.002$ | $0.951 \pm 0.001$ |
| EDDI | $0.764 \pm 0.004$ | $0.705 \pm 0.011$ | $0.414 \pm 0.011$ | $0.843 \pm 0.007$ |
| Fixed MLP | $0.883 \pm 0.001$ | $0.908 \pm 0.001$ | $0.658 \pm 0.002$ | $0.954 \pm 0.000$ |
| GDFS | $0.900 \pm 0.000$ | $0.905 \pm 0.001$ | $0.653 \pm 0.002$ | $0.949 \pm 0.001$ |
| Opportunistic RL | $0.901 \pm 0.000$ | $0.909 \pm 0.000$ | $0.658 \pm 0.001$ | $0.953 \pm 0.000$ |
| VAE | $0.901 \pm 0.001$ | $0.877 \pm 0.002$ | $0.631 \pm 0.005$ | $0.925 \pm 0.002$ |
| IBFA (ours) | $\mathbf{0.904 \pm 0.001}$ | $\mathbf{0.919 \pm 0.001}$ | $\mathbf{0.675 \pm 0.004}$ | $\mathbf{0.957 \pm 0.000}$ |

| Model | MNIST | Fashion MNIST | METABRIC | TCGA |
|---|---|---|---|---|
| DIME | $0.731 \pm 0.002$ | $0.703 \pm 0.002$ | $0.670 \pm 0.006$ | $0.805 \pm 0.002$ |
| EDDI | $0.574 \pm 0.002$ | $0.603 \pm 0.001$ | $0.557 \pm 0.013$ | $0.635 \pm 0.006$ |
| Fixed MLP | $0.708 \pm 0.001$ | $0.690 \pm 0.001$ | $0.685 \pm 0.003$ | $0.799 \pm 0.004$ |
| GDFS | $0.732 \pm 0.001$ | $0.692 \pm 0.002$ | $0.671 \pm 0.004$ | $0.797 \pm 0.001$ |
| Opportunistic RL | $0.740 \pm 0.000$ | $0.708 \pm 0.000$ | $0.708 \pm 0.004$ | $0.839 \pm 0.001$ |
| VAE | $0.715 \pm 0.001$ | $0.685 \pm 0.001$ | $0.686 \pm 0.003$ | $0.800 \pm 0.002$ |
| IBFA (ours) | $\mathbf{0.761 \pm 0.001}$ | $\mathbf{0.717 \pm 0.001}$ | $\mathbf{0.709 \pm 0.002}$ | $\mathbf{0.845 \pm 0.002}$ |

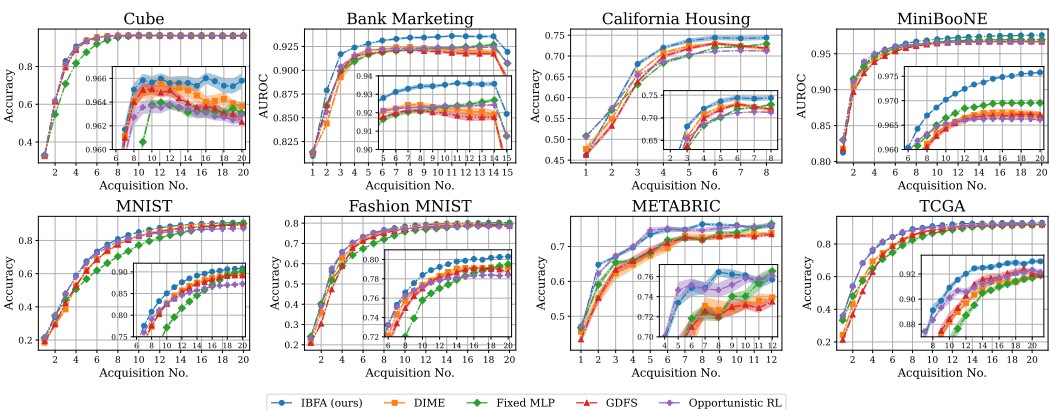

Figure 3: Evaluation metrics plots, starting from the first to the final acquisition across all datasets. Zoomed in curves are shown in the bottom right corner of each plot.

**Real Tabular.** Next, we consider three real tabular datasets. Bank Marketing (Moro et al., 2014), California Housing (Pace & Barry, 1997) and MiniBooNE (Roe et al., 2005; Roe, 2010). The Bank Marketing dataset is a binary classification task, predicting if a customer subscribes to a product based on marketing data. California Housing consists of features about houses in California districts and the label is the median house price. We converted this into four-way classification by bucketing the labels into four equally sized bins. The MiniBooNE dataset is a particle physics binary classification task trying to distinguish between electron-neutrinos and muon-neutrinos. In all cases, IBFA has both the highest average evaluation metric and maintains the best evaluation metric through the acquisition curve, in particular on Bank Marketing and Califonia Housing. Interestingly on MiniBooNE the fixed ordering is the second best method, despite the other methods actively acquiring features. Again, the generative models underperform due to inaccurate CMI estimation.

**Image Classification.** Next we consider MNIST (LeCun et al., 1998) and Fashion MNIST (Xiao et al., 2017), and acquire up to twenty pixels (Table 3 and Figure 3). Here, the fixed ordering is inadequate, and the active methods perform better. Opportunistic RL outperforms DIME and GDFS, demonstrating RL is still an effective method for AFA despite its training difficulties, whereas the problems associated with CMI maximization appear more fundamental. Again, IBFA strongly outperforms all methods by a significant margin, both in terms of average acquisition performance and the acquisition curve being consistently the highest throughout the acquisition.

### 6.3 CANCER CLASSIFICATION

Finally, we look at IBFA in the context of medicine. We consider two cancer classification tasks. The first is METABRIC (Curtis et al., 2012; Pereira et al., 2016), where the task is to predict the PAM50 status of breast cancer subjects from gene expression data. The six classes are Luminal A, Luminal B, HER2 Enriched, Basal Like, Claudin Low, and Normal Like. The second dataset uses The Cancer Genome Atlas (TCGA) (Weinstein et al., 2013). The goal is to predict the location of a tumor based on DNA methylation data. The average accuracies are given in Table 3 and the acquisition curves in Figure 3. On METABRIC, IBFA and Opportunistic RL perform similarly, outperforming all other baselines. On TCGA, IBFA significantly outperforms all baselines with Opportunistic RL a strong second, significantly outperforming DIME and GDFS (which perform worse than the fixed MLP on METABRIC), further demonstrating CMI is a flawed AFA objective.

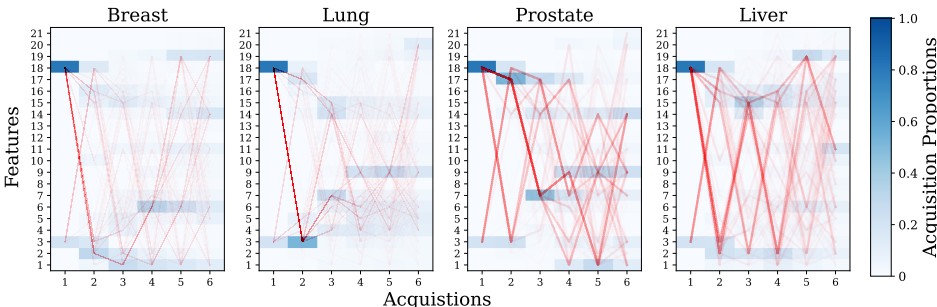

Figure 4: TCGA acquisition heat maps and trajectories for four tumor locations. We show the first 6 acquisitions.

To further validate the acquisitions of IBFA, we visualize the trajectories and heat maps for four cancer types in Figure 4, and provide scientific literature supporting the acquisitions made. The first feature selected is always ST6GAL1 (feature 18), which is known to be upregulated in a number of cancers including Breast, Prostate, Pancreatic, and Ovarian (Garnham et al., 2019). For Breast, Lung, and Liver cancers, DNASE1L3 (feature 3) is often acquired next; this gene has been identified as a potential biomarker in Breast, Liver, and Lung cancer (as well as kidney and stomach) (Deng et al., 2021), and so makes sense as a second feature to acquire for these cancers. For Prostate cancer, the second feature that tends to be acquired is SERPINB1 (feature 17), which is linked to prostate cancer (Lerman et al., 2019). For the third acquisition, for Lung and Liver cancers, IBFA typically acquires PON3 (feature 15). It has been shown that PON3 is largely restricted to solid tumors such as those in Liver, Lung, and Colon cancer (Schweikert et al., 2012).

## 7 CONCLUSION

This paper considered Active Feature Acquisition, the test time task of actively choosing which features to observe to improve a prediction. We introduced a novel approach for AFA, moving away from previous solutions based on RL and CMI maximization, using IB to regularize a stochastic latent embedding space of the features. Our method regularly outperformed previous methods across a range of tasks, and we validated acquired features in the scientific literature.

**Limitations.** Currently our method applies to classification tasks but not to regression tasks. This is because our method requires separation of class probabilities during acquisition and this notion is not well defined for continuous labels. We view this as an interesting avenue for future work. Our method also includes the encoding architecture that features are mapped *separately* to latent components. This means that observed features cannot affect the latent distribution of unobserved features. The trade-off is that it becomes trivial to link latent components to features, we see in the experiments this choice does not prevent IBFA from outperforming the baselines. Finally, due to requiring multiple latent samples, IBFA has larger memory requirements at inference time than RL baselines, depending on how many samples are used. However, CMI maximization methods with generative models also require multiple samples at inference time, so this is not a new limitation for AFA models.

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

## BROADER IMPACT

Our paper is concerned with Active Feature Acquisition, the test time task of acquiring features to iteratively improve a model's predictions on a given data instance. Applications range from medical diagnosis to polling a population. We believe on the whole these applications have a positive benefit on society. Naturally, since this is a general task (any task where features are not all available immediately), malicious applications do exist. For example, iteratively harvesting personal data to send targeted misinformation. However, this work does not focus on those applications, and since this area of machine learning research is still in its relative infancy we do not envisage this occurring for the foreseeable future. An important consideration is if this work is used in a positive setting but gives incorrect predictions. In the medical scenario a doctor might miss an important test to diagnose a patient, or conduct a painful/dangerous but unnecessary test. This work is not in a position to be deployed currently, so this is not an issue yet. However, if it were to be deployed, this problem can be mitigated by being used as a tool by domain experts to aid them in their decision making instead of replacing them.

## A  ADDITIONAL SYNTHETIC HEAT MAPS & TRAJECTORIES

To complement the synthetic experiments presented in Section 6 we provide the heat maps and trajectories for Syn 1 in Figure 5 and Syn 2 in Figure 6. In agreement with Table 1, IBFA can be seen to clearly perform best on both Syn 1 and Syn 2. In both cases $x_{11}$ is acquired first, informing the model where it needs to look next. All features are acquired by the theoretical minimum with the exception of a minority of trajectories. Opportunistic RL and DIME have a small but noticeable portion of sub-optimal trajectories on Syn 1 when $x_{11} < 0$. GDFS performs particularly poorly on Syn 1, when $x_{11} < 0$ a high proportion of required feature acquisitions are made after the theoretical minimum of 3 since initially $x_4$ and $x_5$ are selected. Additionally, GDFS regularly selects $x_{11}$ late into the acquisition process. On Syn 2, the three baselines do not place all attention on $x_{11}$ initially. In fact Opportunistic RL and GDFS mostly acquire $x_7$ first since it provides the best immediate predictive signal. When $x_{11} \geq 0$ the baselines tend to acquire all relevant features in the theoretical minimum albeit in sub-optimal orders (the same applies to Syn 1). However we see when $x_{11} < 0$ this is not the case with many required acquisitions being made after the minimum of 3, since $x_7$ has been selected first.

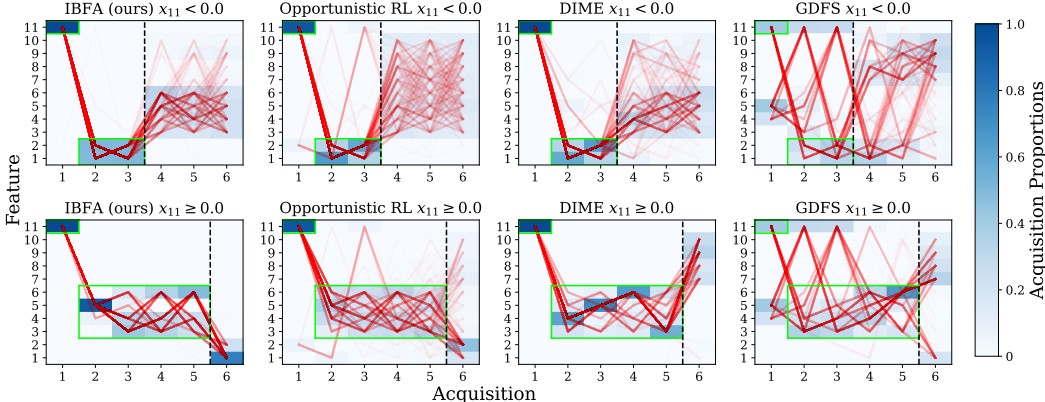

Figure 5: Acquisition heat maps and trajectories on Syn 1. Trajectories are plotted in red, with the acquisition proportions at each step as a heat map behind. We use green boxes to highlight the optimal strategy and a vertical black line to show the minimum number of features required (3 or 5).

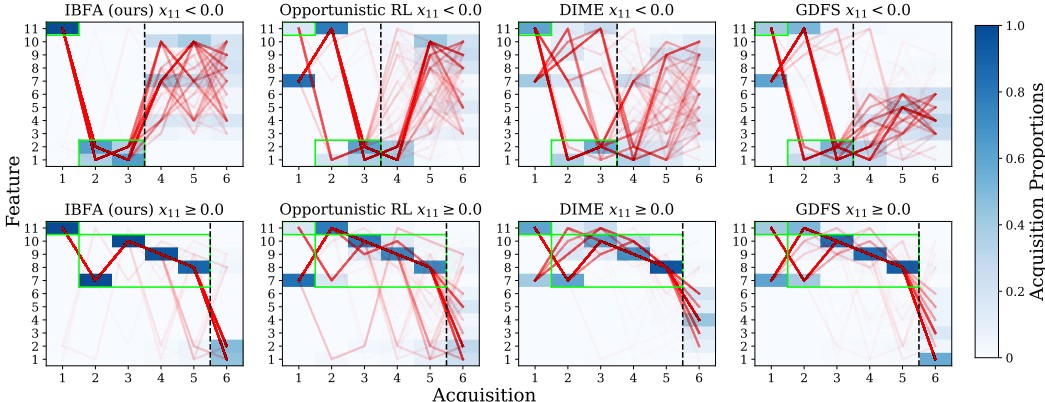

Figure 6: Acquisition heat maps and trajectories on Syn 2. Trajectories are plotted in red, with the acquisition proportions at each step as a heat map behind. We use green boxes to highlight the optimal strategy and a vertical black line to show the minimum number of features required (3 or 5).

## B SYNTHETIC ABLATIONS AND SENSITIVITY ANALYSIS

**Heat maps and Trajectories.** We supplement the synthetic ablations in Table 2 by studying the acquisition heat maps and trajectories with No IB, 1 Train Sample and 1 Acquisition Sample. We plot these for Syn 1-3 in Figures 7, 8 and 9. All three figures show that removing each of our proposed components degrades acquisition performance, confirming Table 2. All three reduced versions of IBFA in all cases select relevant features after the theoretical minimum. Acquiring with one latent sample leads to trajectories that approximately sample uniformly among all features relevant to a given synthetic task. Confirming that we need to take many acquisition samples to see a feature's effect on a diverse range of possible latent realizations. Training with one latent sample and without IB also makes noisy, sub-optimal acquisitions. All three reduced methods regularly select $x_{11}$ late into the acquisition.

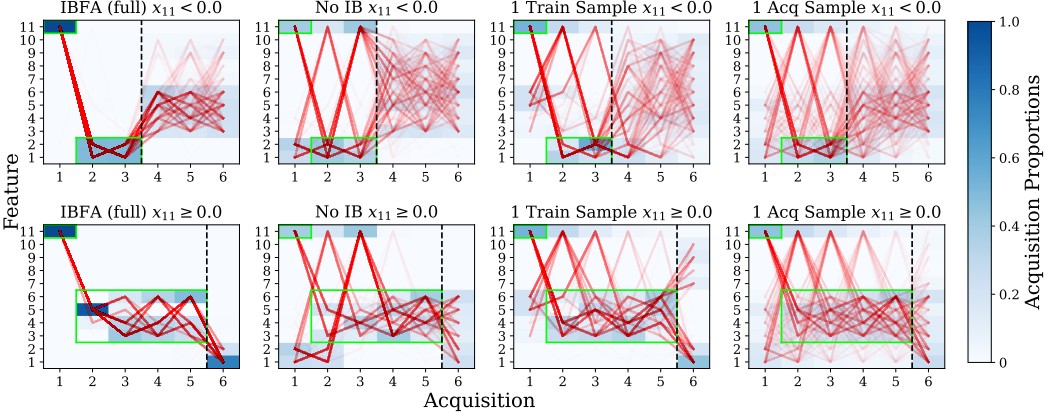

Figure 7: Acquisition heat maps and trajectories on Syn 1 ablations. Individual trajectories are plotted in red, with the acquisition proportions at each step as a heat map behind. We use green boxes to highlight the optimal strategy and a vertical black line to show the minimum number of features required (3 or 5).

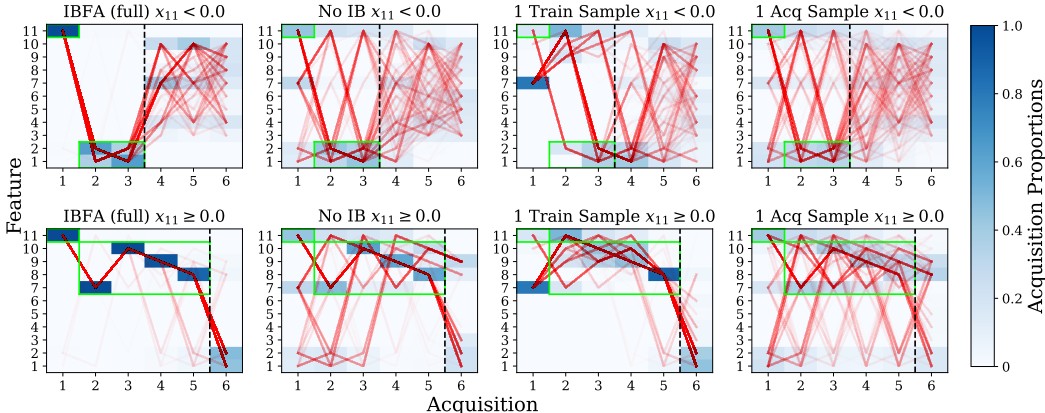

Figure 8: Acquisition heat maps and trajectories on Syn 2 ablations. Individual trajectories are plotted in red, with the acquisition proportions at each step as a heat map behind. We use green boxes to highlight the optimal strategy and a vertical black line to show the minimum number of features required (3 or 5).

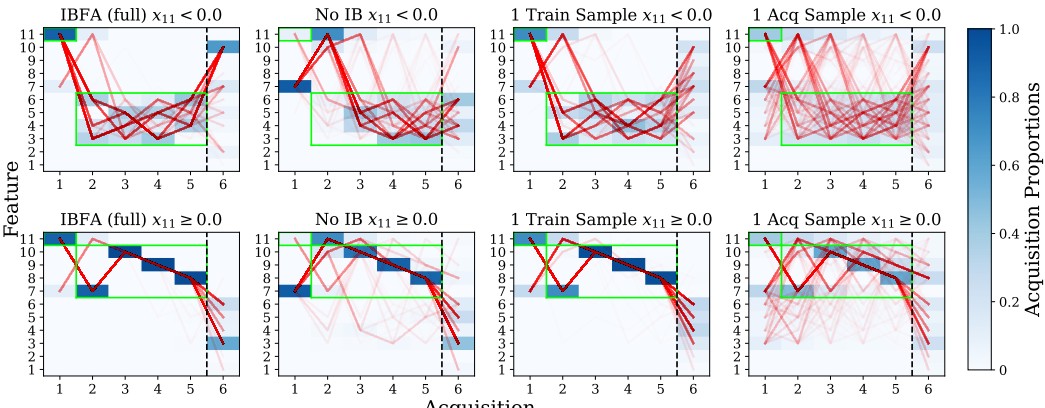

Figure 9: Acquisition heat maps and trajectories on Syn 3 ablations. Individual trajectories are plotted in red, with the acquisition proportions at each step as a heat map behind. We use green boxes to highlight the optimal strategy and a vertical black line to show the minimum number of features required (5).

**Sensitivity Analysis of $\beta$.** To further explore the importance of a well regularized latent space, we conduct a sensitivity analysis on the hyperparameter $\beta$, keeping all other hyperparameters the same. Higher $\beta$ leads to the encoders removing more information about the features. We plot the number of acquisitions required to select all relevant features on the synthetic datasets in Figure 10. For all datasets, as expected, if $\beta$ is too high, the latent space is too heavily regularized. There is not enough label information in the latent space, so decisions made there lead to sub-optimal acquisitions. Equally, by not regularizing the latent space enough, there is nothing explicitly enforcing the latent space to remove irrelevant information about the features, also leading to sub-optimal acquisitions.

**Sensitivity Analysis of Number of Acquisition Samples.** To further investigate the importance of using multiple acquisition samples, to sample the full latent diversity, we run a sensitivity analysis on the synthetic tasks. We plot the number of acquisitions required to select all relevant features in Figure 11. As expected if not enough samples are used the number of acquisitions required is larger. We use 200 acquisition samples in our experiments which is low enough for fast acquisition, and high enough that performance has plateaued.

**Sensitivity Analysis of Number of Train Samples.** To further investigate the importance of using multiple training samples, to shape the latent space for successful acquisitions, we run a sensitivity

analysis on the synthetic tasks. We plot the number of acquisitions required to select all relevant features in Figure 12. For Syn 1 and Syn 2 we see that performance tends to improve with the number of samples as expected. For Syn 3 we see the best performance is achieved with 100 samples, which is the number we used in experiments.

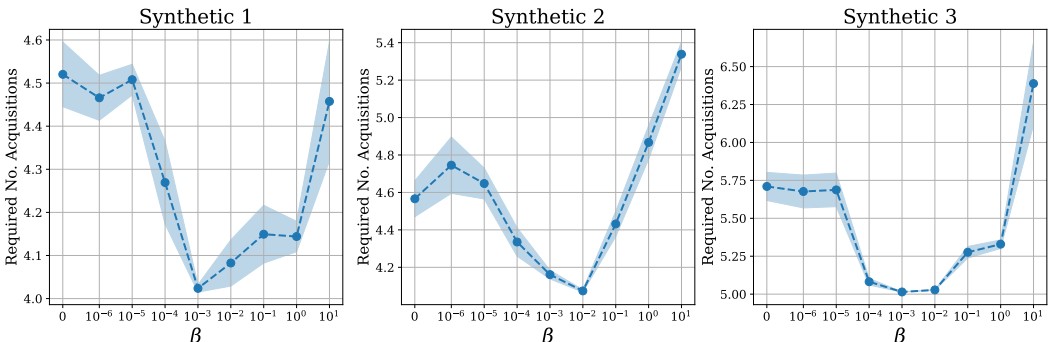

Figure 10: The number of acquisitions to select the correct relevant features for different values of $\beta$ on the synthetic tasks. The x axis is logarithmic and includes zero.

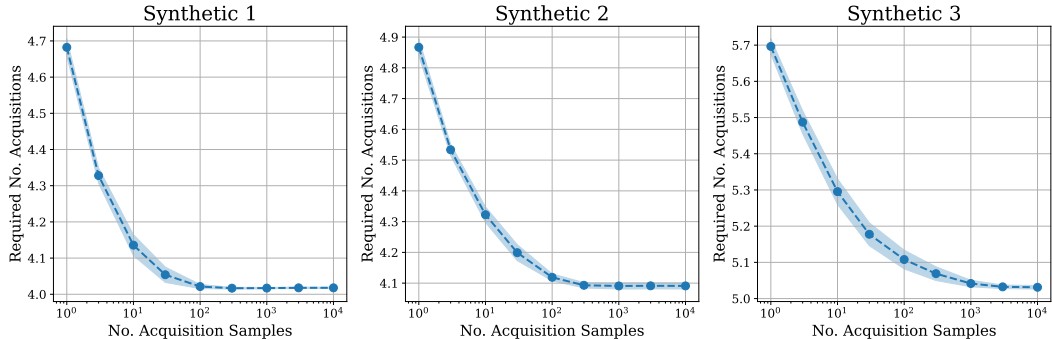

Figure 11: The number of acquisitions to select the correct relevant features for different numbers of acquisition samples on the synthetic tasks. The x axis is logarithmic.

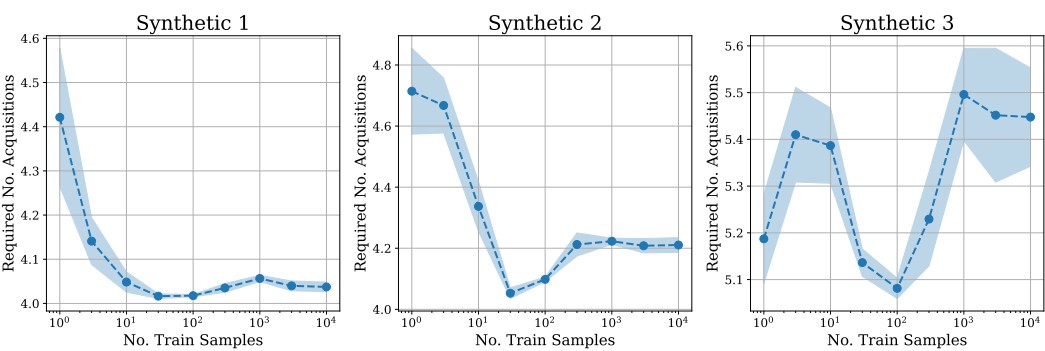

Figure 12: The number of acquisitions to select the correct relevant features for different numbers of training samples on the synthetic tasks. The x axis is logarithmic.

## C  REAL DATA ABLATIONS

To further demonstrate each novel model component leads to performance gains, we also carry out ablations on a subset of the real datasets. Additionally here we investigate the final novelty we introduced, probability weighting, where we weight the scores during acquisition by the predicted probabilities $p_{\theta,\phi}(Y = c|\mathbf{x}_O)$. We investigate the use of this technique by removing the weight and taking a mean, treating each class equally. This was not possible on the synthetic ablations because this does not affect binary classification tasks. To see this, recall how features are scored

$$R(\mathbf{x}_O, i) = \sum_{c \in [C]} p_{\theta,\phi}(Y = c|\mathbf{x}_O) \int p_\theta(\mathbf{z}|\mathbf{x}_O) r(c, \mathbf{z}, i) d\mathbf{z}.$$

Writing this in the binary case gives

$$R(\mathbf{x}_O, i) = p_{\theta,\phi}(Y = 0|\mathbf{x}_O) \int p_\theta(\mathbf{z}|\mathbf{x}_O) r(0, \mathbf{z}, i) d\mathbf{z} + p_{\theta,\phi}(Y = 1|\mathbf{x}_O) \int p_\theta(\mathbf{z}|\mathbf{x}_O) r(1, \mathbf{z}, i) d\mathbf{z}.$$

Since $p_\phi(Y = 1|\mathbf{z}) = 1 - p_\phi(Y = 0|\mathbf{z})$, $\nabla_{\mathbf{z}} p_\phi(Y = 1|\mathbf{z}) = -\nabla_{\mathbf{z}} p_\phi(Y = 0|\mathbf{z})$, therefore $r(0, \mathbf{z}, i) = r(1, \mathbf{z}, i)$, since the gradients point in opposite directions, and taking Euclidean norms and normalizing is agnostic to the negative sign. Therefore

$$R(\mathbf{x}_O, i) = p_{\theta,\phi}(Y = 0|\mathbf{x}_O) \int p_\theta(\mathbf{z}|\mathbf{x}_O) r(0, \mathbf{z}, i) d\mathbf{z} + p_{\theta,\phi}(Y = 1|\mathbf{x}_O) \int p_\theta(\mathbf{z}|\mathbf{x}_O) r(0, \mathbf{z}, i) d\mathbf{z},$$

$$R(\mathbf{x}_O, i) = \big(p_{\theta,\phi}(Y = 0|\mathbf{x}_O) + p_{\theta,\phi}(Y = 1|\mathbf{x}_O)\big) \int p_\theta(\mathbf{z}|\mathbf{x}_O) r(0, \mathbf{z}, i) d\mathbf{z}$$

$$R(\mathbf{x}_O, i) = \int p_\theta(\mathbf{z}|\mathbf{x}_O) r(0, \mathbf{z}, i) d\mathbf{z} = \int p_\theta(\mathbf{z}|\mathbf{x}_O) r(1, \mathbf{z}, i) d\mathbf{z}.$$

The weighting is removed in the binary case, thus proving treating each class equally and taking a mean will only affect the multi-class setting. Therefore, we run the ablations on the multi-class datasets MNIST, Fashion MNIST and TCGA. We provide average acquisition accuracies in Table 4 and the acquisition curves in Figure 13. As hypothesized, probability weighting leads to a significant performance improvement, the average acquisition accuracy is improved, and the full acquisition curves (blue) are consistently higher than without probability weighting (orange). Additionally, taking only either one sample during training or during acquisition also leads to performance degradation, both curves (red and purple respectively) are consistently lower than the full model curve (blue). Setting $\beta$ to zero i.e. training without IB regularization leads to the smallest drop in performance. In fact on Fashion MNIST the average acquisition accuracy is marginally higher, within one standard error. We hypothesize this is due to the MNIST and Fashion MNIST settings, since these datasets are relatively noiseless, IB regularization is not necessary, therefore does not affect performance in these cases. When we consider the noisy real medical dataset TCGA, we see that training without IB does lead to a significant performance drop, both in terms of average acquisition accuracy and that the acquisition curve (green) is consistently slightly lower than the full model curve (blue).

Table 4: Average accuracies during acquisition on multi-class ablations. We give the mean and standard error.

| Model | MNIST | Fashion MNIST | TCGA |
|---|---|---|---|
| WO Prob Weighting | $0.752 \pm 0.001$ | $0.694 \pm 0.001$ | $0.832 \pm 0.001$ |
| No IB | $0.759 \pm 0.000$ | $\mathbf{0.718 \pm 0.001}$ | $0.840 \pm 0.001$ |
| 1 Train Sample | $0.741 \pm 0.001$ | $0.707 \pm 0.001$ | $0.833 \pm 0.002$ |
| 1 Acq Sample | $0.728 \pm 0.000$ | $0.700 \pm 0.000$ | $0.826 \pm 0.002$ |
| IBFA (full) | $\mathbf{0.761 \pm 0.001}$ | $0.717 \pm 0.001$ | $\mathbf{0.845 \pm 0.002}$ |

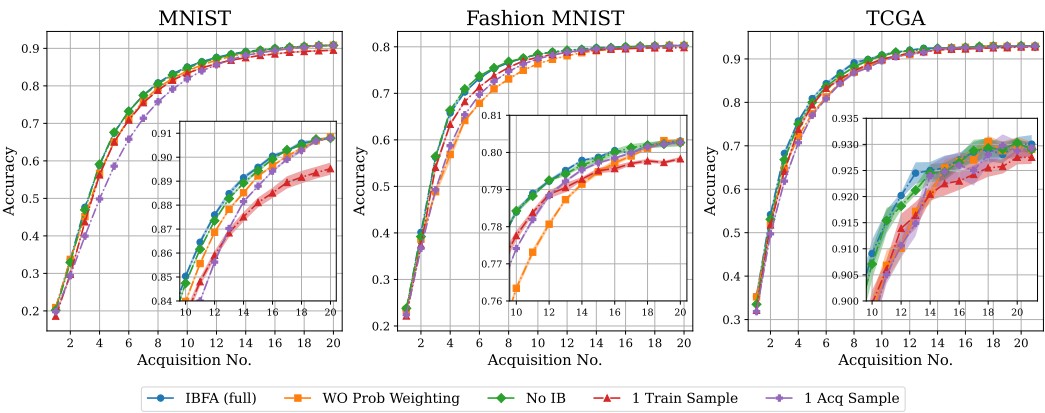

Figure 13: Evaluation metrics starting from the first to the final acquisition for the ablations. To distinguish curves we provide zoomed in versions in the bottom right corner of each plot.

## D    ADDITIONAL TCGA TRAJECTORIES

To further augment the TCGA analysis in Section 6, we provide the heat maps and trajectories across all 17 tumor locations in Figure 14. We indeed see that selections are instance-wise orderings since different trajectories emerge for the different tumor locations. Due to the nature of the task and data there is still associated noise. Further to the justification in the main paper, we see that in many cases after ST6GAL1 (feature 18), DNASE1L3 (feature 3) is selected next. This is because is has been linked to: bladder cancer, breast cancer, gastric carcinoma, liver cancer, lung adenocarcinoma, lung squamous cell carcinoma, ovarian cancer, cervical squamous cell carcinoma, head-neck squamous cell carcinoma, pancreatic adenocarcinoma and kidney renal clear cell carcinoma (Deng et al., 2021). Additionally it has been linked to colon cancer progression (Li et al., 2023), and was found to be downregulated in prostate adenocarcinoma and uterine corpus endometrial carcinoma (Deng et al., 2021). This is why we see it occasionally being selected first, it is a strong predictor on its own. ST6GAL1 is the most commonly selected but subsequently we see DNASE1L3 regularly selected second for Bladder, Breast, Stomach, Liver, Lung, Ovary, Cervical, Endometrial, Head and Neck, Pancreas, Colon, and partially for Kidney and Prostate. However we do not see it being present in the trajectories for Central Nervous System or Thyroid. Showing this acquisition choice is instance-wise and not a global decision. We likely see the selection appearing for Brain and Bone Marrow as a way to rule out these other likely locations after selecting ST6GAL1.

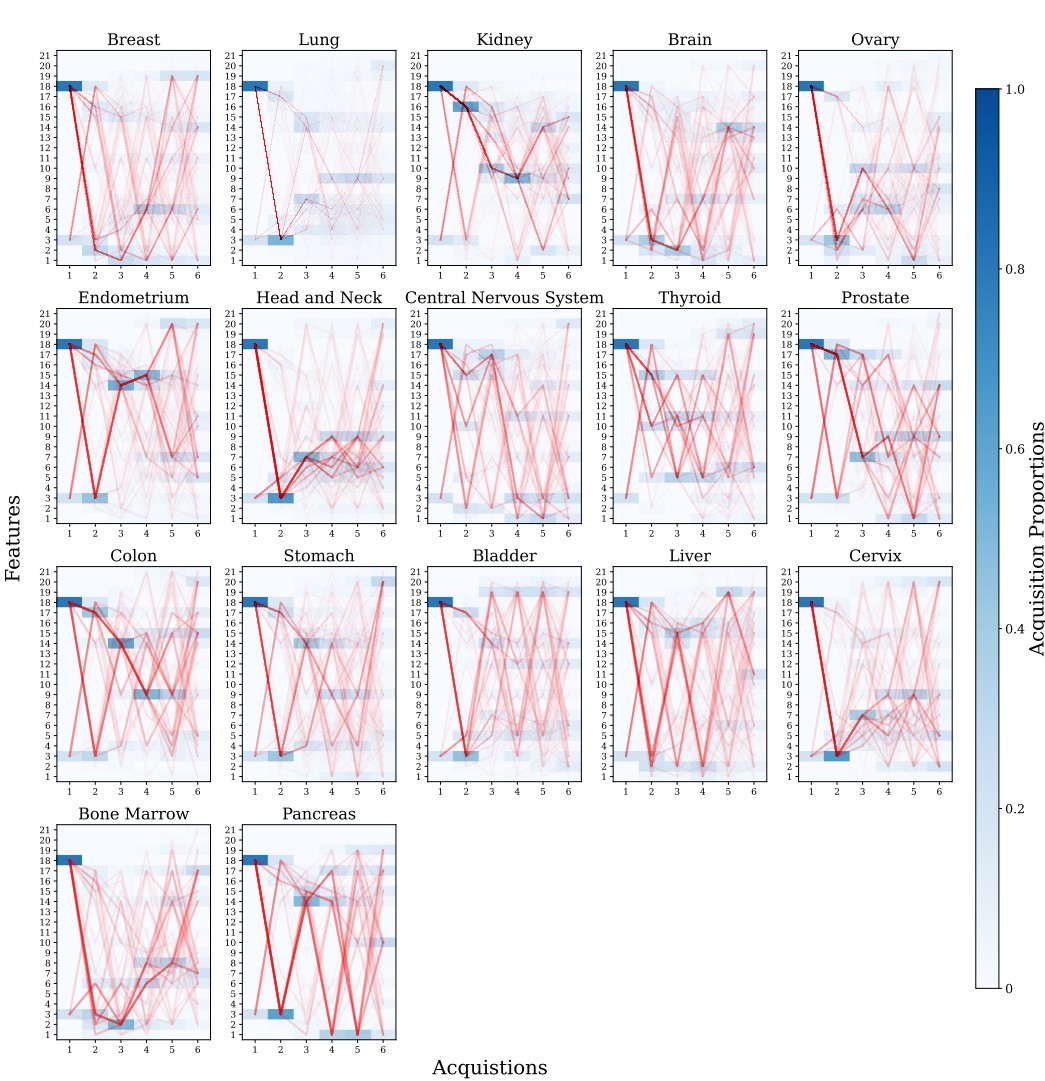

Figure 14: Acquisition Trajectories for TCGA across all classes. The trajectories are given in red, the heat map of acquisition proportions at each step are behind.

## E  Mutual Information Calculation in a Latent Space

Here we prove Theorem 1 claiming that maximizing conditional mutual information between the label and an unknown feature, as calculated by our model, can be framed as an equivalent minimization in the latent space

$$\max_i I_{\theta,\phi}(X_i; Y|\mathbf{x}_O) \equiv \min_i \mathop{\mathbb{E}}_{p_{\theta,\phi}(x_i|\mathbf{x}_O)} I_{\theta,\phi}(Z; Y|x_i, \mathbf{x}_O).$$

Our proof is based on a similar result in EDDI (Ma et al., 2019), however our proof is for a different way of writing out the CMI objective. Consider selecting an unknown feature to acquire that will maximize CMI calculated using our model

$$\max_i \int p_{\theta,\phi}(x_i, y|\mathbf{x}_O) \log \left( \frac{p_{\theta,\phi}(y|x_i, \mathbf{x}_O)}{p_{\theta,\phi}(y|\mathbf{x}_O)} \right) dy dx_i.$$

We can include a marginalization over the latent variable $\mathbf{z}$ without changing the result

$$\max_i \int p_{\theta,\phi}(x_i, y, \mathbf{z}|\mathbf{x}_O) \log \left( \frac{p_{\theta,\phi}(y|x_i, \mathbf{x}_O)}{p_{\theta,\phi}(y|\mathbf{x}_O)} \right) dy dx_i d\mathbf{z}.$$

We can then use Bayes' theorem $p(a|c) = \frac{p(a|b,c)p(b|c)}{p(b|a,c)}$ to introduce the latent variable into the numerator and denominator of the fraction in the logarithm (also using $p_{\theta,\phi}(\mathbf{z}|\mathbf{x}) = p_\theta(\mathbf{z}|\mathbf{x})$)

$$\max_i \int p_{\theta,\phi}(x_i, y, \mathbf{z}|\mathbf{x}_O) \log \left( \frac{p_{\theta,\phi}(y|\mathbf{z}, x_i, \mathbf{x}_O)p_\theta(\mathbf{z}|x_i, \mathbf{x}_O)}{p_{\theta,\phi}(\mathbf{z}|y, x_i, \mathbf{x}_O)} \frac{p_{\theta,\phi}(\mathbf{z}|y, \mathbf{x}_O)}{p_{\theta,\phi}(y|\mathbf{z}, \mathbf{x}_O)p_\theta(\mathbf{z}|\mathbf{x}_O)} \right) dy dx_i d\mathbf{z}.$$

We only need to consider the first part of the logarithm since the second part is not affected by the optimization over $i$. We also use the fact that our model architecture enforces $y$ being independent of $\mathbf{x}$ conditioned on $\mathbf{z}$, ($p_{\theta,\phi}(y|\mathbf{z}, x_i, \mathbf{x}_O) = p_\phi(y|\mathbf{z})$), since the Markov chain of the encoder-predictor architecture is $X - Z - Y$. This gives

$$\max_i \int p_{\theta,\phi}(x_i, y, \mathbf{z}|\mathbf{x}_O) \log \left( \frac{p_\phi(y|\mathbf{z})p_\theta(\mathbf{z}|x_i, \mathbf{x}_O)}{p_{\theta,\phi}(\mathbf{z}|y, x_i, \mathbf{x}_O)} \right) dy dx_i d\mathbf{z}.$$

Again we remove the part that does not depend on $x_i$

$$\max_i \int p_{\theta,\phi}(x_i, y, \mathbf{z}|\mathbf{x}_O) \log \left( \frac{p_\theta(\mathbf{z}|x_i, \mathbf{x}_O)}{p_{\theta,\phi}(\mathbf{z}|y, x_i, \mathbf{x}_O)} \right) dy dx_i d\mathbf{z}.$$

We flip the fraction in the logarithm and turn the maximization into a minimization

$$\min_i \int p_{\theta,\phi}(x_i, y, \mathbf{z}|\mathbf{x}_O) \log \left( \frac{p_{\theta,\phi}(\mathbf{z}|y, x_i, \mathbf{x}_O)}{p_\theta(\mathbf{z}|x_i, \mathbf{x}_O)} \right) dy dx_i d\mathbf{z}.$$

We again apply Bayes' theorem to $p_{\theta,\phi}(x_i, y, \mathbf{z}|\mathbf{x}_O) = p_{\theta,\phi}(\mathbf{z}, y|x_i, \mathbf{x}_O)p_{\theta,\phi}(x_i|\mathbf{x}_O)$ giving

$$\min_i \mathop{\mathbb{E}}_{p_{\theta,\phi}(x_i|\mathbf{x}_O)} \left[ \int p_{\theta,\phi}(\mathbf{z}, y|, x_i, \mathbf{x}_O) \log \left( \frac{p_{\theta,\phi}(\mathbf{z}|y, x_i, \mathbf{x}_O)}{p_\theta(\mathbf{z}|x_i, \mathbf{x}_O)} \right) dy d\mathbf{z} \right].$$

After applying the definition of conditional mutual information this gives

$$\min_i \mathop{\mathbb{E}}_{p_{\theta,\phi}(x_i|\mathbf{x}_O)} I_{\theta,\phi}(Z; Y|x_i, \mathbf{x}_O)$$

as an equivalent latent space minimization. Completing the proof.

## F  Loss Function

Here we prove Theorem 2. Our loss function, when we include an expectation over a subsampled batch, is given by

$$\mathop{\mathbb{E}}_{p_\text{D}(\mathbf{x}_S, y)} \left[ -\log(\mathop{\mathbb{E}}_{p_\theta(\mathbf{z}|\mathbf{x}_S)} [p_\phi(y|\mathbf{z})]) \right] + \beta \mathop{\mathbb{E}}_{p_\text{D}(\mathbf{x}_S)} \left[ D_\text{KL}(p_\theta(Z|\mathbf{x}_S)||p(Z)) \right]$$

Where the difference with the standard variational Information Bottleneck loss is taking the first term, moving the expectation inside the logarithm and taking many samples. The expectation inside the logarithm makes this equal to the model's predicted negative log-likelihood.

$$\mathbb{E}_{p_{\mathrm{D}}(\mathbf{x}_S, y)} \left[ -\log(p_{\theta,\phi}(y|\mathbf{x}_S)) \right]$$

From here, to ease notation, we drop the subsampling index $S$, and include it in the data distribution of $X$. The negative log-likelihood is equal to $-I_{\theta,\phi}(X;Y) + H(Y)$.

Using the chain rule of mutual information $I_{\theta,\phi}(X;Y) = I_{\theta,\phi}(Y;X,Z) - I_{\theta,\phi}(Z;Y|X)$.

Our encoder-predictor architecture enforces the Markov chain $X - Z - Y$, such that $y$ is independent of $\mathbf{x}$ conditioned on $\mathbf{z}$. Therefore, $I_{\theta,\phi}(Y;X,Z) = I_{\phi}(Y;Z) = I_{\phi}(Z;Y)$, giving

$$I_{\theta,\phi}(X;Y) = I_{\phi}(Z;Y) - I_{\theta,\phi}(Z;Y|X).$$

Substituting this back into the loss function gives

$$L = -I_{\phi}(Z;Y) + H(Y) + \beta I_{\theta}(Z;X) + I_{\theta,\phi}(Z;Y|X).$$

Finally, since the entropy of the label does not depend on the model, and is therefore a constant, it is disregarded in the simplified final form giving

$$L = -I_{\phi}(Z;Y) + \beta I_{\theta}(Z;X) + I_{\theta,\phi}(Z;Y|X),$$

completing the proof.

## G    INDICATOR EXAMPLE

Here we elaborate on our indicator example, a simple case where CMI fails. First we demonstrate that CMI fails, and then we show that by considering possible unobserved feature values in the calculation we can recover the optimal policy.

Recall the example, we have features $X \in \{0,1\}^d \times [d]$, i.e the first $d$ dimensions are binary and the final feature is an indicator. The label is given by using the value at the feature index given by the indicator $y = x_{x_{d+1}}$. In the absence of any of the first $d$ features, the indicator and label are independent $p(y, x_{d+1}) = p(y)p(x_{d+1})$. Substituting this into the definition of mutual information gives

$$I(Y;X_{d+1}) = D_{\mathrm{KL}}(p(y,x_{d+1})||p(y)p(x_{d+1})) = D_{\mathrm{KL}}(p(y)p(x_{d+1})||p(y)p(x_{d+1})) = 0.$$

Now consider the mutual information for the other features. Due to the symmetry of the problem, the mutual information for one of these features is the same for all others. The mutual information can be more usefully written as

$$I(Y;X_i) = H(Y) - \int H(Y|x_i)p(x_i)dx_i.$$

The entropy of the label is $\log 2$ since there is equal chance of being 0 or 1. Again using the symmetry of the system, the entropy of $Y$ if $X_i = 0$ is the same as if $X_i = 1$, so we only calculate for one case. When $X_i = 0$, the probability of $Y = 0$ is $\frac{1}{d} \times 1 + \frac{d-1}{d} \times \frac{1}{2}$. Since in $\frac{1}{d}$ cases it takes the exact value of $X_i$ based on the value of the indicator, and in $\frac{d-1}{d}$ cases $Y$ is given by a different unknown feature value. This gives $p(Y = 0|X_i = 0) = \frac{d+1}{2d}$. The expression for binary entropy, $-p\log(p) - (1-p)\log(1-p)$ is maximized by $p = 0.5$, giving $\log 2$. Since $p(Y = 0|X_i = 0) > 0.5$, the entropy is lower than $\log 2$ in this case. Exploiting the symmetry of the system we conclude that $\int H(Y|x_i)p(x_i)dx_i < \log 2$, and therefore $I(Y;X_i) > 0$.

Therefore, the indicator is never chosen first, which is a sub-optimal strategy. It can be shown, but is not necessary, that the indicator will be chosen second, a sketch of the reasoning is that now that the value of one feature is known, the indicator and the label are now correlated. Therefore, there is non-zero CMI which turns out to be larger than for the other features. And once the indicator is chosen the correct feature is the only feature afterward with non-zero CMI. So this strategy will acquire the correct features in 3 selections $\frac{d-1}{d}$ of the time (random feature, indicator, correct feature)

and in 2 selections $\frac{1}{d}$ of the time (correct feature, indicator). Thus the expected number of acquisitons for this strategy is

$$2\frac{1}{d} + 3\frac{d-1}{d} = 3 - \frac{1}{d}$$

So as $d$ gets large, the expected number of required acquisitions approaches 3.

Now consider our proposed solution of using an information theoretic objective that considers the values of other features. Recall Proposition 2, we propose $\int I(Y; X_i | \mathbf{x}_U, \mathbf{x}_O) p(\mathbf{x}_U | \mathbf{x}_O) d\mathbf{x}_U$, recovers the optimal strategy, where $\mathbf{x}_U$ is the vector of all other unobserved features. We prove that this will lead to an optimal strategy below.

Initially there are no features, so the acquisition objective is $\int I(Y; X_i | \mathbf{x}_U) p(\mathbf{x}_U) d\mathbf{x}_U$. Writing this in terms of entropies gives

$$\int I(Y; X_i | \mathbf{x}_U) p(\mathbf{x}_U) d\mathbf{x}_U = \int \left( H(Y | \mathbf{x}_U) - \int H(Y | x_i, \mathbf{x}_U) p(x_i | \mathbf{x}_U) dx_i \right) p(\mathbf{x}_U) d\mathbf{x}_U.$$

The entropy when all features are known is zero, so for any $i$ this is

$$\int H(Y | \mathbf{x}_U) p(\mathbf{x}_U) d\mathbf{x}_U.$$

If we consider one of the first $d$ features, we can again apply symmetry to calculate this quantity for feature $i$ and apply it to all of them. In $\frac{d-1}{d}$ cases the entropy is zero, since we will have all of the information required. However if $x_{d+1} = i$, then $H(Y | \mathbf{x}_U) = \log 2$, since we don't know feature $i$ and therefore $Y$ has equal likelihood of being 0 or 1, this happens in $\frac{1}{d}$ cases so this quantity is $\frac{\log 2}{d}$ for the first $d$ features.

For the indicator, $p(Y = 0 | \mathbf{x}_U)$ is the proportion of the first $d$ features for a given sample $\mathbf{x}_U$ that are also 0. All features are independent with probability 0.5 of being 0, so this becomes a binomial distribution with $d$ trials

$$\sum_{i=0}^{d} \binom{d}{i} \frac{1}{2^d} \left( -\left(\frac{i}{d}\right) \log \left(\frac{i}{d}\right) - \left(1 - \frac{i}{d}\right) \log \left(1 - \frac{i}{d}\right) \right).$$

It is not immediately clear that this is larger than the quantity $\frac{\log 2}{d}$ for the other features. The first thing we can do is calculate this quantity when $d = 3$, which gives 0.477, and this is larger than $\frac{\log 2}{3} = 0.231$. And the next thing is to notice that this quantity is increasing with $d$, since as $d$ gets larger there will be more probability mass at $i = \frac{d}{2}$. As $d \to \infty$ the binomial distribution becomes Gaussian with mean $\frac{d}{2}$ and variance $\frac{d}{4}$, so $\frac{i}{d}$ will approximately be distributed normally with mean $\frac{1}{2}$ and standard deviation $\frac{1}{2\sqrt{d}}$. Therefore this quantity asymptotes towards $\log 2$.

Therefore for $d \geq 3$, this objective will choose the indicator first, and not the other features (for $d = 2$ all features are scores the same, and for $d = 1$ the indicator is not the optimal choice). After choosing the indicator, the second selection is trivial. The relevant feature has non-zero CMI, all other features are independent of the label conditioned on the indicator so they have zero CMI. Therefore the correct feature is chosen. This strategy's expected number of acquisitions is 2, which is less than $3 - \frac{1}{d}$.

This example illustrates that by considering the possible realizations in the calculation, and not marginalizing them out, we can make long-term acquisitions. Note we do not use this specific quantity in our paper, it involves an additional expectation over unobserved values as well as the expectation inside the CMI which is intractable. This does not even account for the difficulty in estimating the conditional distributions in feature space.

## H    ENTROPY EXAMPLE

In Section 4 we claimed that CMI maximization can lead to acquisitions that focus on making low probabilities lower, rather than distinguishing between possible answers. Here we provide a concrete example of this occurring. We have binary feature vectors with six features $X \in \{0, 1\}^6$. The label consists of three classes, where the probabilities of each class are given by

$$p(Y = 0 | \mathbf{x}) = \frac{x_1 + x_2}{\sum_i x_i} \quad , \quad p(Y = 1 | \mathbf{x}) = \frac{x_3 + x_4}{\sum_i x_i} \quad , \quad p(Y = 2 | \mathbf{x}) = \frac{x_5 + x_6}{\sum_i x_i}$$

Now consider the case where $X_1 = 0$. The current distribution of $Y$ is $[0.204, 0.398, 0.398]$.

In this case if we acquire feature 2 then in half the cases $X_2 = 0$ and the distribution becomes $[0.020, 0.490, 0.490]$. In the other half of cases $X_2 = 1$ and the distribution becomes $[0.388, 0.306, 0.306]$. In both cases acquiring feature 2 does not help to distinguish between the possible answers very well.

If instead we acquire any of the other features, lets say feature 3. If $X_3 = 0$, which happens in half the cases, the distribution becomes $[0.255, 0.255, 0.490]$. And in the other half of cases $X_3 = 1$, the distribution becomes $[0.153, 0.541, 0.306]$. In both of these cases the feature has helped to distinguish between likely scenarios more than feature 2.

Finally, we can calculate the CMI for all features when feature 1 is 0.

$$I(X_1; Y | X_1 = 0) = 0$$
$$I(X_2; Y | X_1 = 0) = 0.1389$$
$$I(X_3; Y | X_1 = 0) = 0.0055$$
$$I(X_4; Y | X_1 = 0) = 0.0055$$
$$I(X_5; Y | X_1 = 0) = 0.0055$$
$$I(X_6; Y | X_1 = 0) = 0.0055$$

Naturally there is 0 CMI for feature 1 since it is already known. However feature 2 has the largest CMI, and so a CMI objective would acquire this feature over the other 4, which is undesirable. The results from this example can be calculated by enumerating all possible feature values and label probabilities and calculating the quantities directly with a computer. We include code to reproduce this calculation.

## I    DATASET DETAILS

Here we provide all the details about each dataset, including sizes, number of features, and how to access the real datasets.

**Synthetic.**    The synthetic experiments are based on (Yoon et al., 2019) where we know the features that are predictive, and we know that there is a heterogenous order. The datasets are binary datasets where the feature vector has 11 independent features drawn from a standard normal. There are three possible logits:

$$\ell_1 = 4x_1 x_2, \qquad \ell_2 = \sum_{i=3}^{6} 1.2 x_i^2 - 4.2, \qquad \ell_3 = -10\sin(0.2 x_7) + |x_8| + x_9 + e^{-x_{10}}$$

Then for a given logit value the label is sampled from a Bernoulli distribution with probability $p(Y = 1) = (1 + e^{\ell})^{-1}$. We construct three datasets:

- Synthetic 1: If $x_{11} < 0$ we use $\ell_1$, otherwise $\ell_2$
- Synthetic 2: If $x_{11} < 0$ we use $\ell_1$, otherwise $\ell_3$
- Synthetic 3: If $x_{11} < 0$ we use $\ell_2$, otherwise $\ell_3$

The logits have been adapted from the originals in (Yoon et al., 2019) to produce probabilities closer to 0 or 1. This is so all the models have stronger purely predictive performance. The train set is size 60000, the validation and test set are both size 10000. AUROC is used as the evaluation metric.

**Cube.**    The Cube dataset is a synthetic dataset that is regularly used to evaluate Active Feature Acquisition methods (Rückstieß et al., 2013; Shim et al., 2018; Zannone et al., 2019). We specifically use the normally distributed version (Zannone et al., 2019). There are 20 continuous features, where different features are relevant for different classes. All features are drawn from a normal distribution with mean 0.5 and standard deviation 0.3, except for the following cases:

- Class 1: Features 1, 2, 3 have mean $[0, 0, 0]$ and diagonal standard deviation $[0.1, 0.1, 0.1]$.
- Class 2: Features 2, 3, 4 have mean $[1, 0, 0]$ and diagonal standard deviation $[0.1, 0.1, 0.1]$.

- Class 3: Features 3, 4, 5 have mean $[0, 1, 0]$ and diagonal standard deviation $[0.1, 0.1, 0.1]$.
- Class 4: Features 4, 5, 6 have mean $[1, 1, 0]$ and diagonal standard deviation $[0.1, 0.1, 0.1]$.
- Class 5: Features 5, 6, 7 have mean $[0, 0, 1]$ and diagonal standard deviation $[0.1, 0.1, 0.1]$.
- Class 6: Features 6, 7, 8 have mean $[1, 0, 1]$ and diagonal standard deviation $[0.1, 0.1, 0.1]$.
- Class 7: Features 7, 8, 9 have mean $[0, 1, 1]$ and diagonal standard deviation $[0.1, 0.1, 0.1]$.
- Class 8: Features 8, 9, 10 have mean $[1, 1, 1]$ and diagonal standard deviation $[0.1, 0.1, 0.1]$.

We use a train set with size 60000 and the validation and test sets are both size 10000. Accuracy is the evaluation metric.

**Bank Marketing.** The Bank Marketing dataset (Moro et al., 2014) can be found at: `https://archive.ics.uci.edu/dataset/222/bank+marketing`, we accessed it on 19th April 2024, the dataset has a Creative Commons Attribution 4.0 International license. The data is taken from a marketing campaign conducted by a Portuguese bank. The task is binary classification, where the label indicates whether a client subscribed to a term deposit at the bank. The features are both the client's information and information about the calls. There are 15 features in total (after combining the month and day of the call into one feature), 7 are continuous and 8 are categorical. A full list of features can be found at the dataset origin. We use an 80:10:10 split giving train, validation and test sizes of 36168, 4521 and 4522. The evaluation metric is AUROC.

**California Housing.** The California Housing dataset is obtained through Scikit-Learn (Pedregosa et al., 2011) `https://scikit-learn.org/stable/modules/generated/sklearn.datasets.fetch_california_housing.html`, using a Creative Commons 0 license. The labels are median house prices in California districts expressed in 100,000 dollars. There are 8 continuous features that can be found at the above URL. To convert this to a classification task we bucket the labels into 4 equally sized bins. We use an 80:10:10 split giving train, validation and test sizes of 16512, 2064 and 2064. The evaluation metric is accuracy.

**MiniBooNE.** MiniBooNE is an experiment at Fermilab designed to detect neutrino oscillations, namely muon neutrinos into electron neutrinos (Roe et al., 2005; Roe, 2010). The data was obtained from `https://archive.ics.uci.edu/dataset/199/miniboone+particle+identification` on 23rd February 2024, the dataset has a Creative Commons Attribution 4.0 International license. The task is binary classification, distinguishing electron neutrino events from background events. There are 50 continuous features. The dataset does not have balanced classes, we enforced balance by reducing the number of background events at random to match the number of signal events. We also reduced the feature set down to 20 features using STG as a preprocessing feature selection step (Yamada et al., 2020). The selected features were [ 2, 3, 6, 14, 15, 17, 20, 21, 22, 23, 25, 26, 29, 34, 39, 40, 41, 42, 43, 44]. The train set is size 56499 and the validation and test sets are both size 10000. The evaluation metric is AUROC.

**MNIST and Fashion MNIST.** MNIST and Fashion MNIST are image classificaton datasets with 10 classes, consisting of images of handwritten digits and items of clothing respectively. MNIST is available under the Creative Commons Attribution-Share Alike 3.0 license and Fashion MNIST uses the MIT license. Both datasets have images that are $28 \times 28 = 784$ pixels. We preprocess by reducing the dimensionality to 20 pixels each, for computational reasons - an acquisition trajectory with 784 features, where the majority are redundant, will be very slow, especially for methods such as EDDI and VAE where the whole acquisition is $\mathcal{O}(d^2)$. To do this we use STG (Yamada et al., 2020) a deep learning method for feature selection. After flattening the images to vectors, the features found by STG were:

- MNIST: [153, 154, 210, 211, 243, 269, 271, 295, 327, 348, 350, 375, 405, 409, 427, 430, 461, 514, 543, 655]
- Fashion MNIST: [ 10, 38, 121, 146, 202, 246, 248, 341, 343, 362, 406, 434, 454, 490, 546, 574, 580, 602, 742, 770]

For both datasets we split the provided train set into a train set with size 50000 and validation set with size 10000, we use the provided test sets each with size 10000. The evaluation metric is accuracy.

**METABRIC.** The Molecular Taxonomy of Breast Cancer International Consortium (METABRIC) database consists of clinical and genetic Data for 1,980 breast cancer subjects (Curtis et al., 2012;

Pereira et al., 2016). The data was accessed at `https://www.kaggle.com/datasets/raghadalharbi/breast-cancer-gene-expression-profiles-metabric` on 25th April 2024 under the Apache 2.0 license. We construct a classification task, predicting the Pam50 status using gene expressions as features. There are six classes:

1. Luminal A
2. Luminal B
3. Her2 Enriched
4. Claudin Low
5. Basal Low
6. Normal

As with the other high dimensional datasets we used STG to reduce the dimensionality to twelve continuous gene expressions given by:

1. CCNB1
2. CDK1
3. E2F2
4. E2F7
5. STAT5B
6. Notch 1
7. RBPJ
8. Bcl-2
9. eGFR
10. ERBB2
11. ERBB3
12. ABCB1

We use an 80:10:10 split resulting in train, validation and test sizes of 1518, 189 and 191. The evaluation metric is accuracy.

**TCGA.** The Cancer Genome Atlas (TCGA) consists of genetic data for over 11,000 cancer patients (Weinstein et al., 2013). The data was accessed at `https://www.cancer.gov/ccg/research/genome-sequencing/tcga` on 7th January 2023 under their Data Use Agreement. We construct the classification task of predicting location of the tumor based on DNA methylation data. We use 17 locations as the classes:

1. Breast
2. Lung
3. Kidney
4. Brain
5. Ovary
6. Endometrium
7. Head and Neck
8. Central Nervous System
9. Thyroid
10. Prostate
11. Colon
12. Stomach
13. Bladder
14. Liver
15. Cervix
16. Bone Marrow
17. Pancreas

As the first step of dimensionality reduction we removed features with more than 15% missingness. Following this, we used STG to reduce dimensionality to 21 features:

1. C7orf51
2. DEF6
3. DNASE1L3
4. EFS
5. FOXE1
6. GPR81
7. GRIA2
8. GSDMC
9. HOXA9
10. KAAG1
11. KLF5
12. LOC283392
13. LTBR
14. LYPLAL1
15. PON3
16. POU3F3
17. SERPINB1
18. ST6GAL1
19. TMEM106A
20. ZNF583
21. ZNF790

We then removed subjects with more than 10% missing features and used an 80:10:10 split. This gave train, validation and test sizes of 6327, 790 and 792. The evaluation metric is accuracy.

## J  MODEL DETAILS AND IMPLEMENTATIONS

All models were implemented using PyTorch (Paszke et al., 2017), code shall be released publicly after the review period. It can currently be found in the supplementary material. We implemented all models ourselves to fit into our pipeline, the applicable licenses for the baseline models are:

- Opportunistic RL: MIT License (`https://github.com/mkachuee/Opportunistic`)
- DIME: No license provided (`https://github.com/suinleelab/DIME/tree/main`)
- GDFS: MIT License (`https://github.com/iancovert/dynamic-selection`)
- EDDI: Microsoft Research License (`https://github.com/microsoft/EDDI`)

## J.1 GENERAL MODEL DETAILS

Here we provide details that tend to be shared across models. We explicitly state if a model does not follow the above and provide model specific details in the next section.

**Input Layer.**    In this paper not all features are available all at once. In order to account for this we use a binary mask to indicate whether a feature is available or not to a model. The input $\mathbf{x}$ and the mask $\mathbf{m}$ go through an input layer before the main model which accounts for missing features. For continuous features the input is given by $[\mathbf{x} \odot \mathbf{m}, \mathbf{m}]$, which is the element-wise product between the continuous features & their mask concatenated with the mask. Categorical features use a one-hot encoding, where we include an additional class to indicate a missing feature, i.e. if the mask value is 0 then the encoding has 1 at the first position. Continuous and categorical features are encoded separately as above and then concatenated as input to the main model. This applies to the Fixed MLP, DIME, GDFS, Opportunistic RL and VAE.

**Deep Networks.**    All deep networks follow the same structure. After any specific input layers, we use linear layers. Each hidden layer has a ReLU activation followed by Batch Normalization (Ioffe & Szegedy, 2015). All hidden layers in a given network are the same width, which is a hyperparameter that can be tuned as well as the number of hidden layers. The exception to this is the Opportunistic RL model, where we replace Batch Normalization with dropout with $0.5$ probability in accordance with the method's implementation (Kachuee et al., 2019a).

**Acquiring Features.**    To acquire features each method individually has its own way to positively score all features, where higher scores mean that feature is better to acquire. These scores are multiplied by $(1 - \mathbf{m})$ so that we do not acquire features we already have. This is also multiplied by the full data mask so that we do not acquire features that are not available. This would not apply at deployment where we have the ability to measure all features if desired.

## J.2 MODEL SPECIFIC DETAILS

Here we include any key details that are specific to given models, such as hyperparameter names and roles. We highly recommend seeing each method's paper for full details of each model. Unless otherwise stated, each method follows the general rules stated previously.

**Multi-layer Perceptron.**    The Fixed MLP uses a simple MLP structure as described above. It is trained for 120 epochs. We prevent overfitting during training by choosing the iteration with the best validation accuracy/AUROC. The greedy fixed order is found after training by masking out all features, and calculating the evaluation metric on the train set for each feature individually. The best feature is chosen and is unmasked for the model. The procedure is repeated with the best feature being kept to find the second best feature. This is repeated until all features have been placed in a fixed greedy order.

**GDFS.**    GDFS (Covert et al., 2023) has two separate networks, one for prediction, one for scoring features. Both use the same input layer previously described, both have a softmax final activation to give a probability distribution over the label and a positive score for each feature. Our implementation follows the original. We use the same hidden width and number of hidden layers for both networks. The Boolean "Share Parameters" hyperparameter says whether to share half the hidden layers between the two networks, this is presented in the paper as a possible way to increase performance, we treat it as a hyperparameter. We carry out pretraining on the predictor network for 80 epochs, we then carry out main training on both networks. This is done using a geometric temperature progression of $T \times [1.00, 0.56, 0.32, 0.18, 0.1]$, where the initial temperature $T$ is a hyperparameter. For each temperature in the progression main training is carried out for 15 epochs, please see the original paper for full details, the temperature is used in the reparameterized sampling of features from the network scores (hence why we use softmax to convert to a distribution). Main training consists of sampling feature acquisitions and training the scoring network to choose features with the best greedy prediction from the predictor network.

**DIME.**    DIME (Gadgil et al., 2024) uses two separate networks, one for prediction and one for predicting the CMI of features with the label. The information network is used to score each feature. The prediction network uses softmax to give a distribution as the prediction. The information network limits the output to a minimum of zero and maximum of the entropy of the current predictions $H(Y|\mathbf{x}_S)$ (Cover, 1999). This is done by using a sigmoid followed by multiplying by the entropy as

suggested in the paper. The majority of the DIME implementation follows the GDFS implementation above. Instead of a temperature progression we use an $\epsilon$ progression during main training (as is done in the paper). This is given by $\epsilon$-Initial$\times[1.0, 0.25, 0.05, 0.005]$, this gives the probability of choosing a feature uniformly at random compared to the best feature predicted by the information network. This and the temperature parameter in GDFS allow the models to explore the space of possibilities early and exploit the best ones later. Main training is also done for 15 epochs for each $\epsilon$ value, the information network is trained to predict the change in loss when a given feature is acquired.

**Opportunistic RL.** Opportunistic RL (Kachuee et al., 2019a) is a Deep Q learning method, where the reward is given by the $l_1$ norm of the change in prediction distribution after an acquisition. The target network is updated compared to the main network with a rate of 0.001 as suggested. Batch Normalization is replaced with dropout with probability 0.5 as suggested. Predictions are made by using dropout to provide different network parameters at test time with 50 samples taken and averaged as suggested. The P and Q networks share representations as described in the paper. The $\gamma$ hyperparameter refers to the discount factor associated with RL. The model is trained for 20000 episodes with evaluation every 100 episodes. For the first 2000 episodes only the predictor network is trained, using uniformly random actions. Following this the probability of a random action decays by $0.1^{\frac{1}{20000}}$ every episode to a minimum of 0.1. After 10000 episodes and for every 2000 episodes after that, the learning rate decays by a factor of 0.2. This is all in line with the original implementation. The only change is that each episode we do not consider individual samples from the dataset (it is not an online stream of data), instead we train using a batch of samples each episode, this improves the training, improving Opportunistic RL compared to its original online setting.

**VAE.** The VAE method is a vanilla generative modeling approach to the AFA problem to analyse the viability of generative models. We use a Variational Auto-Encoder (Kingma & Welling, 2013) to model the distribution of the features. Since our input layer allows for missing features this allows us to model the distribution of missing features conditioned on observed ones. We train with the standard ELBO. The encoder and decoder are separate networks with separate width and number of hidden layers. The encoder predicts $\mu_z$ and $\sigma_z$, where $\sigma_z$ is diagonal and is enforced to be positive by pushing the activations through a softplus and adding 0.001 as a minimum. The decoder predicts a mean for continuous features, with $\sigma$ being the standard deviation to estimate the normal log-likelihood (this is a hyperparameter). For categorical features the decoder predicts logits that go through softmax. We then train a separate predictor that uses a standard MLP. Features are scored by taking samples of the unknown features conditioned on the observed ones. These samples go through the predictor to give an estimated label distribution. The mutual information is then estimated with $I(X_i; Y|\mathbf{x}_S) = \mathbb{E}_{p(x_i|\mathbf{x}_S)}[D_{\text{KL}}(p(Y|x_i, \mathbf{x}_S)||p(Y|\mathbf{x}_S))]$. We train for 120 epochs. We prevent overfitting during training by choosing the iteration with the best validation ELBO.

**EDDI.** EDDI (Ma et al., 2019) is an advanced generative modeling method for AFA. The encoder is a Partial VAE. For each continuous feature $x_i$, the input to a shared encoding network is $[x_i, x_i\mathbf{e}_i]$, where $\mathbf{e}_i$ is a learnable vector which is different for each feature. This goes through a shared network giving $\mathbf{s}_i$ for each continuous feature. For categorical features a learnable representation is created for each feature for each possible category including a missing category. So without a network we still create $\mathbf{s}_i$ for a categorical feature by learning a matrix and selecting the row according to the category for each feature. We then take the sum $\mathbf{c} = \sum_i m_i\mathbf{s}_i$ so we only include the representations for observed features. This aggregated representation $\mathbf{c}$ goes through another network to give the latent $\mu_z$ and $\sigma_z$. We enforce $\sigma_z$ to be positive by pushing it through a softplus and adding 0.001 as the minimum. The number of hidden encoder layers refers to both the continuous feature encoder and c-to-latent encoder, the number is divided by 2 and that many are used in each. We encode the label in the same way as a categorical feature. We do not include a separate predictor, instead we follow the original paper to make predictions: features are encoded to a latent distribution and samples are decoded to $y$, the absence of a dedicated predictor negatively impacts the results for EDDI. The decoder follows the same structure as for the VAE. We train for 400 epochs, to prevent overfitting we choose the iteration with the best validation ELBO. Features are scored based on a sampled KL divergence calculated in the latent space as described in the original paper, we use 50 samples.

**IBFA.** Our method, as described in the main paper, encodes each feature separately to a normal distribution (so we have many small encoding networks, one for each continuous feature). For each continuous feature we give $[m_i\tilde{x}_i, m_i]$ to that feature's specific deep encoder, where each continuous feature also goes through a copula transform $\tilde{x}_i = \Phi^{-1}(F_i(x_i))$ initially. The copula

transform is given by $F_i$, the empirical CDF of the continuous feature, followed by $\Phi^{-1}$, the inverse standard normal CDF. This transformation, as described in its paper (Wieczorek et al., 2018), enforces a symmetry associated with Information Bottleneck and encourages sparse, disentangled latent representations, both desired properties. The networks predict $\mu$ and $\sigma$ for each feature by outputing two unbounded vectors whose size is the number of latent components per feature - a hyperparameter. Both of the unbounded vectors go through Batch Normalization which we found sped up training. The first is $\mu$ and the second goes through softplus and has 0.001 added, enforcing it to be positive giving $\sigma$.

For each categorical feature we have a learnable matrix, where each row is a vector whose size is the number of latent components per feature. So for a given category (where missing is the first category) we simply select the row of the matrix. We use two of these for each feature for the $\mu$ and $\sigma$. The selected vectors are unbounded so they go through the same procedure - batch normalization applied to both, and then softplus and +0.001 to the second to get $\mu$ and $\sigma$. After concatenating, samples from the latent distribution can go through an MLP predictor network with softmax to give a predicted label distribution for that sample.

During acquisition we encode the features we have to the latent distribution, and take 200 samples. Each is pushed through the predictor to give a distribution for each sample. To score the features we take the gradient of each classes' probability with respect to every sample. This gives $\mathbf{g} = \nabla_{\mathbf{z}} p_\phi(Y = c | \mathbf{z})$. To convert this to a score we calculate the normalized length of the vector in each feature's latent dimensions. For example, if we have two features and each is encoded to three latent components, the gradient could be

$$[1.19, -0.87, 0.81, 0.63, -0.40, 0.29].$$

We reshape to the number of features by the number of latent components per feature:

$$[[1.19, -0.87, 0.81], [0.63, -0.40, 0.29]].$$

We calculate the length of each of these giving

$$[\sqrt{1.19^2 + 0.87^2 + 0.81^2}, \sqrt{0.63^2 + 0.40^2 + 0.29^2}] = [1.68, 0.801].$$

We then normalize by dividing by the sum of these

$$[1.68, 0.801]/(1.68 + 0.801) = [0.68, 0.32].$$

And this gives us a score for each feature from this latent sample for this class $r(c, \mathbf{z}, i)$. We average across all samples, and sum across all classes weighted by $p(Y = c | \mathbf{x}_O)$. This gives a score for every feature.

To train, we subsample the feature and encode them to the latent distribution. We take 100 samples, these go through the predictor giving 100 label distributions which are averaged as the model's full prediction. We then calculate the log-likelihood of the overall prediction. We then add the KL divergence of the latent distribution with a standard normal to enforce the information bottleneck regularization. We train for 120 epochs, using 200 latent samples for acquisition and prediction during evaluation.

### J.3 MODEL RUNTIMES

There are two places to consider runtime: training and inference. In Table 5 we provide the scaling laws of each method with respect to number of features $d$.

RL, DIME and GDFS train by simulating acquisition, so each step scales linearly with the number of features. Generative models (and IBFA) are constant to train since they only train to predict well. However, during inference, RL, DIME and GDFS only require one forward pass of their policy/CMI network, whereas EDDI and VAE must individually score every feature. IBFA instead takes gradients with respect to the predicted class outputs, so the runtime is linear in the number of classes, which is typically far fewer than the number of features. The main takeaway is that IBFA scales better than half the methods at training time, better than the other half during acquisition (assuming fewer labels than features), and never the worst.

Table 5: Runtimes for the models that actively acquire features.

| Model | Single Training Step | Single Acquisition Step |
|---|---|---|
| DIME | $\mathcal{O}(d)$ | $\mathcal{O}(1)$ |
| GDFS | $\mathcal{O}(d)$ | $\mathcal{O}(1)$ |
| EDDI | $\mathcal{O}(1)$ | $\mathcal{O}(d)$ |
| Opportunistic RL | $\mathcal{O}(d)$ | $\mathcal{O}(1)$ |
| VAE | $\mathcal{O}(1)$ | $\mathcal{O}(d)$ |
| IBFA | $\mathcal{O}(1)$ | $\mathcal{O}(|y|)$ |

## K  EXPERIMENTAL DETAILS

All experiments were run on an Nvidia Quadro RTX 8000 GPU the data sheet can be found at `https://www.nvidia.com/content/dam/en-zz/Solutions/design-visualization/quadro-product-literature/quadro-rtx-8000-us-nvidia-946977-r1-web.pdf`. All experiments were repeated five times over parameter initializations to obtain means and standard error estimates. Experiments took approximately one month to complete.

**Training.**  We train all models using the Adam optimizer (Kingma & Ba, 2015), the learning rate and batch size are treated as hyperparameters that are tuned using a validation set. All methods (except for Opportunistic RL) use a learning rate scheduler that multiplies the learning rate by 0.2 when there have been a set number of epochs without validation metric improvement - the patience, which is also tuned.

We prevent overfitting during training by tracking a validation metric every epoch and using the iteration with the best value. The validation metric we choose (unless explicitly stated for a given model) is the area under the acquisition curve, starting from zero features we acquire features individually, calculating the accuracy/AUROC at each acquisition, and then the validation metric is the area under the acquisition curve divided by the total number of features.

**Hyperparameter Tuning.**  For every model, initial hyperparameter tuning was conducted by finding ranges for each hyperparameter that produced strong acquisition performance on the synthetic datasets. Following this, for each model we generated 9 random hyperparameter configurations using the ranges.[3] For each method we test each configuration 3 times producing a mean value for the area under the acquisition curve. The configuration with the highest mean value is separately trained 5 times in the main experiments. The nine configurations for each method are provided in Tables 6, ,7, 8, 9, 10, 11 and 12. We give the selected hyperparameter configurations for each dataset in Table 13.

Table 6: Hyperparameter configurations for Fixed MLP.

| Hyperparameter | 1 | 2 | 3 | 4 | 5 |
|---|---|---|---|---|---|
| Hidden Width | 200 | 100 | 200 | 100 | 300 |
| No. Hidden Layers | 2 | 2 | 1 | 1 | 2 |
| Learning Rate | 0.001 | 0.001 | 0.001 | 0.001 | 0.001 |
| Batch Size | 128 | 128 | 128 | 128 | 256 |
| Patience | 5 | 5 | 5 | 5 | 5 |

| Hyperparameter | 6 | 7 | 8 | 9 |
|---|---|---|---|---|
| Hidden Width | 100 | 250 | 50 | 120 |
| No. Hidden Layers | 2 | 3 | 2 | 2 |
| Learning Rate | 0.001 | 0.001 | 0.001 | 0.0005 |
| Batch Size | 128 | 256 | 64 | 128 |
| Patience | 2 | 10 | 5 | 5 |

---

[3]We did a random search because we did not have the computational resources to carry out a full grid search.

Table 7: Hyperparameter configurations for DIME.

| Hyperparameter | 1 | 2 | 3 | 4 | 5 |
|---|---|---|---|---|---|
| Hidden Width | 200 | 200 | 200 | 200 | 100 |
| No. Hidden Layers | 2 | 2 | 2 | 2 | 2 |
| Share Parameters | False | False | False | True | True |
| Pretraining Learning Rate | 0.001 | 0.001 | 0.001 | 0.001 | 0.001 |
| Main Training Learning Rate | 0.001 | 0.001 | 0.001 | 0.001 | 0.001 |
| Batch Size | 128 | 128 | 128 | 128 | 128 |
| Patience | 5 | 5 | 5 | 5 | 2 |
| $\epsilon$ Initial | 0.4 | 0.2 | 0.1 | 0.4 | 0.2 |
| **Hyperparameter** | **6** | **7** | **8** | **9** | |
| Hidden Width | 200 | 100 | 100 | 100 | |
| No. Hidden Layers | 2 | 1 | 3 | 1 | |
| Share Parameters | True | False | False | True | |
| Pretraining Learning Rate | 0.001 | 0.001 | 0.001 | 0.001 | |
| Main Training Learning Rate | 0.001 | 0.001 | 0.001 | 0.0001 | |
| Batch Size | 128 | 512 | 256 | 512 | |
| Patience | 5 | 5 | 3 | 5 | |
| $\epsilon$ Initial | 0.1 | 0.4 | 0.2 | 0.1 | |

Table 8: Hyperparameter configurations for GDFS.

| Hyperparameter | 1 | 2 | 3 | 4 | 5 |
|---|---|---|---|---|---|
| Hidden Width | 200 | 200 | 200 | 200 | 200 |
| No. Hidden Layers | 2 | 2 | 2 | 2 | 2 |
| Share Parameters | False | False | False | True | True |
| Pretraining Learning Rate | 0.001 | 0.001 | 0.001 | 0.001 | 0.001 |
| Main Training Learning Rate | 0.001 | 0.001 | 0.001 | 0.001 | 0.001 |
| Batch Size | 128 | 128 | 128 | 128 | 128 |
| Patience | 2 | 2 | 2 | 2 | 2 |
| Temp Initial | 2.0 | 1.0 | 0.1 | 2.0 | 1.0 |
| **Hyperparameter** | **6** | **7** | **8** | **9** | |
| Hidden Width | 200 | 100 | 200 | 200 | |
| No. Hidden Layers | 2 | 1 | 2 | 2 | |
| Share Parameters | True | True | False | True | |
| Pretraining Learning Rate | 0.001 | 0.001 | 0.001 | 0.001 | |
| Main Training Learning Rate | 0.001 | 0.001 | 0.001 | 0.001 | |
| Batch Size | 128 | 512 | 512 | 512 | |
| Patience | 2 | 2 | 2 | 2 | |
| Temp Initial | 0.1 | 2.0 | 1.0 | 0.1 | |

Table 9: Hyperparameter configurations for VAE.

| Hyperparameter | 1 | 2 | 3 | 4 | 5 |
|---|---|---|---|---|---|
| Latent Width | 30 | 10 | 50 | 30 | 50 |
| No. Hidden Decoder Layers | 2 | 2 | 2 | 1 | 2 |
| Decoder Hidden Width | 100 | 200 | 150 | 200 | 200 |
| No. Hidden Encoder Layers | 2 | 2 | 2 | 1 | 1 |
| Encoder Hidden Width | 100 | 200 | 150 | 200 | 150 |
| No. Hidden Predictor Layers | 2 | 2 | 2 | 1 | 2 |
| Predictor Hidden Width | 100 | 100 | 200 | 200 | 200 |
| Learning Rate | 0.001 | 0.001 | 0.001 | 0.001 | 0.001 |
| Batch Size | 128 | 128 | 128 | 128 | 256 |
| $\sigma$ Decoder | 0.2 | 0.2 | 0.2 | 0.2 | 0.2 |
| Patience | 5 | 5 | 5 | 5 | 5 |

| Hyperparameter | 6 | 7 | 8 | 9 |
|---|---|---|---|---|
| Latent Width | 10 | 30 | 40 | 20 |
| No. Hidden Decoder Layers | 2 | 2 | 2 | 3 |
| Decoder Hidden Width | 100 | 100 | 200 | 250 |
| No. Hidden Encoder Layers | 2 | 2 | 2 | 3 |
| Encoder Hidden Width | 100 | 100 | 200 | 250 |
| No. Hidden Predictor Layers | 2 | 2 | 2 | 2 |
| Predictor Hidden Width | 100 | 100 | 200 | 100 |
| Learning Rate | 0.001 | 0.0005 | 0.001 | 0.001 |
| Batch Size | 512 | 64 | 128 | 512 |
| $\sigma$ Decoder | 1.0 | 0.2 | 0.2 | 0.2 |
| Patience | 5 | 5 | 3 | 5 |

Table 10: Hyperparameter configurations for EDDI.

| Hyperparameter | 1 | 2 | 3 | 4 | 5 |
|---|---|---|---|---|---|
| C Dim | 200 | 200 | 50 | 100 | 20 |
| Latent Width | 200 | 200 | 100 | 50 | 20 |
| No. Hidden Decoder Layers | 2 | 2 | 2 | 2 | 2 |
| Decoder Hidden Width | 200 | 200 | 200 | 200 | 200 |
| No. Hidden Encoder Layers | 2 | 2 | 2 | 2 | 2 |
| Encoder Hidden Width | 200 | 200 | 200 | 200 | 200 |
| Learning Rate | 0.001 | 0.001 | 0.001 | 0.001 | 0.001 |
| Batch Size | 128 | 512 | 128 | 256 | 512 |
| $\sigma$ Decoder | 0.2 | 1.0 | 0.2 | 0.2 | 0.2 |
| Patience | 5 | 5 | 5 | 5 | 5 |

| Hyperparameter | 6 | 7 | 8 | 9 |
|---|---|---|---|---|
| C Dim | 80 | 250 | 100 | 60 |
| Latent Width | 80 | 250 | 40 | 60 |
| No. Hidden Decoder Layers | 1 | 2 | 2 | 1 |
| Decoder Hidden Width | 100 | 100 | 75 | 200 |
| No. Hidden Encoder Layers | 2 | 3 | 2 | 3 |
| Encoder Hidden Width | 100 | 100 | 75 | 200 |
| Learning Rate | 0.001 | 0.001 | 0.001 | 0.001 |
| Batch Size | 128 | 128 | 256 | 512 |
| $\sigma$ Decoder | 0.2 | 0.2 | 0.2 | 0.2 |
| Patience | 5 | 5 | 5 | 5 |

Table 11: Hyperparameter configurations for Opportunistic RL.

| Hyperparameter | 1 | 2 | 3 | 4 | 5 |
|---|---|---|---|---|---|
| Hidden Width | 200 | 200 | 200 | 100 | 200 |
| No. Hidden Layers | 2 | 2 | 2 | 2 | 2 |
| RL $\gamma$ | 0.5 | 0.75 | 0.25 | 0.5 | 0.75 |
| Learning Rate | 0.001 | 0.001 | 0.001 | 0.001 | 0.001 |
| Batch Size | 128 | 128 | 128 | 256 | 256 |

| Hyperparameter | 6 | 7 | 8 | 9 |
|---|---|---|---|---|
| Hidden Width | 200 | 100 | 200 | 100 |
| No. Hidden Layers | 2 | 1 | 1 | 1 |
| RL $\gamma$ | 0.25 | 0.5 | 0.75 | 0.25 |
| Learning Rate | 0.0001 | 0.001 | 0.0001 | 0.001 |
| Batch Size | 128 | 256 | 256 | 128 |

Table 12: Hyperparameter configurations for IBFA.

| Hyperparameter | 1 | 2 | 3 | 4 | 5 |
|---|---|---|---|---|---|
| Latent Components per Feature | 4 | 4 | 4 | 6 | 4 |
| No. Hidden Predictor Layers | 2 | 2 | 2 | 2 | 2 |
| Predictor Hidden Width | 100 | 250 | 100 | 150 | 250 |
| No. Hidden Encoder Layers | 2 | 2 | 2 | 2 | 2 |
| Encoder Hidden Width | 20 | 150 | 20 | 50 | 150 |
| IB $\beta$ | 0.0005 | 0.001 | 0.001 | 0.0005 | 0.005 |
| Learning Rate | 0.001 | 0.0005 | 0.001 | 0.001 | 0.0003 |
| Batch Size | 128 | 128 | 128 | 128 | 128 |
| Patience | 5 | 5 | 5 | 5 | 5 |

| Hyperparameter | 6 | 7 | 8 | 9 |
|---|---|---|---|---|
| Latent Components per Feature | 8 | 4 | 6 | 8 |
| No. Hidden Predictor Layers | 1 | 2 | 3 | 2 |
| Predictor Hidden Width | 250 | 180 | 250 | 250 |
| No. Hidden Encoder Layers | 1 | 2 | 3 | 2 |
| Encoder Hidden Width | 100 | 40 | 100 | 100 |
| IB $\beta$ | 0.0001 | 0.0008 | 0.001 | 0.005 |
| Learning Rate | 0.001 | 0.0005 | 0.0005 | 0.0005 |
| Batch Size | 256 | 128 | 256 | 128 |
| Patience | 5 | 8 | 5 | 5 |

Table 13: Selected hyperparameter configurations for each dataset.

| Dataset | Opportunistic RL | DIME | GDFS | Fixed MLP | VAE | EDDI | IBFA |
|---|---|---|---|---|---|---|---|
| Syn 1 | 1 | 5 | 9 | 7 | 1 | 3 | 4 |
| Syn 2 | 1 | 6 | 6 | 7 | 2 | 5 | 1 |
| Syn 3 | 5 | 4 | 6 | 7 | 8 | 3 | 6 |
| Cube | 3 | 4 | 6 | 3 | 2 | 4 | 5 |
| Bank Marketing | 4 | 4 | 3 | 7 | 9 | 8 | 4 |
| California Housing | 3 | 6 | 5 | 7 | 3 | 7 | 7 |
| MiniBooNE | 6 | 4 | 2 | 7 | 9 | 9 | 7 |
| MNIST | 3 | 6 | 6 | 7 | 3 | 7 | 8 |
| Fashion MNIST | 3 | 4 | 9 | 7 | 5 | 4 | 8 |
| METABRIC | 9 | 5 | 2 | 5 | 4 | 4 | 5 |
| TCGA | 6 | 4 | 2 | 1 | 4 | 4 | 4 |

## L  INTERPRETABILITY

To explore one possible way of interpreting our model, we consider the log determinant of the latent covariance matrix. This tells us the latent uncertainty as we acquire features. We consider the synthetic datasets where we know the optimal behavior. We consider the case with no features, all features, feature 1, feature 5, feature 10 and feature 11 since each of those is associated with one of the logits of the synthetic datasets and feature 11 tells us which logit to use. The results are given below in Table 14:

Table 14: Latent uncertainties as features are acquired in the synthetic experiments.

|  | Syn 1 | Syn 2 | Syn 3 |
|---|---|---|---|
| $\log(|\Sigma|)$ No Features | $1.366 \pm 0.551$ | $2.557 \pm 0.314$ | $19.468 \pm 1.294$ |
| $\log(|\Sigma|)$ All Features | $-25.002 \pm 0.306$ | $-21.004 \pm 0.398$ | $-44.015 \pm 2.032$ |
| $\log(|\Sigma|)$ Feature 1 | $-3.787 \pm 0.592$ | $-2.524 \pm 0.214$ | $19.236 \pm 1.227$ |
| $\log(|\Sigma|)$ Feature 5 | $-1.516 \pm 0.485$ | $2.489 \pm 0.317$ | $12.424 \pm 0.854$ |
| $\log(|\Sigma|)$ Feature 10 | $1.307 \pm 0.557$ | $0.020 \pm 0.417$ | $11.905 \pm 0.804$ |
| $\log(|\Sigma|)$ Feature 11 | $-3.144 \pm 0.626$ | $-1.792 \pm 0.236$ | $6.450 \pm 1.106$ |

We see that in all cases we have the most latent uncertainty when we have no features, and the least uncertainty when we have all features. If we were to acquire the uninformative feature for each dataset (10 for Syn 1, 5 for Syn 2 and 1 for Syn 3) we see that the latent uncertainty does not reduce significantly, IB has worked effectively and (mostly) disregards these features. We see in the case of Syn 3 that feature 11 reduces the uncertainty the most, showing that even though it does not reduce uncertainty in the label at first it is able to reduce the uncertainty in the latent space. It also significantly reduces the uncertainty for Syn 1 and Syn 2, although not as much as Feature 1 in those cases. An interpretive insight is that an effective acquisition reduces latent uncertainty, although this does not explain the *exact* ordering of acquisitions.

We augment this table by plotting TSNE projections of the latent space (Van der Maaten & Hinton, 2008), in Figure 15. The plot shows TSNE projections for Syn 3, where we color the data points based on the actual class and if a given feature is positive or negative. This has been done for Features 1, 10 and 11. Feature 11 is able to cluster the latent space more distinctly than the other features, showing it has a significant effect on the encodings. We also see that Feature 10 is able to cluster more distinctly than Feature 1, showing that it is also more important for prediction on Syn 3, as expected.

## M  ADDITIONAL BASELINE - GSMRL

Finally, we consider an additional baseline GSMRL (Li & Oliva, 2021). This is a hybrid approach that uses a Generative Model to improve the RL approach, this is done by providing intermediate reward based on information gain and providing additional information to the RL agent. We consider GSMRL against IBFA and Opportunistic RL on the Cube dataset. We provide the mean accuracy during acquisition in Table 15 and plot the acquisition curves in Figure 16. We see that GSMRL does not perform as well as IBFA or Opportunistic RL.

Table 15: Average Acquisition accuracies on the Cube dataset.

| Model | Average Acquisition Performance |
|---|---|
| IBFA (ours) | $\mathbf{0.904 \pm 0.001}$ |
| Opportunistic RL | $0.901 \pm 0.000$ |
| GSMRL | $0.823 \pm 0.002$ |

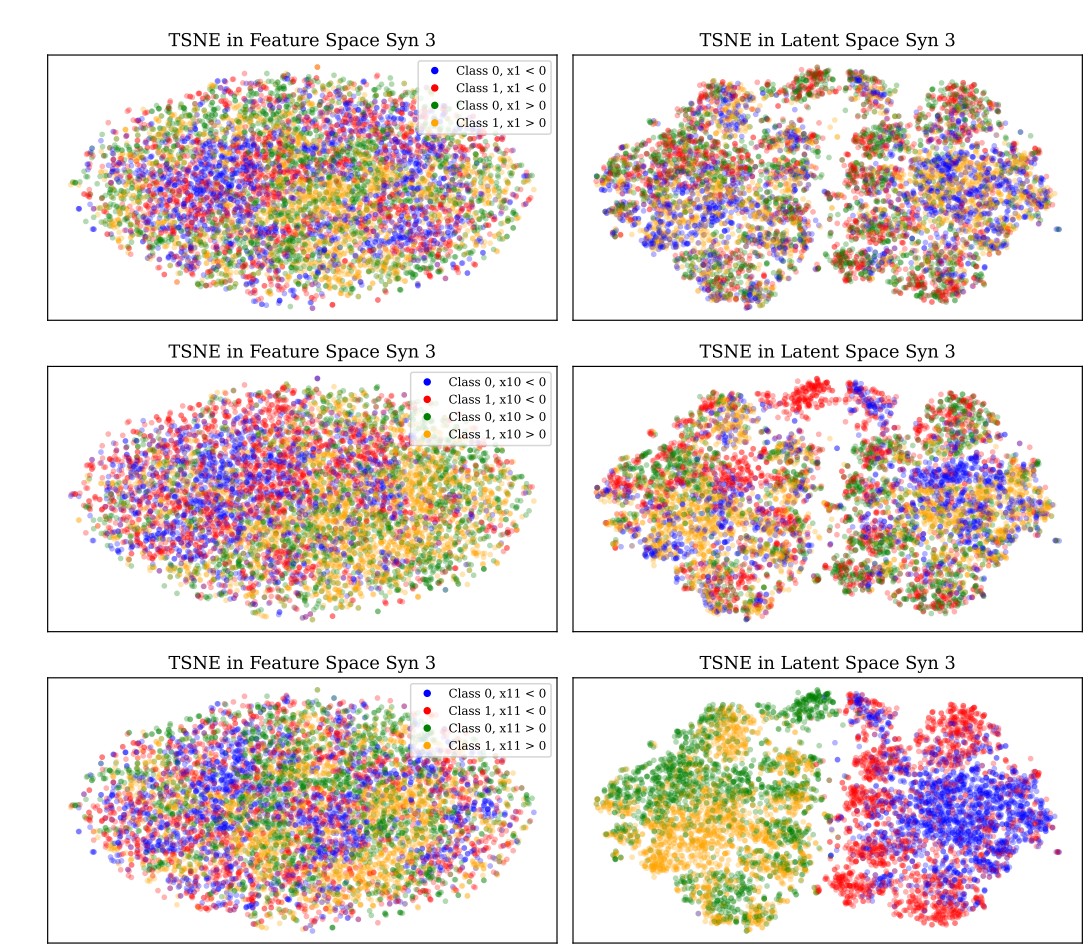

Figure 15: TSNE projections of the latent encodings, coloring is based on the class and if a given feature value is positive or negative.

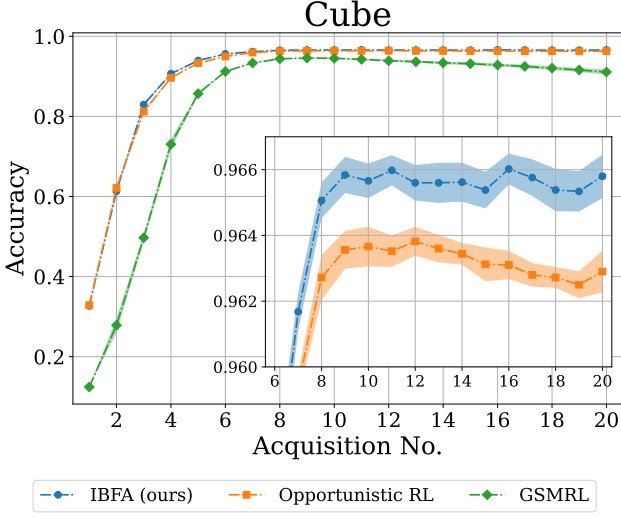

Figure 16: Acquisition curves on the Cube dataset, forIBFA, Opportunistic RL and GSMRL.

