# OpenReview forum: "Information Bottleneck for Active Feature Acquisition"
_ICLR.cc/2025/Conference — Submitted to ICLR 2025_

### Official Review · Reviewer_FV6m · 2024-10-15

**Soundness:** 3
**Presentation:** 3
**Contribution:** 3
**Rating:** 6
**Confidence:** 3

**Summary:**

This paper identifies shortcomings of CMI methods (i.e. it marginalizes unobserved features and thus makes myopic acquisitions, it doesn't distinguish between possible answers) and RL (unstable training) and proposes a simple yet effective method for AFA. They propose to uses several stochastic encoders to embed each feature individually into a latent space and then make predictions. An information bottleneck is added to regularize the latent space. For acquisition, they design an objective based on the sensitivity of the label to individual features and encourage them to pick features that distinguish high-likelihood answers. Experiments show that this method achieves better performance on both synthetic and real-world datasets compared to several baselines.

**Strengths:**

1. They identify several shortcomings of CMI methods and explain these limitations well with examples.
2. The proposed methods are simple but effective.
3. experiments are thorough.

**Weaknesses:**

1. This paper should compare to more recent RL baselines, such as [1].
2. The acquisition objective is based on the sensitivity of class wrt features. Could justify why using sensitivity is a good choice here?

[1] Yang Li and Junier Oliva. Active feature acquisition with generative surrogate models. In International Conference on Machine Learning, pp. 6450–6459. PMLR, 2021.

**Questions:**

1. How do we compute the integral over z in eq 1?

---

> ### Author Response · Authors · 2024-11-22
> **Official Response to Reviewer FV6m**
>
> We are grateful for the positive feedback and for the time reviewing. Following your review we have updated our manuscript to include an additional baseline - GSMRL. Please see our answers to all questions below.
>
> # Sensitivity of Classes with Respect to Features
>
> To explain the use of gradients, it is first useful to consider our full proposed acquisition objective (Equation 1):
>
> $R(\mathbf{x}\_{O}, i) = \mathbb{E}\_{p\_{\theta}(\mathbf{z} | \mathbf{x}\_{O})} \mathbb{E}\_{p\_{\theta, \phi}(y | \mathbf{x}\_{O})} r(y, \mathbf{z}, i)$
>
> The gradient-based subscoring $r(y, \mathbf{z}, i)$, is one component of this objective. The main contribution of our method is to perform the scoring using the expectations over $\mathbf{z}$ and $y$, since this way we are able to place more weight on the competing classes (average over $y$). Additionally, we are able to consider many latent samples (average over $\mathbf{z}$), effectively searching through possible latent realizations that match the current observations. These are the key contributions that address the weaknesses of CMI maximization we identified in Section 4. We construct $r$ so that it tells us the following:
>
> **If** the true value of the label is $c$, how important would feature $i$ be for confirming that prediction in the context of an entire sampled latent vector $\mathbf{z}$.
>
> We can think of our method as considering possible instances that match the current observations, finding which measurement would be the most useful for identifying the label for that instance, and then averaging over them.
>
> This is under the assumption that $r$ is able to tell us this information. We chose to use a gradient-based sensitivity, since this is scalable and tells us what local changes lead to the largest change in prediction, that is, which latent variables have the largest effect on the prediction. These gradients are not taken with respect to the features, since missing features being replaced by observations is not a continuous change, and thus the gradient will not be an accurate representation of the label’s sensitivity with respect to the feature. Instead, the gradient is taken with respect to the latent encodings. We designed the encoding architecture so that each latent component is only encoded by one feature, allowing us to trivially accumulate the gradients for the latent components to score each feature.
>
>
> # How Integrals are Calculated
>
> The integrals over the latent variable in our work do not have an analytical solution, and are high dimensional. Therefore, we solve using Monte-Carlo estimates, which is standard practice for integrals of this nature.
>
> We have also conducted sensitivity analyses to examine how important the number of samples are in these estimates, these can be found in Figures 11 and 12 in the Appendix. They show that both at training and evaluation time the performance improves with more samples, and plateaus at around 100.
>
>
> # Comparison to GSMRL
>
> We appreciate the feedback about comparing to a more recent RL baseline. Whilst we compare to state-of-the-art AFA methods (DIME from ICLR 2024 and GDFS from ICML 2023), we chose to compare to Opportunistic RL, since it is regularly used as an RL baseline (see DIME and GDFS) and is not a hybrid method like GSMRL (Li et al. 2021), so we can compare directly to the overall RL approach. Following the feedback we have tested GSMRL on the Cube dataset. We have used the public implementation of GSMRL (https://github.com/lupalab/GSMRL), using the exact hyperparameters suggested in the public repository for the task. We provide the average accuracies during acquisition below, we also include the same numbers for IBFA and Opportunistic RL (from Table 3) for easy comparison:
>
> | Model | Average Acquisition Accuracy |
> | --- | --- |
> | IBFA (ours) | $0.904 \pm 0.001$ |
> | Opportunistic RL | $0.901 \pm 0.000$ |
> | GSMRL | $0.823 \pm 0.002$ |
>
> IBFA and Opportunistic RL outperform GSMRL. We have also added this Table and the acquisition curves to the updated manuscript to show this result.
>
> # References
>
> 1. Li, Y. and Oliva, J., 2021, July. Active feature acquisition with generative surrogate models. In International conference on machine learning (pp. 6450-6459). PMLR.

---

> > ### Author Response · Authors · 2024-11-29
> > **Is there anything outstanding to discuss?**
> >
> > Dear Reviewer FV6m,
> >
> > We are just checking in to see if we have addressed all your questions, or if there is anything outstanding we can address in the remaining discussion period. Many thanks again for the time and effort writing the review and providing feedback.

---

### Official Review · Reviewer_PiTL · 2024-10-18

**Soundness:** 3
**Presentation:** 3
**Contribution:** 2
**Rating:** 3
**Confidence:** 4

**Summary:**

This work considers the active feature acquisition (AFA) problem, where you make selections sequentially based on partial information to achieve high prediction accuracy with a small budget. The authors begin by describing the main existing approaches that use RL or greedily maximize the conditional mutual information (CMI), and point out specific issues that arise with the CMI approach: namely 1) a failure to account for how selections might pay off after one or more additional selections, and 2) a focus on minimizing entropy rather than other measures of predictive uncertainty.

They then propose a new approach that they call "information bottleneck for feature acquisition" (IBFA). The idea is to train a model that predicts $y$ given a partially observed input $x_S$ via a stochastic latent variable $z$. The model is trained to make accurate predictions subject to a penalty that pushes $p(z \mid x_S)$ towards a prior $p(z)$, typically with a very small weighting $\beta \in [10^{-3}, 10^{-2}]$. At inference time, they make selections based on an expected gradient sensitivity measure of the model's predictions to different latent dimensions, which are set up to correspond to different input dimensions.

The results show encouraging performance in two senses: 1) fast selection of the correct features in simulated datasets, and 2) high accuracy under small acquisition budgets for several real datasets.

**Strengths:**

- The beginning of the paper is nicely written. It covers many related works, points out the main issues with each approach, and is generally easy to follow.
- The section on limitations of the CMI approach is useful. It makes the issue of myopic selections more concrete through a specific and intuitive example with the indicator variable.
- The results are consistently strong across a large number of datasets.
- The IBFA approach seems to have some underlying idea that’s useful, although I’m not sure what it is from reading the paper.

**Weaknesses:**

My main issue with this work is that the IBFA approach is also greedy, only it greedily maximizes a new criterion with no clear probabilistic interpretation. This was a bit disappointing, because the work began by making some reasonable points about flaws in the greedy CMI approach, and as a reader I was hoping for a nice clear solution to those problems. Instead, the authors propose a gradient-based sensitivity measure that seems a bit random, which they calculate in expectation over a stochastic latent variable, and it's not obvious why this works or what it has to do with the problems they discuss for the greedy CMI approach. It empirically works well so there's clearly something to it (although I have a couple qualms with the experiments that I'll mention below), but as a reader I can't tell what's useful about IBFA.

I'll describe a few issues from throughout the paper below.

In the exposition of the AFA problem (Section 3), there are a couple aspects of the presented objective that seem odd. 1) The $S$ optimization needs to be over a distribution of data points, not a single data point as currently shown. 2) The measure of predictive accuracy can arguably be one of a few things, but the max class probability is confusing - why not the probability of the true class? And 3) the constraint $|S| \leq B$ and penalty $\lambda |S|$ look redundant - you could formulate the problem with both, but your experiments only seem to consider the constrained problem (unlike some works that also learn stopping criteria).

In the discussion of limitations of the CMI approach, there are a couple parts that could be improved:

- The claims about the necessity and sufficiency of "considering values of the other unobserved features" aren't substantiated beyond the simple indicator scenario. Relatedly, it's unclear whether IBFA incorporates this idea of considering other feature values beyond the current candidate. Its acquisition criterion considers multiple sampled values for each $z_{G_i}$ which is somewhat like considering multiple values for the unobserved features, but it also samples latents for the known features, and the sampled latents for unobserved features fail to condition on the observed features. So ultimately I'm not sure this whole perspective is relevant to IBFA.
- In Proposition 2, the criterion you present 1) is intractable, 2) is only claimed to be applicable to this indicator example, and 3) has little to do with IBFA. The failure cases of this approach are easy to see: in a scenario with duplicated features, this measure would assess every feature as having zero value because $I(X_i ; Y \mid x_O, x_U) = 0$ for all $x_U$. Your general point that *selections that fail to account for future selections might be bad* is clear, but this idea doesn't seem to generalize beyond the toy problem and is unrelated to your solution.
- In the part about how minimizing entropy is bad, it would be good to be more precise. In what sense is [0.7, 0.15, 0.15] preferable to [0.5, 0.5, 0]? The current vague explanation is that it's "more favorable for making a prediction." If you formalize that as an alternative uncertainty measure, you could perhaps greedily minimize that at each step and improve over existing CMI methods. You could probably even incorporate it into an existing method like DIME. If you think your approach is effectively doing some related reformulation of the greedy selection criterion, it would great to say so explicitly.

About the IBFA method design:

- Like I said above, the main issue I see with IBFA is a missing probabilistic interpretation for the greedy selection criterion. There's arguably some interpretation to the learning objective (although I think your math is wrong, see below), but I don't see any reasonable interpretation to your selection criterion $R(x_O, i)$ that relies on a heuristic gradient sensitivity measure. The main justification I saw in the paper was that it pays more attention to high-probability classes, but there are easier ways to achieve this.
- The information bottleneck (IB) idea seems non-integral to the method. I noticed you have an experiment studying sensitivity to $\beta$, which often results in very small values relative to the main predictive objective (more on that below). The reasoning provided in Section 5.3 is also quite vague: "we reason about the acquisition in the latent space, we want to use representations that only contain information relevant to predicting the label." How would the network be incentivized to learn anything else? Also, even though the IB bit helps, it's not central (the results are reasonably good without it), so it seems like an odd choice to name the IBFA method after it. A more accurate name should probably allude to the gradient-based selection criterion, although admittedly I'm not sure what you would call it because it seems a bit heuristic.
- In designing the learning objective, the authors take motivation from the greedy CMI approach when choosing the term $\mathbb{E}[- \log(p_\phi(y \mid z))]$, and they seem to think of this as a form of regularization. That's a very odd perspective, and I'd like to share another interpretation: the first term is simply an upper bound on the standard predictive loss. You can see this in the following derivation: $- \log p_\phi(y \mid x_S) = - \log \int p_\phi(y \mid z) p_\theta(z \mid x_O) \leq - \int p_\theta(z \mid x_O) \log p_\phi(y \mid z)$ (the inequality is just Jensen's). The third term here is what you initially show in line 312 (which is a potentially loose upper bound), and the second is what you eventually use in eq. 3. The exposition of your objective is overly complicated, it seems like you're just learning to predict $y$ via $z$ while imposing some posterior variance via a KL divergence penalty.

There are a couple mistakes in the derivations behind IBFA and its objective:

- In your appendix E proof of Theorem 1, I think your problems start on line 1212 when you conflate underlying probability distributions with your parametric model. Under your model you've factorized the prediction as $p_\phi(y \mid z) p_\theta(z \mid x_O)$, but that's not the same thing as conditional independence of the form $p(y \mid z, x_O) = p(y \mid z)$. The observed variables $x_O$ could provide information about $y$ not captured by $z$ depending on how it's learned. For some arbitrary $p_\theta(z \mid x_O)$ distribution like the one you have at initialization, $z$ certainly does not provide all of $x_O$'s information about $y$.  I believe everything after this point is therefore wrong, including the overall claim.
- In your appendix F proof about the IBFA objective function, I think your problems start on line 1241 when you claim that $\int p_\phi(y \mid z)p_\theta(z \mid x_O) = p(y \mid x_O)$. Again, you're conflating your parametric model with underlying probability distributions, and you can't guarantee that this holds. I believe this invalidates the later parts of this section and the claim in Theorem 2.

About the experiments:

- I saw the IB ablations in the main text and appendix, and they mostly confirm my concern that the IB part of IBFA is non-essential. In Table 2 it seems helpful, but note that 1) the method basically works without this ingredient, 2) it could be helpful mainly because these are tiny synthetic datasets. In Figure 10 we see that the best $\beta$ values are very small, even for these toy datasets. In the real-data results in Table 4, the benefits of IB are very small - in one case it even slightly hurts performance. Tables 12-13 also reflect that the $\beta$ values here are even smaller than for the toy datasets, with $\beta = 0.001$ being the largest possible value. Overall, IB seems more like a secondary implementation detail of your method, and the paper's focus on it is a bit misleading.

**Questions:**

Some questions about the experiments:

- I wanted to sanity check some of your results, and I noticed that the DIME and GDFS accuracies for MNIST are both worse here than in the original works (you can see this for 5 and 10 acquisitions). Why do you think that is?
- I noticed that a pre-processing step for many of your real-data experiments is reducing the number of candidate features using STG. You claim it's so EDDI and the VAE method can work, but I'm curious if it's also hard for IBFA to operate on larger numbers of features. If so, that would be somewhat limiting and merit some discussion.
- I saw that in Table 5 you provide "scaling laws" (this is called computational complexity) for each of the algorithms. I'm wondering if you could provide wall-clock times, because it's hard to get a sense for how slow/fast IBFA is, particularly at inference time where it requires 1) potentially many latent samples $z$ and 2) iterating over classes to calculate the gradient sensitivity measure.
- I noticed that you adjusted the class balance for MiniBoone. That's odd, are your results different under the original version of the dataset?
- I noticed that for IBFA you pre-processed features using a copula transform. Did you try ablating this to see if it's important for strong performance? If so, that would be interesting and merit further discussion. I would also be curious to know if it helps to incorporate into other methods, which seems pretty straightforward to try.

A few other questions about IBFA:

- How sensitive is performance to your specific gradient sensitivity measure in eq. 2? For example, what if all the norms were squared? What if you didn't normalize?
- Perhaps you can try providing intuition for the gradient sensitivity measure by considering a scenario example where the predictor conditioned on the available features is linear in the latents for the unobserved variables?
- How would you incorporate feature costs into IBFA? Some existing methods to do by weighting expected improvements in loss against the cost of acquiring features (see DIME for example). Doing so might be tricky here because the units of your selection criterion $R$ are unclear.

---

> ### Author Response · Authors · 2024-11-22
> **Official Response to Reviewer PiTL Part 1**
>
> We are incredibly grateful for the very detailed feedback and the time spent on the review. Following the review we have updated our manuscript by improving clarity in the notation throughout our paper to make the proofs clearer; we have also updated Section 5.3 to make the motivation behind our loss clearer. We answer your queries below.
>
> # Clarifying the Notation in the Proofs
>
> We appreciate this point about our proofs, this is a case of overly-simplified notation. In various places we have dropped subscripting probability densities with the model parameters. In these cases the intended meaning was the likelihood as calculated by the model, and not the true underlying density (since this is not available). For example:
>
> $p(y|\mathbf{x}\_O) = \int p\_{\phi}(y | \mathbf{z}) p\_{\theta}(\mathbf{z} | \mathbf{x}\_O) d\mathbf{z}$
>
> Should instead be written as:
>
> $p\_{\theta, \phi}(y|\mathbf{x}\_O) = \int p\_{\phi}(y | \mathbf{z}) p\_{\theta}(\mathbf{z} | \mathbf{x}\_O) d\mathbf{z}$
>
> To indicate that this is the likelihood calculated by the model. The same applies to any mutual information terms $I(X; Y)$ should be written as $I\_{\theta, \phi}(X; Y)$.
>
> We apologise for any inconvenience our choice of notation had, and will make our notation more precise in our updated manuscript to reflect this.
>
> The use of the model to calculate these terms is justified since in real applications we do not have access to the true distributions, and therefore need to approximate them with a model. For example, classification is framed as maximizing the log-likelihood of the true class **calculated by the model**. Similarly, Deep Variational Information Bottleneck (Alemi et al. 2017) uses deep networks, and the IB loss is calculated using the model’s distributions, here the loss is written as:
>
> $I(Z ; Y | \theta) - \beta I(X ; Z | \theta)$
>
> In other works, the mutual information is written as $I\_{\theta}(Z; Y)$. In both cases it is clear that this is calculated with respect to the model by including $\theta$ in the term, we will opt for using a subscript to make this clear. Since these terms are calculated using the model’s distribution, introducing the modelling assumption to simplify the terms is a valid operation, which we do in both proofs. An example of this is EDDI (Ma et al. 2019), which uses the VAE approximation in their Appendix proofs to yield a tractable acquisition objective which approximates the true mutual information.

---

> ### Author Response · Authors · 2024-11-22
> **Official Response to Reviewer PiTL Part 2**
>
> # Importance of Information Bottleneck
>
> The $\beta$ hyperparameter in Information Bottleneck is typically quite a small value, as correctly identified we tried values between 0.0001 and 0.001 when hyperparameter tuning. This is in line with the expected order of magnitude of $\beta$, for example see Deep Variational Information Bottleneck (Alemi et al. 2017), where the values of $\beta$ are just as small. From that paper for instance Table 1 $\beta = 0.001$, Figure 1 values of $\beta$ are investigated from $10^{-9}$ up to $1.0$, with any values higher than $10^{-2}$ showing poor performance, Figure 2 also visually shows this. That paper also demonstrates there are critical values that are too high, e.g. in their MNIST results if $\beta > 0.01$ the model fails because not enough information is present in the latent variable to distinguish between 10 classes. Thus our values of $\beta$ are within an expected range. It is similar to weight decay, the weight decay parameter is also typically very small in an optimizer, the reason for both values being small is that regularization is only useful if the model is able to learn in the first place, therefore the predictive loss has a larger weight in the total loss.
>
> Our results demonstrate that when $\beta$ is set to 0.0 (no IB regularization), IBFA performs worse. As mentioned, in Table 2 for each synthetic example the number of acquisitions required increases by approximately 0.5, which is more than a 10% increase. Without IB in these examples the performance is worse than DIME, as well as GDFS and Opportunistic RL in some cases. This is further shown in Figure 10, our sensitivity analysis, where the number of acquisitions is large if $\beta$ is too high and if it is too low. There is an optimal value which is non-zero and moving away from this value in either direction makes the performance worse. The same tests on real data in Table 4 show that in relatively noiseless regimes such as MNIST and Fashion MNIST IB is less beneficial, but in noisy regimes like the TCGA Cancer task the performance improves with IB.
>
> Regarding the reasoning behind using IB to regularize the latent space, it is well established that IB is designed to encode $\mathbf{x}$ to $\mathbf{z}$ to only contain information about the label (see the introduction of the original IB paper Tishby et al. 2000 for example). This means that the features cannot be reconstructed from the latent variable and any noise at the feature level has been removed during the encoding process. You have correctly pointed out that the network cannot learn anything unrelated to the prediction task, however, the IB principle is more about removing the irrelevant information that is present in the features. And this is our motivation for using IB, since our acquisition objective is calculated using samples from the latent space we wish to use information that is only relevant to the label. On top of this, within the IB framework we are able to derive a further regularization term in our loss to make the latent space more conducive to effective acquisitions as given by Theorems 1 and 2 (we discuss this more later). Therefore we have given both theoretical motivation for the use of IB and empirically shown it is a necessary component of the model.
>
> Regarding renaming the method (and subsequently the paper), we are not against this. However, we disagree that the gradient-based component is the main focus or contribution of our work. We discuss our objective in more detail later, briefly, the gradient-based subscoring in $r$ is only one part of the overall objective. The more fundamental part of the objective is using both an expectation over the latent space to capture effects of non-observed features, and an expectation over the labels to distinguish between competing classes. Therefore a more appropriate name might be “Active Feature Acquisition via the Latent Space”. Our preference is to IB in our method name, since for people familiar with IB it makes a clear implication that we use a latent space and that it is a stochastic encoding. Additionally, IB was an important part of our model (performance is worse without it), and we are able to motivate the loss function we use to train within the IB framework.

---

> ### Author Response · Authors · 2024-11-22
> **Official Response to Reviewer PiTL Part 3**
>
> # Acquisition Objective and Model Architecture
>
> There is an important distinction to be made here between greedily maximizing an objective and greedy acquisitions. Nearly **all** methods will greedily maximize some objective at inference time, even Deep RL methods at inference time will choose the action that maximizes their value or Q function. The important question is what does the objective measure, and is maximizing this leading to myopic behavior. RL methods train such that their value function takes long-term reward into account, so even though it is being maximized the selections made are not greedy. On the other hand, CMI maximization leads to greedy acquisitions, since it will never select a feature that gives no immediate information about the label but might give us information about which features to measure next. Whilst IBFA greedily maximizes our new criterion, this does not mean the acquisition itself is greedy, and in fact the purpose of our work was to design this new architecture and criterion that fundamentally does not make greedy acquisitions.
>
> To see why this is the case we first consider the architecture. Our encoding architecture is factorized so that each latent component can only be associated with a single feature. Therefore, if we can identify which latent components we most want to reduce the uncertainty for, then we know which feature needs to be measured. Now we consider our criterion, where the order of expectations has been flipped for the purpose of this discussion:
>
> $R(\mathbf{x}\_{O}, i) = \int p\_{\theta}(\mathbf{z} | \mathbf{x}\_{O}) \bigg( \sum\_{c \in [C]} p\_{\theta, \phi}(Y=c | x\_{O}) r(c, \mathbf{z}, i) \bigg) d\mathbf{z}$
>
> We construct $r$ so that it tells us the following:
>
> $r(c, \mathbf{z}, i)$ tell us that **if** the true label is $c$, how important is feature $i$ for confirming that prediction in the context of an entire sampled latent vector $\mathbf{z}$.
>
> Under a subscoring that matches this, our criterion works in the following way:
>
> For a given latent sample (which represents one possible instance with the current observations in the latent space), if the true label were $c$, score the latent components based on how important they are for confirming this. Sum over all possible labels, weighted by current predicted likelihoods so that we give more weight to more likely classes. And then we average over many samples from the latent space, effectively searching through possible instances that match the current observations. This allows us to score each group of latent components and our architecture makes it trivial to link the highest scoring group to a feature. The key components are the two expectations, as well as the architectural and training decisions that allow the objective to work effectively, rather than the exact form of $r$.
>
> This criterion and architecture addresses the identified issues of CMI as follows:
>
> 1. Myopic acquisitions are avoided since many latent samples are used which represent latent realizations of unobserved features, which we identify as a necessary condition for non-myopic behavior. Note: there is no technical fault in sampling the latents associated with observed features, they are still sampled to make full predictions, and uncertainty in the latent variable is still non-zero but reduced. Our synthetic results confirm that IBFA does not make myopic acquisitions as the only model to consistently select feature 11 first in the synthetic datasets.
>
> 2. The issue of not distinguishing effectively between high likelihood classes is addressed by using the current predictions as weights in the criterion. This is also empirically shown since removing the weighting produces worse performance (Table 4).
>
> Intuitively, we can think of our method as considering the possible instances that match the current observations, finding which measurement would be the most useful for identifying the label for that instance, and then averaging over these. So whilst we do not believe a probabilistic interpretation is necessary for a model to be performant, we disagree that there is no interpretation of our criterion.
>
> Crucially, this is under the assumption that $r$ is able to give us the information we required. $r$ cannot be any function, we chose to use a gradient-based sensitivity, since this is scalable, and tells us locally which latent variables have the largest effect on the prediction, so it is not entirely heuristic. This builds on the requested intuitive explanation using a linear predictor (the second to last question in the review) since for each latent sample we can consider a locally linear approximation of the predictor. We stress again that whilst there were design choices constructing $r$, its gradient based scoring is still not the main contribution of our method, but a solution that worked well with the overarching method.

---

> ### Author Response · Authors · 2024-11-22
> **Official Response to Reviewer PiTL Part 4**
>
> # Problem Formulation
>
> We address each point separately:
>
> 1. The core purpose of AFA is that we do not select the same $S$ for each test instance. Instead a model is able to adapt to each test subject based on the evolving understanding of that specific instance. Therefore the $S$ optimization **must not** be carried out over a distribution of many data points, since we do not desire a fixed global ordering (the fixed MLP in our experiments uses a fixed ordering and performs poorly as a result). Instead, we desire a separate optimization for each test point (giving a separate $S$ for each instance). The long-term goal provided in Section 3 is for one test point, and it generalizes to a test set by running separate optimizations for each instance.
>
> 2. It is important to recognize here that the $S$ optimization given is what an AFA model is solving at test time. It is therefore impossible to know what the true class is, and so it cannot be maximized by the model. Instead it has to increase the confidence in its own predictions, as measured by its predictive component $p(Y=c | \mathbf{x}\_{O \cup S})$. This is what happens in the real world. We cannot know the true label, there is not an oracle giving it to us, so we have to make measurements until we have confidence in our predictions, if we knew the true label we would not need to acquire any features. Note: this assumes the predictive power of the model is well-calibrated in the first place, i.e. it has good test predictive performance, but also that when there are few features it does not make overly confident incorrect predictions.
>
> 3. By including both the budget and the penalty we are remaining more general, we can remove the budget constraint by setting $B=\infty$, and remove the penalty by setting $\lambda=0$. We have indeed set up our experiments to show how each model would perform if we were to stop acquisition at different points. We see in Figure 3 that if we were to stop the acquisition at certain points then the overall accuracy/AUROC of IBFA is higher. However, the inverse is also true, if the aim is to each a certain level of accuracy/AUROC we see that IBFA can achieve this in fewer acquisitions, so the penalty is not entirely redundant.

---

> ### Author Response · Authors · 2024-11-22
> **Official Response to Reviewer PiTL Part 5**
>
> # CMI Shortcomings: Indicator Example
>
> We discuss the indicator example in the paper because it concretely provides insights into why CMI maximization can fail (this has been identified in the review as a strength). The purpose of proposition 1 is to show an example where CMI fails and more importantly what the failure is (in this case the failure is not selecting the indicator first, a feature that provides no immediate value but significant long-term value). The purpose of proposition 2 is to show that with the knowledge from proposition 1, **it is possible** to construct a criterion for this problem, that when maximized will be optimal. It is true that we do not use this criterion (since as we stated, it is intractable). However, the purpose was not to give an exact objective that would generalize to any data, but to explore the CMI failure cases in a fully analytical setting that is relatively straightforward to understand; providing insight into the reasons for failure and a possible solution. We disagree that the example is unrelated to IBFA because we take the **ideas** that we discuss in the indicator example, using them as motivation for our solution, where there is a requirement to be more robust. As we have shown previously, considering the values of unobserved features in the criterion is carried out in IBFA by sampling the regularized latent space. And similarly placing more weight on higher probability classes is addressed in our criterion by taking the average over $p\_{\theta, \phi}(y | \mathbf{x}\_{O})$.
>
> A phenomenon from the indicator example that does generalize is the importance of considering interactions between features, and not features in isolation. Individually a feature may not have much predictive power, however if it is jointly known with another feature (or features) the predictive power can increase significantly. This is a common phenomenon, in genetics for example it is known as epistasis, where the effect of a gene mutation is dependent on the presence or absence of a mutation in another gene. Another recent paper (Kim et al. 2024) calls these interactions synergistic, where variable $X\_1$ gives more information about $Y$ when conditioned on $X\_2$ than it does unconditionally. These interactions would be missed by CMI maximization, whereas a criterion that considers values of unobserved features has the ability to capture these interactions.
>
> We can construct another example, let $X\_1 \in \\{-1, 1 \\}$, $X\_2 \in \\{-1, 1 \\}$, and let $y = x\_1 x\_2$ (this is effectively XOR). If for all other features we have $x\_i = 0.0001 y + \varepsilon$, $\varepsilon \sim \mathcal{N}(0, 1)$, then features 1 and 2 can only ever give information about $y$ if the other is known. And so CMI maximization would focus on acquiring the noisy features before the useful features, because they are jointly predictive but not individually. The purpose here was to show that despite the exact details of the indicator example not necessarily generalizing, the high-level ideas do.
>
> Finally, on the indicator example, you are quite right that the objective in proposition 2 would not generalize outside of the indicator example due to highly correlated features giving zero value. Firstly, we reiterate that we do not use this criterion, and that it was never intended to be a fully general criterion, the main purpose was to demonstrate that in this indicator example it is possible to construct a criterion that considers the unobserved features and is optimal (i.e. it is sufficient), giving motivation for our solution. That being said, there are ways it can be fixed. For example, we could instead extend it to take a sum over all possible subsets of $U$ including the empty set. This criterion would tell us how much information a feature will give us if we had any subset of the unobserved features (including no additional features), in addition to the observed ones. This would be even more intractable, but the main takeaway again is that it is not enough to consider the mutual information with no conditioning on the unobserved features, and this insight does generalize to situations like those described above where we have interactions between features. Another **practical** solution for the criterion in proposition 2 could be to include a preprocessing feature selection step to remove highly correlated features. There is no reason why an AFA method has to act in isolation, and using feature selection prior to AFA would both reduce the search space and remove correlated features. But we stress again, the main purpose of propositions 1 and 2 was not to create a criterion that generalizes. But to provide valuable insight into the CMI failure cases, why it is suboptimal and what a possible solution could look like, motivating the actual design of our method.

---

> ### Author Response · Authors · 2024-11-22
> **Official Response to Reviewer PiTL Part 6**
>
> # CMI Shortcomings - Entropy Minimization
>
> Regarding the point about the weakness of entropy, the purpose of classification is identifying a single class. And so we argue that entropy is not a useful measure of uncertainty in the classification setting, because it is measured across all classes. Instead, a better measure of certainty is the predicted probability of the most likely class $\max\_{c} p\_{\theta, \phi}(Y=c | \mathbf{x}\_{O})$. We apply the same concept of certainty to our formalism in Section 3: We want to continue to acquire features that make the model more confident in its prediction, but we care only about the prediction for what the label is, and not the prediction of what the label is not. The difference between $[0.7, 0.15, 0.15]$ and $[0.5, 0.5, 0.0]$ is that in the first distribution there is a relatively confident prediction for what the class is (class 1), whereas in the second there is only a confident prediction for what the class is not (class 3). As we see, minimizing entropy can be achieved by further reducing already low probabilities rather than distinguishing between likely classes. Of course, in real-world settings, it is important to rule out options, but the analogy in the medical setting would be suspecting a fracture in the leg and requesting an arm X-Ray to confirm the problem is not in the arm. As more classes are added, more extreme examples are possible, for example if we have 101 possible classes with two distributions:
>
> - 0.5 for two classes, and 0 for the rest, entropy = 0.693
> - 0.9 for one class and 0.001 for the rest, entropy = 0.786
>
> We used this as motivation for weighting our criterion by the current probabilities, to place more focus on disambiguating the most likely classes. We see in Table 4 that empirically this improves the results.
>
> You are correct it might be possible to formalize this as an alternative uncertainty measure, for example a selection criterion for a general model could be
>
> $\text{argmax}\_i \mathbb{E}\_{p\_{\theta}(x\_i | \mathbf{x}\_{O})}\bigg[\max\_{c} p\_{\theta}(Y=c | \mathbf{x}\_{O}, x\_i)\bigg]$,
>
> instead of minmizing the entropy. However, greedily maximizing this will still make myopic acquisitions because it does not consider the values of unobserved features. And since the whole purpose of our approach is to be a non-greedy alternative to CMI and RL, this is not desirable. Incorporating this into other methods like DIME is outside the scope of this work, which concerns our IBFA method  (especially given this uncertainty measure would still make myopic acquisitions).

---

> ### Author Response · Authors · 2024-11-22
> **Official Response to Reviewer PiTL Part 7**
>
> # Loss Function
>
> You are completely correct that one interpretation of our loss is learning to predict $y$ via $\mathbf{z}$ and imposing a KL penalty on the latent space. The same argument can be made about VAEs, saying that they are learning to reconstruct $\mathbf{x}$ via $\mathbf{z}$ while imposing a KL penalty on the latent space. However, it does not mean that the Evidence Lower Bound interpretation is incorrect, and in fact using the ELBO interpretation provides more theoretical motivation for the loss, the use of a KL divergence rather than Jensen-Shannon divergence for example. The same argument applies to our loss function. One correct, simpler interpretation is the one you have given, another correct interpretation is the one we have given, using the IB framework. However, the IB framework provides more theoretical grounding for our loss, the terms involved and more importantly how we can adapt it for our situation. For example, the IB interpretation explains the use of a KL divergence over any other regularizer. Additionally, the purpose of Theorems 1 and 2 is to explain why we have moved away from the Deep Variational Information Bottleneck loss (Alemi et al. 2017). The VIB loss is:
>
> $L\_{\text{VIB}} = \mathbb{E}\_{p\_{\theta}(\mathbf{z} | \mathbf{x}\_S)} \bigg[-\log(p\_{\phi}(y | \mathbf{z}))\bigg]+ \beta D\_{\text{KL}}(p\_{\theta}(Z|\mathbf{x}\_S) || p(Z))$
>
> Where one sample is taken in the expectation. This is the standard loss for deep IB methods given on line 312. We instead use our custom loss:
>
> $L = -\log \big( \mathbb{E}\_{p\_{\theta}(\mathbf{z} | \mathbf{x}\_S)}\big[p\_{\phi}(y | \mathbf{z})\big] \big) + \beta D\_{\text{KL}}(p\_{\theta}(Z|\mathbf{x}\_S) || p(Z))$
>
> Where we have moved the expectation inside the logarithm and take multiple samples. It is important to explain why we have made this change and using the interpretation of a predictive loss with KL regularization does not provide any motivation for moving away from the established VIB loss. However, within the IB framework we do motivate this, since Theorem 1 says that further regularization of the latent space is possible to make it more conducive to acquisition, and Theorem 2 says that our loss is the IB loss with this added regularization term. The change in the loss does not improve predictive performance (hence why the simpler interpretation cannot explain why we make the change), however it does improve acquisition performance. This is seen in ablations when the VIB loss is used, see the "1 Train Sample" row in Tables 2 and 4.
>
> For completeness in our response, an alternative, more intuitive reason for this change is as follows: We desire to construct a latent space where there are separated regions in the latent space associated with each class. Such that initially we have a large latent variance, covering all of these regions. As more features are acquired the latent variance is reduced. However, if we train with only one sample of the latent space with subsampled feature subsets (VIB loss), then for the case of no features the model should predict high uncertainty for every sample (rather than the integral over samples). And so the label uncertainty is encoded into the position of the latent space which will create regions of the latent space that predict no class. The latent encoding with no features is around one region with a low latent variance where every sample predicts no class. Crucially taking samples in the latent space to make an acquisition will not explore a diverse set of possible latent realizations anymore and acquisitions suffer. Theorems 1 and 2 formalize this intuitive reasoning.
>
> We appreciate the feedback on this section, and as a result have improved the clarity and motivation for these theorems in the context of our model and its training.

---

> ### Author Response · Authors · 2024-11-22
> **Official Response to Reviewer PiTL Part 8**
>
> # Additional Questions
>
> - STG: In our experiments we saw that IBFA carries out inference in a reasonable amount of time (see our approximate wall-clock times). Note: increasing the number of features also has a significant effect on the training times for Opportunistic RL, DIME and GDFS, not just the inference times for EDDI. We chose to include a feature selection preprocessing so that within our computational resources (a single Nvidia Quadro RTX 8000, modest by modern standards) we were able to run a large variety of models and datasets rigorously. That is, hyperparameter tuning with multiple seeds, using multiple runs to have uncertainty estimates etc. It is not a new experimental choice to reduce the number of features in AFA, for example Li et al. 2021 also reduce the size of their MNIST experiment “to accommodate baselines such as EDDI that have trouble scaling”. The number of features is also within the range in the majority of current AFA papers. For example Figure 2 from GDFS or Figure 2 from DIME, the standard ranges tested for real tabular datasets are on the order of 20. We stress that we have used the same experimental setting for all models equally. Additionally, this is the first iteration of a method that solves AFA using our new approach. Scalability is one potential area of research, and is in fact an active area of research in AFA (see Towards Robust Active Feature Acquisition by Li et al. 2021 B for example), it is not the case that all other models are scalable. Finally, we argue that in many real-world applications there are no limitations of using a hybrid approach, a single model making up the entire pipeline is not a requirement. In this case the practical approach of a combined **scalable** feature selection method followed by an advanced AFA method might be preferable.
>
> - MNIST: The discrepancy in performance is likely due to our feature preprocessing as mentioned above. Since we have limited the number of features, there are some features that are not available to the models, and so the prediction performance is slightly worse. Note: All models were evaluated equally within the same experimental setup, which was used so that within our limited resources we could conduct a large variety of experiments on multiple baselines and datasets. We also respectfully point out there is a larger discrepancy between our Opportunistic RL results on MNIST and those from DIME and GDFS, our results are almost 50% greater in accuracy, and match the result from Opportunistic RL more closely (see Figure 2 in Opportunistic RL).
>
> - Wall-clock Times: We have looked over the training and inference times and provided a table of wall-clock times to augment the computational complexities. The table below gives the aggregated times over the datasets as orders of magnitude. Note the caveat is that many methods were running in parallel on the same GPU, which will affect the exact times, hence reporting the times as orders of magnitude.
>
> |Model | Full Training | Full Inference |
> | --- | --- | --- |
> | DIME | 10s Hours | Minutes |
> | GDFS | 10s Hours | Minutes |
> | EDDI | Hours | Hours |
> | Opportunistic RL | 10s Hours | Minutes |
> | VAE | Hours | Hours |
> | | | |
> | IBFA | Hours | 10s Minutes |
>
> These empirical observations match the computational complexities provided in Appendix 5.
>
> - MiniBooNE: We found that using the full class balance was too “easy”, all models were able to achieve very high AUROC with very few features and the results were indistinguishable between models. By enforcing class balance we have made the task harder (not easier), with less data and more balance. In this setting **where all models have been evaluated equally**, we see that IBFA performs best.

---

> ### Author Response · Authors · 2024-11-22
> **Official Response to Reviewer PiTL Part 9**
>
> - Copula: We have rerun a subset of experiments on IBFA without the copula transform. Note that since the features are normally distributed for the Synthetic and Cube experiments, the copula transform does not affect those datasets. We have included the average AUROC during acquisition below, we have also included the results for DIME and Opportunistic RL for convenience:
>
> |Model | Bank Marketing | MiniBooNE |
> | --- | --- | --- |
> | IBFA (full) | $0.919 \pm 0.001$ | $0.957 \pm 0.000$ |
> | WO Copula | $0.917 \pm 0.001$ | $0.952 \pm 0.001$ |
> | | | |
> | DIME | $0.905 \pm 0.002$ | $0.951 \pm 0.001$ |
> | Opportunistic RL | $0.909 \pm 0.000$ | $0.953 \pm 0.000$ |
>
> We see the change is small, but the copula does lead to improved performance for IBFA. For bank marketing, IBFA without the copula would still have been the best model, and for MiniBooNE it is still able to outperform DIME. Thus, the copula transform does improve results, but is not the main source of IBFA performance. We use this transform as a method for IB models introduced by Wieczorek et al. 2018 because the method encourages sparse, disentangled latent representations, it also enforces an invariance for IB (the method is invariant to monotone transformations) which is not in place by default. We do not include this transform in the baselines because: 1) we stayed as close as possible to the original methods’ implementations, tuning the hyperparameters that they tune, and not making changes outside of their original implementations; and 2) this is a transform designed for methods that use IB, to enforce the invariance and encourage sparse disentangled latent representations. There is theoretical justification for including this in IBFA but not the other models.
>
> - Sensitivity to Gradient Measure: Below we give the number of acquisitions to acquire the correct Synthetic features when we don’t normalize the $r$ scores. We also provide the results from DIME and Opportunistic RL for convenience:
>
> |Model | Synthetic 1 | Synthetic 2 | Synthetic 3 |
> | --- | --- | --- | --- |
> |IBFA (full) | $4.017 \pm 0.003$ | $4.098 \pm 0.007$ | $5.081 \pm 0.021$ |
> |No Norm | $4.036 \pm 0.008$ | $4.103 \pm 0.008$ | $5.100 \pm 0.014$ |
> | | | | |
> |DIME | $4.079 \pm 0.057$ | $4.581 \pm 0.194$ | $5.667 \pm 0.034$ |
> |Opportunistic RL | $4.203 \pm 0.034$ | $4.846 \pm 0.020$ | $5.856 \pm 0.063$ |
>
> The performance is slightly worse than when we normalize. However, IBFA is still the best model on these datasets. We can also consider the performance on the real world datasets. Below we provide the mean accuracy/AUROC during acquisition:
>
> |Model | Cube | California Housing | Fashion MNIST | MNIST | TCGA |
> | --- | --- | --- | --- | --- | --- |
> | IBFA (full) | $0.904 \pm 0.001$ | $0.675 \pm 0.004$ | $0.717 \pm 0.001$ | $0.761 \pm 0.001$ | $0.845 \pm 0.002$ |
> | No Norm | $0.904 \pm 0.000$ | $0.675 \pm 0.005$ | $0.719 \pm 0.001$ | $0.761 \pm 0.001$ | $0.845 \pm 0.002$ |
>
> The results here are approximately unchanged, only for Fashion MNIST the performance changes within error. Therefore, the performance is not particularly sensitive to this normalization, but we do see on the synthetic datasets, where we know the optimal behavior, it leads to a small improvement. This supports our claim that the exact form of the subscoring $r$ is not as important as the overall architecture and scoring features by taking expectations over both the latent space and the current predictions.

---

> ### Author Response · Authors · 2024-11-22
> **Official Response to Reviewer PiTL Part 10**
>
> - Stopping Criteria and Feature Costs: It is correct that our model currently does not learn stopping criteria, and the objective does not consider feature costs. It is important to note our contribution in this work is a novel overall approach to AFA, which steps away from both RL and CMI maximization. We have demonstrated that this approach (and this is the first iteration of an approach like this) can perform better than these other two approaches. Ultimately it is not possible to solve every problem in a single paper, but we have set the foundation for further research into methods that use our approach. Which includes stopping criteria and feature costs. An example of stopping criteria could be to stop acquisition when a certain cutoff prediction uncertainty is reached. Or alternatively a user can decide when to stop, we believe this is quite an important option, we do not make the claim that AFA needs to be fully automated, or that IBFA has to work entirely in isolation, and often a human in the loop is very important when it comes to high stakes applications. One approach to feature costs could be to use the Opportunistic RL approach, there the reward is given by $|| \Delta p ||\_1/ \text{cost}$, where $\Delta p$ is the change in probability distribution. Here we can adjust our score by dividing each feature score by the feature cost. Note: Our scoring has no units, since we take a gradient divided by a sum of gradients, and then take expected values of this. Other approaches could take the score minus the cost, or can incorporate higher order information, such as the cost already spent acquiring the features. Or as suggested, an expected increase in the confidence, perhaps by sampling reduced regions of the latent space for each feature and seeing how the predictions change. There are various ways to include feature costs and stopping criteria which may be domain dependent, for now this moves away from the main message of this paper, but we agree these are valuable areas for future research.
>
> - Factorized Latent Space: When discussing our Indicator example, in the first bullet point, there was a brief statement about the unobserved latent encodings not conditioning on the observed features, which we address now. The factorized latent space means that each latent component is only encoded by one feature. This allows us (by choice) to design an acquisition objective so that we can score latent components, and then this allows us to trivially score features. Note: the acquisition objective still accounts for the observed features, since our objective uses the whole latent vector in $r(c, \mathbf{z}, i)$ so IBFA still makes acquisitions based on current observations. Ultimately, we had to make a tradeoff between the trivial linking of latent components to features and using a more general encoding architecture. Empirically IBFA performed better than the baselines, justifying this decision.
>
> # References
>
> 1. Alexander A. Alemi, Ian Fischer, Joshua V. Dillon, and Kevin Murphy. Deep Variational Information Bottleneck. In International Conference on Learning Representations, 2017.
> 2. Chao Ma, Sebastian Tschiatschek, Konstantina Palla, José Miguel Hernández-Lobato, Sebastian Nowozin, and Cheng Zhang. EDDI: Efficient Dynamic Discovery of High-Value Information with Partial VAE. International Conference on Machine Learning, 2019.
> 3. Tishby, N., Pereira, F.C. and Bialek, W., 2000. The information bottleneck method. arXiv preprint physics/0004057.
> 4. Li, Y. and Oliva, J., 2021, July. Active feature acquisition with generative surrogate models. In International conference on machine learning (pp. 6450-6459). PMLR.
> 5. Li, Y., Shan, S., Liu, Q. and Oliva, J.B., 2021. Towards robust active feature acquisition. arXiv preprint arXiv:2107.04163.

---

> ### Comment · Reviewer_PiTL · 2024-11-25
> **Response (Part 1)**
>
> Thanks to the authors for their detailed response, I appreciate the effort put into these clarifications and believe they'll improve the work. I still see some important issues, as I'll describe below, and in the interest of efficient communication I'll keep my responses brief.
>
> **Clarifying notation in the proofs.** I don't think the issues in your proofs are a matter of unclear notation. In your revised submission, you now show parametric versions of distributions that you *don't actually estimate*, like $p_{\theta, \phi}(x_i, y \mid x_O)$. Furthermore, I've never heard of parameterized mutual information terms like $I_{\theta, \phi}(X_i; Y \mid x_O)$: when you show this in Appendix E it's in the context of the greedy CMI technique, which seeks to maximize the true CMI $I(X_i; Y \mid x_O)$, not some parameterized version $I_{\theta, \phi}(X_i; Y \mid x_O)$. I also cannot understand the purpose of substituting every distribution in your analysis like $p(y \mid x_O)$ for its parametric version $p_{\theta, \phi}(y \mid x_O)$, besides quickly resolving the issues mentioned in my review. These seem like hasty revisions that need to be thought through more carefully. You may be able to address the problems by revisiting your proofs assuming that $p(y \mid x_O) = \mathbb{E}\_{p_\theta(z \mid x_O)} [ p_\phi(y \mid z) ]$, then considering the disconnect introduced by these not being equal, then revisiting the point about your loss function basically learning to predict $y$ given $z$, and showing how this minimizes some KL divergence between the aforementioned distributions. For now, I believe the current results are unsound and I'm going to recommend rejection.
>
> **Importance of information bottleneck.** It sounds like you agree there are settings where IB doesn't help, and that there are other more central aspects of your method's design like the choice of $r$, the choice to sample in a latent space, and the use of a gradient-based sensitivity measure. It's up to you what to do for the paper/method name, but leaning into one of these ideas seems to make more sense than the IB component, which looks like more of an add-on.
>
> **Acquisition objective and architecture.** There are aspects of your method that make sense, like the idea of focusing on classes in proportion to their predicted probability. The parts that are less intuitive are 1) how to reason about unconditionally sampling latent dimensions $z_i$ vs conditionally sampling unobserved features $x_i \mid x_O$ (which is how you would ideally make optimal selections given access to the data distribution), 2) what value there is in sampling latents $z_i$ for observed features, 3) whether the gradient-based sensitivity is what we actually want or just a heuristic that works empirically. For example, based on your description, I find it difficult to reason analytically about how your method would handle the situation outlined in Section 4. I won't belabor the point because it's not the biggest issue in the paper, but this all remains somewhat unclear to me.
>
> **Problem formulation.** Thanks for your clarifications, this mostly makes sense. What still seems odd about your problem formulation is it doesn't show the correct object we should be maximizing over: it's a *policy* whose rollouts should result in a low loss in expectation over a data distribution, perhaps subject to constraints or penalties on the resulting subsets. Again, not the biggest issue in the paper.
>
> **CMI shortcomings: indicator example.** I think including this example is fine, but the proposal you give to resolve it doesn't make sense and is barely related to IBFA. I would recommend removing it, or ideally replacing it with something more closely related to IBFA.
>
> **CMI shortcomings: entropy minimization.** Great. The uncertainty criterion you're talking about, $\max_c [p(y = c \mid x_O)]$ can be justified from the perspective of expected 0-1 loss, whereas entropy is related to expected log-loss. It seems straightforward to design a procedure to do maximize this at each selection step, and I would be curious to see how it performs, but agree it's beyond the scope of this work.
>
> **Loss function.** I would argue the reverse of what you're saying: minimizing the predictive error is clearly the right thing to do (you're trying to predict $y$, after all), a regularizer on the posterior to ensure non-zero variance might be helpful (otherwise there might be no ability to sample latents), and the harder bit is to explain is the connection with the standard VIB formulation. Up to you what you'd like to do here, your clarifications in the paper are probably an improvement, but this argument about starting from VIB and moving the integral inside the log comes off as a bit odd.

---

> > ### Comment · Reviewer_PiTL · 2024-11-25
> > **Response (Part 2)**
> >
> > **STG and MNIST.** It sounds like you agree that reducing the candidate features limits performance. I understand that certain baselines cannot handle high feature counts, but I'm not quite clear on your answer regarding IBFA's ability to do so. It would help to be more clear about this limitation in the paper, if it is a limitation.
> >
> > **Wall-clock times.** Thanks, this would be good to add to the appendix.
> >
> > **MiniBooNE.** Sounds reasonable, this would be good to add to the appendix.
> >
> > **Copula.** Thanks for the extra comparisons. It would be good to add this description of what you're doing and these new comparisons to the appendix.
> >
> > **Gradient norm ablations.** Thanks for the extra comparisons, these would be good to add to the appendix.
> >
> > **Stopping criteria.** Yes, it makes sense that you can't solve every problem in one paper. It would make sense to describe this limitation in the paper, and perhaps your ideas for resolving it in future work.

---

> > > ### Author Response · Authors · 2024-11-26
> > > **Reply 1 - The Case for our Theory**
> > >
> > > Thank you for the fast and detailed reply, we appreciate the continued effort to provide feedback on our work. Since following the rebuttal the main concern is now the theorems, we discuss that in detail first, and respond to your other points in an additional response. We respectfully disagree that the theory in our manuscript is “unsound” and hope the below clarifies any remaining concerns that you have, we are very happy to discuss.
> > >
> > > In our initial submission we did not use subscripts because our theory applies to any model that uses the Markov Chain $X-Z-Y$. In our framework, our focus is not on quantifying the MI, but leveraging it as a guiding  principle for our optimization framework. Rather than explicitly calculating the mutual information we employ an indirect approach to estimate it within the model. However, as pointed out, the original proofs did not make this fully clear, we chose to add subscripts to help this. The proofs themselves work whether we use a parameterized model or unparameterized, the important part is that we estimate the terms using a model that has the assumption $p(y, z | x) = p(y|z)p(z|x)$.
> > >
> > > The initial concern was whether using our modelling assumption was valid, which we justified by stating we are considering properties using the distributions of the model. Subsequently, the concern is if this is valid. Whilst CMI maximization methods ideally seek to maximize the true CMI, this is only possible in very specific circumstances where the underlying distribution is known. Instead, the widely accepted approach is to estimate the CMI either by estimating the distributions with generative models, or estimating the CMI with other approaches. In deep learning this is done by parameterized models, and therefore these methods maximize a **parameterized estimate of CMI**, so this is a common approach to estimate $I\_{\theta}(X; Y)$ with a model (this is also common notation). This is not new or specific to our work. Below we provide a non-exhaustive list of papers that do this, not all are AFA works:
> > >
> > > - Mutual Information Neural Estimation (https://arxiv.org/pdf/1801.04062) in Section 3 uses a parameterized network $T\_{\theta}$ which produces a parameterized estimate of mutual information $I_{\theta}(X; Z)$
> > > - DIME (https://arxiv.org/pdf/2306.03301), one of the baselines we compare to, in Section 4.1 uses a value network $v(\mathbf{x}\_{S}; \phi)$, “designed to estimate the CMI for each feature, $v\_{i}(\mathbf{x}\_{S}; \phi) \approx I(Y ; X\_{i} | \mathbf{x}\_{S})$”
> > > - Learning Robust Representations via Multi-View Information Bottleneck (https://arxiv.org/pdf/2002.07017), in Section 3.2 presents the loss $\mathcal{L}\_{1}(\theta ; \lambda\_{1}) = I\_{\theta}(\mathbf{z}\_{1} ; \mathbf{v}\_{1} | \mathbf{v}\_{2}) - \lambda\_{1} I\_{\theta}(\mathbf{v}\_{2}; \mathbf{z}\_{1})$, “where $\theta$ denotes dependency on the parameters of the encoder $p\_{\theta}(\mathbf{z}\_{1} | \mathbf{v}\_{1})$”, this leads to their loss which uses the term $I\_{\theta \psi}(\mathbf{z}\_{1}; \mathbf{z}\_{2})$
> > > - Deep Variational Information Bottleneck (https://arxiv.org/pdf/1612.00410), in the introduction, “We regard the internal representation of some intermediate layer as a stochastic encoding $Z$, of the input source $X$, defined by a parametric encoder $p(\mathbf{z} | \mathbf{x}; \theta)$.” with this notation the loss is written as $I(Z, Y ; \theta) - \beta I(Z, X; \theta)$
> > > - EDDI (https://arxiv.org/pdf/1809.11142), in Section 3.3, the acquisition objective is initially framed as the true mutual information (equation 6), the remainder of the section is about how to approximate the quantity using their architecture, the final part being equation 9, where “we use the partial VAE approximation $p(\mathbf{z}| \mathbf{x}\_{\phi}, \mathbf{x}\_{i}, \mathbf{x}\_{o}) \approx q(\mathbf{z}| \mathbf{x}\_{\phi}, \mathbf{x}\_{i}, \mathbf{x}\_{o})$”.
> > >
> > > Thus, estimating the mutual information using a parametric model (e.g. conditional densities using a neural network) is common practice (including using the estimates in the training loss and in the acquisition objective). Using modelling assumptions when required to simplify expressions is also accepted practice, see Appendix A.1 of EDDI for example, their approximation is used at the beginning of their proof. We have followed this standard practice to show that for a given latent variable model with the assumption $p(y, z | x) = p(y|z)p(z|x)$, Theorem 1 applies. The purpose of the theorem is to show that minimizing the term $I(Y ; Z | X)$ during training will improve the acquisition performance at inference time, exact calculation is not required. Then Theorem 2 shows that this term is built into our loss when framed using IB. This is to justify our loss function from a principled information theory approach.

---

> > > > ### Author Response · Authors · 2024-11-26
> > > > **Reply 2 - Remaining Points**
> > > >
> > > > **Importance of information bottleneck.** To clarify, in our ablations using IB outperforms not using IB 5/6 times, and in that one case (Fashion MNIST) the performance is within error. We hypothesize $\beta$ is too high at 0.001, which is the largest value we try. We are testing again using a lower value, and will report the results. Our reasoning for using IB is still valid, since we score features by scoring their latent components we want the latent space to remove feature level noise.
> > > >
> > > > **Acquisition objective and architecture.**
> > > >
> > > > 1) We agree an ideal scenario is to have access to the conditional distributions $p(x\_i | \mathbf{x}\_{O})$, however, this requires training a generative model, and we saw the generative models in our experiments performed worse than the other methods. Regarding why latent samples are effective, if we have a distribution of a feature, where samples are always encoded, then we also have a distribution for the associated latent components. We have opted to focus on this latent space distribution, rather than the distribution of the feature, because we can parameterize this as Gaussian and the predictor is trained to work effectively with samples from this distribution. Whereas, if we trained a generative model in the feature space the distributions can be more complex, and therefore it may be more difficult to accurately estimate the underlying distribution (again we point to the empirical results of the generative models). Additionally, since we have applied IB, our latent distribution does not contain the feature level noise, which might affect $p(x\_i | \mathbf{x}\_{O})$ more than the latent distribution. The fact that we have factorized the latent space so each latent component is encoded by one feature is a modelling choice that we discuss in our limitations and in Part 10 of our initial rebuttal.
> > > >
> > > > 2) Just as in IB methods, and other methods that use stochastic encodings, a benefit to sampling the values of observed features is that we can consider the effects of noise in the feature measurement, by modeling that uncertainty in the latent encoding.
> > > >
> > > > 3) On further reflection we agree that using gradients for $r$ is a heuristic that works empirically, demonstrated by our results. We saw when we removed normalization the change was minimal, so there are various forms the function $r$ could take. However, it should still be constructed so that for a given latent sample it can score latent components (or groups) based on how important they are for predicting each class. We shall make this clear about our design choice in Section 5.2.
> > > >
> > > > **Problem Formulation.** We have given the test time objective of an AFA model, which is, for each point separately to iteratively acquire features, making the feature subset $S$ larger, with the long-term aim of optimizing the objective given. Can you please clarify how this is different to what has been suggested: “a policy whose rollouts should result in a low loss in expectation over a data distribution”.
> > > >
> > > > **Indicator Example.** We agree the precise details of the solution in proposition 2 are not used in IBFA, but we still believe it is a net positive for Section 4. Without Proposition 2, we have introduced a shortcoming of CMI without providing a solution. Proposition 2 follows from Proposition 1 to add justification to our claim that unobserved features are an important but overlooked consideration in AFA. This idea is used as motivation for our method since it shows what properties an optimal objective should have, but since we have non-linear networks it is non-trivial to reason about our exact objective in this example.
> > > >
> > > > **Loss Function.** We are in agreement that optimizing the predictive loss with a KL term is initially the natural approach. However, within the context of the surrounding literature, this is not the standard approach, which is to use the VIB loss for methods that use Information Bottleneck. Therefore, we believe it is necessary to describe the accepted practice and explain why we need to diverge from it.
> > > >
> > > > **Remaining points.** We are glad that we are in agreement regarding the remaining points. We shall add the relevant discussions and results to the appendix as suggested.

---

> ### Comment · Reviewer_PiTL · 2024-11-26
> **Theory issues**
>
> I've taken some time to go through your results to explain exactly what's wrong with them. Let's start with Theorem 1 as this is enough to illustrate the problem. As setup, let's see if we can agree on this:
>
> - There exists a data distribution $p(x, y)$, and we can do arbitrary marginalization and conditioning to get $p(x_i \mid x_o)$ and $p(y \mid x_o)$. At your suggestion *we will not use these terms*, especially $p(y \mid x_o)$, and instead opt for versions given by your models.
> - Your model specifies two distributions $p_\theta(z \mid x_o)$ and $p_\phi(y \mid z)$. An issue that occurred to me upon first reading your paper, which I forgot to mention in the review, is that the notation is imprecise: these distributions should have separate subscripts/superscripts specifying the variables they're conditioned on. For example, the latent you sample when conditioning on $x_o$ is conditionally independent of any other $x_i$'s, and yet if your network conditioned on both $(x_o, x_i)$ you would have another latent variable that clearly *does* depend on $x_i$; these cannot both be denoted as $z$, it's ambiguous because they're two different random variables. To be more precise and ensure we handle conditional independence correctly and consistently, you should call these something like $z_o$ and $y_o$ for each $x_o$. This also helps distinguish your network's output from the real $y$, because you want your theory to be based entirely on the model's approximations (there should be no reference to $y$ whatsoever). Now, perhaps we can agree that what your networks actually specify is two distributions $p_\theta(z_o \mid x_o)$ and $p_\phi(y_o \mid z_o)$; because these variables only exist in the context of your model, there's no risk of ambiguity so you could even drop the $\theta$/$\phi$ subscripts. Returning quickly to the $x_o$/$x_i$ conditional independence bit mentioned previously, we now have $p(z_o \mid x_o, x_i) = p(z_o \mid x_o)$, but no similar conditional independence for $p(z\_{o \cup i} \mid x_o, x_i)$.
>
> **As a first check, I would ask if you agree up until this point.**
>
> Now returning to Theorem 1, this starts by considering a mutual information term you've written as $I\_{\theta, \phi}(x_i, y \mid x_o)$. A couple remarks:
>
> - I can't tell which $y$ you're trying to characterize here. In the greedy CMI method, we would be trying to characterize the real $y$, but you claimed your analysis is all about the outputs from your model. One interpretation is that you're thinking of $y_o$, the random variable that gets sampled using the Markov chain $x_o \to z_o \to y_o$. **If there's another interpretation please let me know**, but I'll proceed with this for now.
> - If we interpret the mutual information as described above, we come to an interesting conclusion: that $I(x_i; y_o \mid x_o) = 0$ for all $x_i$. You can follow your proof's mechanics to see this, or you can notice that $y_o$ is simply independent of any $x_i$ conditional on $x_o$. So this result doesn't mean anything; by making it about the "$y$" implied by your model, you ensured by construction that no $x_i$ can provide any information about it.
>
> Now, you might be wondering why this wasn't apparent previously. I can track it down to a couple inconsistencies in how you applied conditional independence in your proof. I'll revert to your ambiguous notation for this part:
>
> - To get to line 1216, you used the fact that $p(y \mid z, x_i, x_o) = p(y \mid z)$. If we replace $y \to y_o$ and $z \to z_o$, this is actually true: $p(y_o \mid z_o, x_i, x_o) = p(y_o \mid z_o)$ by construction.
> - However, in line 1216 you also preserved the terms $p(z \mid x_i, x_o)$ and $p(z \mid y, x_i, x_o)$. If we were consistent with interpreting $y \to y_o$ and $z \to z_o$ then these should be simplified to $p(z_o \mid x_o)$ and $p(z_o \mid y_o, x_o)$. When combined with the remaining terms inside the log, this would lead you to conclude that the integral is equal to 0! Crucially, you did not make this simplification, presumably because you now began to view $z \to z\_{o \cup i}$ which actually does depend on $x_i$. This seems to be the main mistake that makes this result wrong.
>
> **As a second check, I'll ask if you agree up until this point.**
>
> Separately, another issue is that you integrate over $p\_{\theta, \phi}(x_i, y \mid x_o)$. Like I mentioned previously, *your models don't estimate this distribution,* it's never defined. At some point this becomes an integral over $p\_{\theta, \phi}(x_i, y, z \mid x_o) = p\_{\theta, \phi}(x_i \mid x_o) p\_{\theta, \phi}(y, z \mid x_i, x_o)$. I don't know what the difference is between $p(x_i \mid x_o)$ and $p\_{\theta, \phi}(x_i \mid x_o)$, since the latter isn't defined by your model.
>
> Overall, I think the remedy is 1) improve the ambiguous notation, 2) reconsider whether you truly want your analysis to be about distributions implied by the model vs real ones under the data distribution. Let me know what you think.

---

> > ### Author Response · Authors · 2024-11-27
> > **Different feature subsets change the conditional distribution of z, it does not become a new random variable**
> >
> > Thank you again for the fast reply, we appreciate the time to carefully look over the theory. It is important to us that we align here. Our response will start by going over your new points and how we disagree. We will then re-present the proof which should be more clear than originally written, especially following the first part of this response.
> >
> > The main place where we disagree is that having a feature subset $x\_O$ requires subscripting the latent variable and the model's prediction and in particular the implications of this. That is, that $y$ (the outcome variable) and $y\_O$ (the outcome conditioned on the observations) need to be notionally distinguished and that $z$ conditioned on different features are different random variables. We disagree with this premise and therefore the remaining points presented.
> >
> > Our encoder models a distribution of the full latent vector no matter how many features have been observed. For simplicity let’s assume we have two features each encoding one latent component. For the neural network implementation of our encoder: $p\_{\theta}(z | x\_1) = p\_{\theta\_1}(z\_1 | x\_1)p\_{\theta\_2}(z\_2 | *)$, our encoders can take a missing feature as input (there is a mask variable indicating presence or missingness of a feature), and they still produce a distribution. If we were to use unparameterized distributions this would look like $p(z | x\_1) = p(z\_1 | x\_1)p(z\_2 ) = p(z\_1 | x\_1)\int p(z\_2 | x\_2)p(x\_2)dx\_2$.
> >
> > _The random variable $z$ is the full latent vector. It does not become a different random variable when conditioned on different feature subsets, it just has different conditional distributions._
> >
> > It is true that $x\_O$ only affects the distribution of a subset of the components (in our model), but the unconditional distribution of the other components still exists. We are still able to sample a full latent vector, and push it through the predictor to make a prediction. Therefore our notation $p(z | x), p(y | z)$ does not need to be replaced by $p(z\_O | x\_O), p(y\_O | z\_O)$, we don't use subsets of the latent vectors anywhere, we always use the full latent vector. In the surrounding literature, papers that use a subset of input features in the same predictor are built under the same spirit, and use the same notation; this is not restricted to AFA. For example:
> > - GDFS (https://arxiv.org/pdf/2301.00557) one of the baselines we compare to
> > -  Discovering Features with Synergistic Interactions in Multiple Views (https://openreview.net/pdf?id=hFEgae0od4)
> > - Feature Selection using Stochastic Gates (https://arxiv.org/pdf/1810.04247)
> >
> > Just as in these papers (and all feature acquisition papers) we do not write $p(Y=y| X\_O=x\_O)$ as $p(Y\_O=y| X\_O=x\_O)$, it is not necessary, and as we see next can lead to spurious conclusions. Regarding the last sentence of the first point, stating there is no conditional independence for $p(z\_{O \cup i} | x\_O, x\_i)$, this is incorrect, the purpose of our encoder architecture is that $p(z\_{O \cup i} | x\_O, x\_i) = p(z\_O | x\_O)p(z\_i | x\_i)$. Another point is that in our proof we do not assume the factorized latent space. We use it in our architecture to take latent scores and trivially turn them into feature scores, but our proofs are for any latent variable model.
> >
> > Following this, there is a misconception that predictions from our model cannot change under new observations. The claim is that by considering the $y$ predicted by our model $I(Y ; X\_i ; x\_O) = 0$. This is incorrect, as we acquire new features the latent distribution changes, since we can now condition on this feature. Therefore the predicted distribution of $y$ changes, so mutual information is not 0. The misunderstanding is that the MI has been interpreted for $y\_O$ which is the prediction if we only have $x\_O$. Calculating mutual information with $y\_O$ would indeed be zero, as you've stated, but this is not a meaningful result. The predictions adapt to the feature subset present, so we consider the mutual information with the prediction for the features present and the feature being scored, not just the current subset.
> >
> > The final error is rewriting terms that cannot be rewritten, $p(z | x\_i, x\_O) = p(z\_O | x\_O)$ and $p(z | y, x\_i, x\_O) = p(z\_O | y\_O, x\_O)$. Recall the distributions of the full latent variable exist no matter how many features we have. $p(z | x\_1, x\_2) = p(z\_1 | x\_1)p(z\_2 | x\_2)$ cannot be rewritten as $p(z | x\_1, x\_2) = p(z\_1 | x\_1)$. This simplification cannot be made, and so the final point is also incorrect.

---

> ### Author Response · Authors · 2024-11-27
> **Re-presenting the Proof of Theorem 1**
>
> As mentioned, it is important to us that we are able to align on these proofs. So to help discussion we re-present the proof below, where we have improved the clarity. Our intention by introducing the parameter subscripts was to avoid the supposed conflation of true and approximate distributions, however, it is clear this has unfortunately doen the opposite. Therefore, as you have suggested, we reason about the true mutual information, using underlying distributions for as long as possible before introducing assumptions. Once we bring in modelling assumptions or distributions it becomes the estimated MI. We hope this removes the ambiguity around notation, we shall say explicitly where assumptions are brought in. Consider the mutual information:
>
> $I(X\_i ; Y | x\_O) = \int p(x\_i, y | x\_O) \log \frac{p(y |x\_i, x\_O)}{p(y | x\_O)} dx\_i dy$
>
> Since we only need to consider maximizing this with respect to $i$ we remove terms that do not depend on $i$ (the $p(y | x\_O)$ in the logarithm). We include a marginalization over **any** new random variable.
>
> $I(X\_i ; Y | x\_O) = \int p(x\_i, y, z | x\_O) \log p(y |x\_i, x\_O) dx\_i dydz$
>
> Bayes' theorem gives us $p(y | x\_i, x\_O) = \frac{p(y| z, x\_i, x\_O)p(z| x\_i, x\_O)}{p(z | y, x\_i, x\_O)}$
>
> Replacing this in the logarithm, flipping the fraction inside the logarithm and subsequently multiplying by -1 gives
>
> $I(X\_i ; Y | x\_O) = -\int p(x\_i, y, z | x\_O) \log \frac{p(z | y, x\_i, x\_O)}{p(z| x\_i, x\_O)p(y| z, x\_i, x\_O)} dx\_i dydz$
>
> Separating out the logarithm gives
>
> $I(X\_i ; Y | x\_O) = -\int p(x\_i, y, z | x\_O) \log \frac{p(z | y, x\_i, x\_O)}{p(z| x\_i, x\_O)} dx\_i dydz + \int p(x\_i, y, z | x\_O) \log p(y| z, x\_i, x\_O) dx\_i dy dz$
>
> The first term by definition is $-\mathbb{E}\_{p(x\_i | x\_O)}I(Z; Y | x\_i, x\_O)$. In the second term we can multiply the tem in the logarithm by $\frac{q(y |z)}{q(y|z)}$. This becomes:
>
> $\int p(x\_i, y ,z | x\_O)\log \frac{p(y| z, x\_i, x\_O)}{q(y |z)}dx\_i dy dz - \int p(x\_i, y ,z | x\_O)\log q(y |z)dx\_i dy dz$
>
> The second part of this is not affected by the optimization over $i$ so can be disregarded. This gives:
>
> $I(X\_i ; Y | x\_O) = -\mathbb{E}\_{p(x\_i | x\_O)}I(Z; Y | x\_i, x\_O) + \mathbb{E}\_{p(z, x\_i | x\_O)}D\_{KL}( p(y| z, x\_i, x\_O)|| q(y |z))$
>
> With additional terms not affected by the regularization. Note: we have not specified the additional variable $z$ or $q(y |z)$ yet, everything has been achieved by simple manipulation of the expressions. What we see is that minimizing $\mathbb{E}\_{p(x\_i | x\_O)}I(Z; Y | x\_i, x\_O)$ is maximizing a lower bound of the mutual information. We introduce our modelling choices by letting $z$ be the latent variable of our model and $q(y|z)$ be our predictor network. Our modelling assumption is used to reason that the second term is zero, using the approximating $p(y| z, x\_i, x\_O) \approx q(y |z)$. Assuming agreement with the previous lines, the only place where there can be an error is in this assumption step. And as the provided citations have shown, it is widely accepted to include the modelling assumptions. The modelling assumptions we have used are common in deep information bottleneck models (https://arxiv.org/pdf/1912.13480 provides theoretical reasoning why this modelling assumption is an effective one).
>
> How does this affect Theorem 1? Theorem 1 can be restated as, minimizing $\mathbb{E}\_{p(x\_i | x\_O)}I(Z; Y | x\_i, x\_O)$ maximizes a lower bound on $I(X\_i ; Y | x\_O)$, and under our modelling assumption $p(y, z |x) = p(y|z)p(z|x)$ it maximizes the MI. This can be viewed as an estimate using the model, rather than the exact quantity, recall we don't have access to the underlying distributions, and this involves distributions that are parts of our learnt model as well as our assumption. The result of the theorem still stands, minimizing $I(Z; Y | X_S)$ during training will help regularize the loss to improve the acquisitions, even if it is not used as the acquisition objective. Without explicitly calculating these MI terms, we are still able to reason about them, and how they can guide our loss function. Note: we made no assumption on the modelling architecture, we only include our modelling assumption in the last step to make the lower bound tight.

---

> > ### Author Response · Authors · 2024-11-27
> > **New Empirical Results**
> >
> > In addition to the discussion on theory, we wanted to briefly add that we have run two more experiments in support of our method.
> >
> > **Fashion MNIST $\beta=0.0001$.** We reran Fashion MNIST with a lower $\beta$, $\beta=0.0001$ rather than $0.001$. As shown in the surrounding literature, this is within the normal range for $\beta$. The new average accuracy during acquisition is $0.720 \pm 0.001$, this is larger than when $\beta=0.001$ ($0.717 \pm 0.001$) and when $\beta=0.0$ ($0.718 \pm 0.001$). As hypothesized, after a slight adjustment to $\beta$ to a lower but crucially non-zero value the performance improves, and is now better than when $\beta=0.0$ (no IB). Again this further supports the use of IB in our model, both from an empirical and fundamental perspective. Now we have shown on 6/6 ablations that removing IB makes IBFA worse.
> >
> > **Indicator Example.** We implemented and tested the indicator example, using 10 "ordinary" features and 1 indicator. The train set is 60,000 the validation and test are both 10,000. IBFA recovered the optimal behavior. Across 5 runs, on all test points the indicator was chosen first followed by the corresponding feature. All necessary features are acquired in 2 acquisitions every time. Recall the expected number of acquisitions for CMI maximization is $3 - 1/d = 2.9$. So whilst it is difficult to reason exactly about our model and objective on the indicator example (due to the non-linear networks) we have empirically shown it can solve this problem optimally, directly relating the example to IBFA.
> >
> > Following these new results, and our discussion on theory, we kindly ask that you reconsider your recommendation and the lowering of your score. Especially given that the main reason was concerns about theory (which we hope we have alleviated), and that the theory is not central to our paper or required by our method. The small section on theory is an alternative justification for the loss function, not a fundamental foundation on which the entire method is built, our intial reasoning from the perspective of latent space variance is unaffected. The model design, loss function, acquisition objective and the positive results are also unaffected.

---

> > > ### Author Response · Authors · 2024-12-02
> > > **Is there anything outstanding to discuss?**
> > >
> > > Dear Reviewer PiTL,
> > >
> > > Thank you for your time reviewing and the feedback so far. Since the end of the discussion period is close, we wanted to check in to see if we have addressed all of your concerns, or if there is anything further we can clarify in the remaining time.
> > >
> > > We have discussed the theory in depth, and we hope our _most recent replies_ have alleviated this concern. As we have said it is important to us that we are able to align, so if there are any remaining concerns we will be happy to discuss these. We also kindly direct you to our new empirical results, showing that IBFA obtains optimal performance on the Indicator task, directly linking our model to the Section 4 analysis, and the new results on the ablation demonstrating using IB outperforms not using it in 6/6 ablations.

---

> ### Comment · Reviewer_PiTL · 2024-12-02
> **Theory issues (cont.)**
>
> Apologies for the delayed reply. Thanks for your last message, let me respond to your latest proposed changes. A couple preliminary points before I get into the details of your new derivation:
>
> - It appears my suggestion to distinguish between latent variables depending on the conditioning subset was misunderstood. The notation $z_o$ was intended to indicate which subsets of $x$ are conditioned on, not a subset of latent dimensions. So a few of my points were lost in translation, for example the conditional independence bit $p(z_o \mid x_o, x_i) = p(z_o \mid x_o)$. To avoid ambiguity, perhaps better notation would have been $z^o$ so you can still subscript into each dimension, e.g., $z^o_i$. I'll return to this later in my reply.
> - One of your main changes was to switch back to analyzing the mutual information with the true random variable $y$ rather than the output distribution induced by your model. I believe that’s the right decision, because the alternative leads to vacuous results like variables $x_i$ contributing zero CMI.
>
> I’ve gone through your new proof, and your algebraic transformations are correct. However, there are two flaws in your derivation:
>
> 1. Your assumption $q(y | z) = p(y | z, x_i, x_o)$ is not justified. You can’t simply state that your network matches this distribution, you must show that some component of your learning algorithm encourages them to be close. Please show where this appears in your objective, perhaps in the form of a KL divergence between the two distributions. Because we've discussed this previously, I believe you can show that your objective minimizes an upper bound on $D\_{KL}(p(y \mid x_o) \mid \mathbb{E}\_{p_\theta(z \mid x_o)}[p_\phi(y \mid z)])$, but you're assuming something stronger than that.
>
> 2. More fundamentally, and as mentioned in my last response, you don't have a single random variable $z$ that's valid to analyze in this fashion. Your algebraic transformations are correct for any random variable, but $z$ is not a random variable that exists like $x$ and $y$. It's induced by your network separately for each conditioning subset; this is a subtle point, but that means there are different random variables $z^o$ for each subset $x_o$. Consider this: if there were a single random variable $z$, then we should be able to get its distribution $p(z \mid x_o)$ directly from your network $p_\theta(z \mid x_o)$ and identically by marginalizing over another variable, say $\int p(x_i \mid x_o) p_\theta(z \mid x_o, x_i) dx_i$. You cannot guarantee that these distributions are identical, therefore there isn't a random variable $z$. Fixating on a single dimension in your factorized setup, you can see that $p_\theta(z_i \mid *)$ is not guaranteed to equal $\int p_\theta(z_i \mid x_i) p(x_i \mid x_o)$. Surely you can agree that this point is true and acknowledge that it's odd; it's also inconsistent with the notion of a random variable $z$. Anyway, as a result, besides your modeling assumption being unjustified, the entire derivation is invalid because you haven't correctly specified the random variable you're analyzing.
>
> I've tried to think about how to fix your derivation. Since your algorithm generates the latent variable conditioned on $x_o$, it could make sense to replace all instances of $z$ with $z^o$ in your derivation. This will unfortunately lead you to trying to show $p(y \mid z^o, x_o, x_i) = p_\phi(y \mid z^o)$, which of course is not true, both because 1) single sampled latent values do not contain of all $x_o$'s information, and 2) $x_i$ contains new information beyond the latent variable and $x_o$. For that reason, I cannot see how to fix your derivation.
>
> To summarize, I believe your theoretical results are still incorrect. The AC can be a judge of that, and my concerns with the different versions of your theory are all laid out here. I tend to agree with you that the theory is not a central part of your paper, and the empirics are promising on their own, but I can't recommend acceptance for a paper with incorrect theorems.

---

> > ### Author Response · Authors · 2024-12-04
> > **Fixing Theory with Specific Notation for Z**
> >
> > Thank you for your continued engagement in the review process and for taking the time to explain the issue. We sincerely appreciate your efforts to help improve our manuscript. We are glad you agree that the theoretical results are not central to the work. That said, we appreciate your concern and address it below.
> >
> > After reading your explanation, we understand and agree with your point on latent variables, and the assumptions being made about independence within the model in our derivation. To that end, we have updated the theorems and their associated proofs, using this more precise notation, and without the assumptions. We present the updated theorems and proofs below.
> >
> > # Theorem 1 Using Precise Notation for Latent Variables
> >
> > From our discussion, we are in agreement that for some random variable $T$, maximizing CMI can be written as:
> >
> > $\max\_{i} I(X\_i ; Y | x\_O) \equiv \max\_{i} -\mathbb{E}\_{p(x\_i | x\_O)}I(T; Y | x\_i, x\_O) + \int p(x\_i, y, t | x\_O) \log p(y| t, x\_i, x\_O) dx\_i dy dt$
> >
> > The second term can be written as $-\mathbb{E}\_{p(x\_i | x\_O)}\mathbb{E}\_{p(t | x\_i , x\_O)}H(Y | t, x\_i, x\_O)$, so this becomes
> >
> > $\max\_{i} I(X\_i ; Y | x\_O) \equiv \max\_{i} -\mathbb{E}\_{p(x\_i | x\_O)} \big[I(T; Y | x\_i, x\_O) + \mathbb{E}\_{p(t | x\_i , x\_O)}H(Y |t, x\_i, x\_O)\big]$
> >
> > We can replace the maximization with the minimization of the negative quantity. As discussed, this is true for any random variable, for example, if $T$ is always independent of $Y$ no matter the conditioning, then the first term is zero, and the second term becomes minimizing the entropy. Since we provide a latent variable model, it is useful to think about this as the latent variable, importantly specifying which feature subset induces the latent variable as you have demonstrated. Using the superscript notation, we could consider $Z^{O}$ the random variable induced by measuring feature subset $O$. However, it is more helpful to consider the latent variable $Z^{i, O}$, which is induced by measuring feature subset $O$ and a possible value of $x\_i$. Therefore, **the theorem is updated to**:
> >
> > $\max\_{i} I(X\_i ; Y | x\_O) \equiv \min\_{i} \mathbb{E}\_{p(x\_i | x\_O)} \big[ I(Z^{i,O}; Y | x\_i, x\_O) + \mathbb{E}\_{p(z^{i, O} | x\_i , x\_O)}H(Y | z^{i, O}, x\_i, x\_O) \big]$
> >
> > Where the change is to use the superscript notation on $Z$ to make it clear we are referring to the latent variable induced by a given set of features, and not making any independence assumptions, which includes this second term. Since the Markov Chain of our architecture is $X \rightarrow Z  \rightarrow Y$, we can consider how the above would be achieved if $Y$ is independent of $(x\_i, x\_O)$ conditioned on $z^{i, O}$, but note **we are not making the assumption, just exploring the implication if it holds**. The implication is that maximizing CMI would be achieved by choosing a feature where for all samples of $z^{i, O} | x\_i, x\_O$ (those with high likelihood), the prediction for $p(Y | z^{i, O})$ is constant (or changes minimally), this corresponds to the first term and that prediction has low entropy, this corresponds to the second term. As we say, we will not use the assumption, but it is informative to consider the case.
> >
> > Next we discover how this is in our loss function in an updated theorem 2, and how this will change the manuscript.

---

> > > ### Author Response · Authors · 2024-12-04
> > > **Relation to Theorem 2 and Manuscript Changes**
> > >
> > > Now we can relate this to our loss function, updating Theorem 2. We consider an integral over $p(x\_S, y)$, where $x\_S$ is representing both the feature values, and the feature availability, so this distribution considers the distribution of feature values and $S$.
> > >
> > > The first term of the loss is:
> > >
> > > $-\int p(x\_S, y) \log (\mathbb{E}\_{p\_\theta(z^{S} |x\_{S})} p\_\phi(y | z^{S} ) )dx\_{S}dy$
> > >
> > > Which, if we multiply the part in the logarithm by $p(y | x\_S)/p(y | x\_S)$ gives
> > >
> > > $-\int p(x\_S, y) \log p(y | x\_S)dx\_S dy +\mathbb{E}\_{p(x\_S)}D\_{KL}(p(y | x\_S) || \mathbb{E}\_{p\_\theta(z^{S} |x\_{S})} p\_\phi(y | z^{S} ))$
> > >
> > > Focusing on the first term, this can be written by  marginalizing over the latent variable induced by $S$, and multiplying the term in the logarithm by $p(y | z^{S} , x\_S)/p(y | z^{S} , x\_S)$
> > >
> > > $= -\int p(x\_S, y, z^{S}) \log \big(\frac{p(y | x\_S)}{p(y| z^{S} , x\_S)}p(y | z^{S} , x\_S)\big)dx\_S dy dz^{S}$
> > >
> > > Separating out the logarithm, flipping the fraction and multiplying that term by -1 gives
> > >
> > > $\int p(x\_S, y, z^{S}) \log \frac{p(y| z^{S} , x\_S)}{p(y | x\_S)}dx\_Sdydz^{S} - \int p(x\_S, y, z^{S})\log p(y| z^{S}, x\_S)dx\_Sdydz^{S}$
> > >
> > > $= \mathbb{E}\_{p(x\_S)}I(Z^{S}; Y | x\_S) + \mathbb{E}\_{p(x\_S)} \mathbb{E}\_{p(z^{S} | x\_S)}H(Y | z^{S}, x\_S)$
> > >
> > > $=\mathbb{E}\_{p(x\_S)} \big[I(Z^{S}; Y | x\_S) + \mathbb{E}\_{p(z^{S} | x\_S)}H(Y | z^{S}, x\_{S})\big]$
> > >
> > > We of course do not have access to $p(x\_S, y)$ so we sample mini-batches and the masking variable for the input feature subsets from the training data. The update to Theorem 2 is that our loss can be written in this way, including a KL term between the true conditional distribution of $y | x\_S$ and the predicted distribution via the latent variable.
> > >
> > > **How Theorems 1 and 2 work together.** This is unchanged. In Theorem 1, we are able to derive CMI maximization as a different minimization considering the effect features have on a latent variable, which is helpful since our model uses one. This is still _not our acquisition criterion_, but it is used to say that such a term can be viewed as added regularization to shape the latent space for acquisition, with the specific term being $\mathbb{E}\_{p(x\_S)} \big[I(Z^{S}; Y | x\_S) + \mathbb{E}\_{p(z^{S} | x\_S)}H(Y | z^{S}, x\_{S}) \big]$. If this term is zero, for any $x\_S$, the samples $z^{S}$ produce the same predicted distribution over $Y$ and this is low entropy. That is, the latent space is separated into regions, associated with each class. Intuitively it makes sense that a latent space like this is better for acquisition than a very complex one. Theorem 2 says that this regularization term is in our loss function, so we have the added regularization. We see from our results that this loss function change from standard VIB improves the acquisition performance.
> > >
> > > **How the manuscript changes.** We agree that these changes are required. However, we believe the changes to the manuscript are relatively minor in comparison to the entire paper.. The required changes are: updating Appendix E and F, adding the superscript notation for $Z$ in Section 5 (the model description) and briefly in Appendix C, and updating Section 5.3 to clarify the new statements.
> > >
> > > Crucially this does not change the empirical results, the model itself, the theory in Section 4 describing the shortcomings of CMI, or any ablations. The initial reasoning for moving away from the VIB loss (needing diversity among $z$ samples in the low feature regime) is still valid so this is also unchanged, it is the secondary reasoning and the associated explanations in the Appendix that have changed. We view these changes as necessary, but small in the context of the whole paper; the narrative, the method and the results are unchanged.
> > >
> > > Once again, we would like to thank you for the time and effort reviewing the work, especially in bringing this precise point to our attention. The resulting changes implemented have improved the paper significantly. We hope this change addresses your final point and that you will reconsider your recommendation.

---

### Official Review · Reviewer_B3o7 · 2024-11-03

**Soundness:** 3
**Presentation:** 3
**Contribution:** 3
**Rating:** 6
**Confidence:** 4

**Summary:**

The paper proposes to use IB for scoring the utility of features, so to enable dynamic AFA. The idea is interesting and seems legit to me. Unlike the widely used CMI (which leans towards myopic feature selection) and RL (which is intractability in high-dimensional feature spaces), the proposed IBFA achieves non-greedy acquisitions using stochastic encodings sets and eliminates the sparse rewards and the deadly-triad problem in RL. Empirical performance of the proposed IBFA is robust across diverse datasets, and it consistently outperforms CMI-based, RL-based, and other baseline methods in predictive accuracy and feature acquisition efficiency.

**Strengths:**

+ The paper provides IBFA as a viable alternative to CMI and RL for AFA problems.

+ The idea of applying IB to regularize the latent space, so to capture only label-relevant information to reduce reliance on noisy or redundant features, is simple but solid. This regularization enables IBFA to make feature acquisitions by considering the impact of unobserved feature realizations on prediction, which improves existing AFA methods that generally ignore these dependencies.

+ The adapted synthetic experiment is useful. Ablations on the number of latent samples, variational IB regularization, and acquisition methods highlight the necessity of each model component for optimal performance.

**Weaknesses:**

- Why Theorems 1 & 2 matter? What are their findings (in an AFA context)?

-  Although IBFA offers robust empirical results, the paper provides limited information on computational requirements, especially in terms of scaling the model to larger or real-time applications, i.e., dynamic AFA.

- The interpretability of its feature acquisition process remains somewhat opaque. For example, providing more interpretative insight, such as visualizing how different features affect the latent space, could improve transparency for end-users in sensitive fields like healthcare, which appears to be the very motivating example of this paper.

- My main concern to give an even high rating is that the approach could be perceived as a reapplication of IB rather than a fundamentally new method. Its novelty primarily lies in adapting IB within an AFA context rather than a new technique in IB itself. With that being said, the paper could strengthen its contribution by exploring how IBFA might be adapted to more generalized tasks or by testing it against other latest AFA models with hybrid feature acquisition strategies.

**Questions:**

I reproduce the weaknesses as questions to be addressed:

- Why Theorems 1 & 2 matter? What are their findings (in an AFA context)?

-  Although IBFA offers robust empirical results, the paper provides limited information on computational requirements, especially in terms of scaling the model to larger or real-time applications, i.e., dynamic AFA.

- The interpretability of its feature acquisition process remains somewhat opaque. For example, providing more interpretative insight, such as visualizing how different features affect the latent space, could improve transparency for end-users in sensitive fields like healthcare, which appears to be the very motivating example of this paper.

- My main concern to give an even high rating is that the approach could be perceived as a reapplication of IB rather than a fundamentally new method. Its novelty primarily lies in adapting IB within an AFA context rather than a new technique in IB itself. With that being said, the paper could strengthen its contribution by exploring how IBFA might be adapted to more generalized tasks or by testing it against other latest AFA models with hybrid feature acquisition strategies.

**Details Of Ethics Concerns:**

N/A, it's a basic research.

---

> ### Author Response · Authors · 2024-11-22
> **Official Response to Reviewer B3o7 Part 1**
>
> We are grateful for your time reviewing and the positive feedback. Based on your review we have updated our manuscript by: adding a result on a GSMRL baseline; added a discussion on the interpretability of the latent space and updated Section 5.3 to add clarity around Theorems 1 and 2. We answer your questions below.
>
> # Theorems 1 and 2
>
> In standard Deep Information Bottleneck works, the Variational Information Bottleneck loss is used (Alemi et al. 2017). However, we do not use this loss, we instead use a custom loss and Theorems 1 and 2 provide theoretical justification for this shift. The reason being it adds a level of regularization that further improves acquisition.
>
>  As a reminder, the Variational Information Bottleneck loss is:
>
> $L\_{\text{VIB}} = \mathbb{E}\_{p\_{\theta}(\mathbf{z} | \mathbf{x}\_S)} \bigg[-\log(p\_{\phi}(y | \mathbf{z}))\bigg]+ \beta D\_{\text{KL}}(p\_{\theta}(Z|\mathbf{x}\_S) || p(Z))$
>
> Where **one sample** is taken in the expectation. We instead use the loss:
>
> $L = -\log \big( \mathbb{E}\_{p\_{\theta}(\mathbf{z} | \mathbf{x}\_S)}\big[p\_{\phi}(y | \mathbf{z})\big] \big) + \beta D\_{\text{KL}}(p\_{\theta}(Z|\mathbf{x}\_S) || p(Z))$
>
> The difference being that we have moved the expectation inside the logarithm with **more than one sample** used in the Monte-Carlo estimate.
>
> Theorem 2 tells us this adjusted loss is equivalent to:
>
> $-I\_{\phi}(Z ; Y) + \beta I\_{\theta}(Z ; X) + I\_{\theta, \phi}(Z; Y | X)$
>
> Which is the IB loss with the added term $I\_{\theta, \phi}(Z; Y | X)$. Theorem 1 is used to explain why this additional term is one that is desired and regularizes the latent space even further to be more conducive to effective acquisitions. The change in the loss does not improve predictive performance but does improve the acquisition performance. This is seen in ablations when the VIB loss is used, see the "1 Train Sample" row in Tables 2 and 4.
>
> An alternative, more intuitive reason for the change in loss functions is as follows: We desire to construct a latent space where there are separated regions in the latent space associated with each class. Such that initially we have a large latent variance, covering all of these regions. As more features are acquired the latent variance is reduced. However, if we train with only one sample of the latent space with subsampled feature subsets (VIB loss), then for the case of no features the model should predict high uncertainty for every sample (rather than the integral over samples). And so the label uncertainty is encoded into the position of the latent space which will create regions of the latent space that predict no class. The latent encoding with no features is around one region with a low latent variance where every sample predicts no class. Crucially taking samples in the latent space to make an acquisition will not explore a diverse set of possible latent realizations anymore and acquisitions suffer. Theorems 1 and 2 formalize this intuitive reasoning.
>
> We appreciate the feedback on this section, and as a result have improved the clarity and motivation for these theorems in the context of our model and its training.
>
> # Computational Complexity
>
> A table of computational complexities can be found in the Appendix, Table 5. Additionally in Appendix K (line 1676) we discuss the computational requirements for our experiments. They were run on a single Nvidia Quadro RTX 8000, and took approximately a month to complete, by modern standards these are relatively modest requirements. We have included the computational complexity table and discussion below for your convenience.
>
> There are two places to consider runtime: training and inference. The computational complexities of each method with respect to number of features $d$ are:
>
> |Model | Single Training Step | Single Acquisition Step |
> | --- | --- | --- |
> | DIME | $\mathcal{O}(d)$ | $\mathcal{O}(1)$ |
> | GDFS | $\mathcal{O}(d)$ | $\mathcal{O}(1)$ |
> | EDDI | $\mathcal{O}(1)$ | $\mathcal{O}(d)$ |
> | Opportunistic RL | $\mathcal{O}(d)$ | $\mathcal{O}(1)$ |
> | VAE | $\mathcal{O}(1)$ | $\mathcal{O}(d)$ |
> | | | |
> | IBFA | $\mathcal{O}(1)$ | $\mathcal{O}(\vert y\vert)$ |
>
> RL, DIME and GDFS train by simulating acquisition, so each step scales linearly with the number of features. Generative models (and IBFA) are constant to train since they only train to predict well. However, during inference RL, DIME and GDFS only require one forward pass of their policy network, whereas EDDI and VAE must individually score every feature. IBFA instead takes gradients with respect to the predicted class outputs, this is done with backpropagation so scales in the same way as the forward pass. Therefore, the runtime is linear in the number of classes, which is typically far fewer than the number of features.
>
> The main takeaway is that IBFA scales better than half the methods at training time, better than the other half during acquisition (assuming fewer labels than features), and never the worst.

---

> ### Author Response · Authors · 2024-11-22
> **Official Response to Reviewer B3o7 Part 2**
>
> # Interpretability
>
> Ultimately interpretability was not the main focus of our work, but we agree it is an interesting research direction. Whilst the latent space is regularized, it is still high dimensional and the predictor network is non-linear, therefore it is difficult to interpret how features affect the latent space. We would like to add that no other AFA methods provide interpretability, they either use Deep RL, deep networks to estimate MI or deep generative models.
>
> However, there is a strong case that a model like IBFA can be used as a more interpretable model than a black box. In Interpretability research, models that dynamically select features have been investigated for this property (Chattopadhyay et al. 2022, Chattopadhyay et al. 2023), since in these papers acquisitions are framed as interpretable queries. In both of these two works and our own, the deep networks used are not themselves interpretable, but the property that AFA provides is claimed to be interpretable.
>
> To explore one way of interpreting our model, we consider the log determinant of the latent covariance matrix. This tells us the latent uncertainty as we acquire features. We consider the synthetic datasets where we know the optimal behavior. We consider the case with no features, all features, feature 1, feature 5, feature 10 and feature 11 since each of those is associated with one of the logits of the synthetic datasets and feature 11 tells us which logit to use. The results are given below:
>
> | | Syn 1 | Syn 2 | Syn 3|
> | --- | --- | --- | --- |
> | $\log(\vert \Sigma \vert)$ No Features| $1.366 \pm 0.551$ | $2.557 \pm 0.314$ | $19.468 \pm 1.294$ |
> | $\log(\vert \Sigma \vert)$ All Features | $-25.002 \pm 0.306$ | $-21.004 \pm 0.398$ | $-44.015 \pm 2.032$ |
> | $\log(\vert \Sigma \vert)$ Feature 1| $-3.787 \pm 0.592$ | $-2.524 \pm 0.214$ | $19.236 \pm 1.227$ |
> | $\log(\vert \Sigma \vert)$ Feature 5 | $-1.516 \pm 0.485$ | $2.489 \pm 0.317$ | $12.424 \pm 0.854$ |
> | $\log(\vert \Sigma \vert)$ Feature 10 | $1.307 \pm 0.557$ | $0.020 \pm 0.417$ | $11.905 \pm 0.804$ |
> | $\log(\vert \Sigma \vert)$ Feature 11| $-3.144 \pm 0.626$| $-1.792 \pm 0.236$| $6.450 \pm 1.106$|
>
>
> We see that in all cases we have the most latent uncertainty when we have no features, and the least uncertainty when we have all features. If we were to acquire the uninformative feature for each dataset (10 for Syn 1, 5 for Syn 2 and 1 for Syn 3) we see that the latent uncertainty does not reduce significantly, IB has worked effectively and (mostly) disregards these features. We see in the case of Syn 3 that feature 11 reduces the uncertainty the most, showing that even though it does not reduce uncertainty in the label at first it is able to reduce the uncertainty in the latent space. It also significantly reduces the uncertainty for Syn 1 and Syn 2, although not as much as Feature 1 in those cases. An interpretive insight is that an effective acquisition reduces latent uncertainty, although this does not explain the **exact** ordering of acquisitions.
>
> We augment this table by plotting a TSNE projection of the latent space, which can be found in our updated manuscript. This has been done for Syn 3. In this case we see that Feature 11 is able to cluster the latent space more distinctly than the other features. This visualization of the latent space shows the importance of this Feature and its early selection. We also see that Feature 10 is able to cluster more distinctly than Feature 1, showing that it is also more important for prediction as we know from the design of the dataset.

---

> ### Author Response · Authors · 2024-11-22
> **Official Response to Reviewer B3o7 Part 3**
>
> # Novelty
>
> We agree that our paper’s novelty primarily lies in the application of an IB based model rather than improving the IB technique itself. However, there is a lot more to our work than just IB. The main contribution and novelty of the work is not using IB in isolation, but the overall approach to AFA: using distributions of unobserved features in a latent space and a scoring weighted by predicted probabilities to overcome the failure cases of CMI. Information Bottleneck is one piece of this framework that regularizes the stochastic encodings.
>
> It is also important to note that an IB based model cannot directly solve AFA on its own. It required designing an acquisition criterion, structuring the encoders, adapting the loss and deeper understanding of the failure cases of CMI to construct a solution, all of which are novel elements of our work. Conversely, IB is a useful regularization technique that allows our method to work, we see empirically in our ablations the method would not work as well without IB. Furthermore it is not just standard IB that we use, we have adapted the loss to include further regularization within the IB framework (see the discussion on Theorems 1 and 2), which our ablations demonstrate is necessary for performance.
>
> We agree a method that applies to a larger range of problems would be a stronger contribution, but this could be said of many works. Ultimately our paper introduces a novel overarching framework to conduct AFA specifically - an important real-world problem. The IBFA model has moved away from standard approaches of CMI Maximization and RL, laying a foundation for further research into this approach for AFA, which we argue is a significant contribution.
>
> We appreciate the feedback about comparing to a more recent hybrid AFA baseline. Whilst we already compare to state-of-the-art AFA methods (DIME from ICLR 2024 and GDFS from ICML 2023), we have subsequently tested GSMRL (Li et al. 2021). GSMRL is a hybrid AFA model that uses generative models to improve the RL agent, by providing intermediate reward about the information gain and providing additional information to the RL agent. We have tested GSMRL on the Cube dataset using the public implementation of GSMRL (https://github.com/lupalab/GSMRL). We have used the exact hyperparameters suggested in the public repository for the Cube task. We provide the average accuracies during acquisition below, we also include the same numbers for IBFA and Opportunistic RL (from Table 3) for easy comparison:
>
> | Model | Average Acquisition Accuracy |
> | --- | --- |
> | IBFA (ours) | $0.904 \pm 0.001$ |
> | Opportunistic RL | $0.901 \pm 0.000$ |
> | GSMRL | $0.823 \pm 0.002$ |
>
> IBFA and Opportunistic RL outperform GSMRL. We have also added this Table and the acquisition curves to the updated manuscript to show this result visually.
>
>
>
>
> # References
>
> 1. Alemi, A.A., Fischer, I., Dillon, J.V. and Murphy, K., 2016. Deep variational information bottleneck.
>
> 2. Chattopadhyay, A., Slocum, S., Haeffele, B.D., Vidal, R. and Geman, D., 2022. Interpretable by design: Learning predictors by composing interpretable queries.
>
> 3. Chattopadhyay, A., Chan, K.H.R., Haeffele, B.D., Geman, D. and Vidal, R., 2023. Variational information pursuit for interpretable predictions.
>
> 4. Li, Y. and Oliva, J., 2021, July. Active feature acquisition with generative surrogate models. In International conference on machine learning (pp. 6450-6459). PMLR.

---

> > ### Author Response · Authors · 2024-11-29
> > **Is there anything outstanding to discuss?**
> >
> > Dear Reviewer B3o7,
> >
> > We are just checking in to see if we have addressed all your questions, or if there is anything outstanding we can address in the remaining discussion period. Many thanks again for the time and effort writing the review and providing feedback.

---

### Official Review · Reviewer_BL3b · 2024-11-03

**Soundness:** 3
**Presentation:** 4
**Contribution:** 3
**Rating:** 8
**Confidence:** 4

**Summary:**

The paper proposes a novel approach called IBFA to address the problem of AFA. AFA is the task of dynamically selecting which features to measure during test time to improve the prediction performance, while minimizing the number of acquired features. The key contributions are:

a) Identifying limitations of existing approaches based on RL and CMI maximization, such as training difficulties in RL and myopic feature selection in CMI.

b) Proposing a new method, IBFA, that uses an encoder-predictor architecture with an IB regularized latent space for feature acquisition. The acquisition objective considers the effect of unobserved features and focuses on distinguishing between the most likely classes.

c) Extensive evaluation on synthetic and real-world datasets, including cancer classification tasks, demonstrating that IBFA consistently outperforms various baselines.

**Strengths:**

- The paper proposes a novel approach to AFA by leveraging the Information Bottleneck principle and stochastic encodings. The idea of using a regularized latent space for feature acquisition and considering the effect of unobserved features is unique and well-motivated.
- The authors provide a thorough theoretical analysis of the limitations of existing CMI-based methods and the motivations behind their proposed approach. The experiments are extensive, covering synthetic and real-world datasets, including challenging medical tasks. The results and ablation studies are well-presented and support the claims made.
- The paper is well-written, and the methods are explained clearly. The use of block diagrams and examples aids in understanding the proposed approach.
- AFA is an important problem in many real-world scenarios, such as medical diagnosis, where features are acquired sequentially based on evolving observations. The proposed IBFA method addresses key limitations of existing approaches and demonstrates promising results, potentially leading to more effective and efficient feature acquisition in practical applications.

**Weaknesses:**

- Limited theoretical analysis of the IB regularization: While the authors provide a thorough analysis of the CMI objective's limitations, the theoretical justification for using the IB regularization in the proposed method could be stronger. A more in-depth discussion of how IB addresses the identified issues would be beneficial.
- The paper does not discuss the computational complexity of the proposed method, particularly for the gradient estimation step. An analysis of the time and memory requirements, especially for large-scale datasets, would be valuable for assessing the practical applicability of IBFA.

**Questions:**

- The proposed method relies on stochastic encodings and sampling from the latent space during acquisition. How sensitive is the performance to the number of samples used, and how can this be optimized for different datasets or computational constraints? Also, how sensitive i the method to the number of latent factors per feature?
- The paper mentions that the proposed method is currently limited to classification tasks. Could you discuss potential extensions or modifications to handle regression tasks or other problem settings where the notion of class separation is not well-defined?
- The authors mention that the factorized encoding of features to latent components introduces a modeling restriction. Could you elaborate on the implications of this restriction and discuss potential alternative architectures or formulations that could address this limitation while maintaining the benefits of the proposed approach?

---

> ### Author Response · Authors · 2024-11-22
> **Official Response to Reviewer BL3b Part 1**
>
> We’re very grateful for the highly positive feedback on our work and the time spent reviewing. Following your review we have updated our manuscript to include memory requirements of IBFA in the discussion in our conclusion and to improve the structure of Section 5.3 where we provide theoretical justification for our loss. Please find our answers to your questions below.
>
> # Theoretical Analysis
>
> Despite not primarily focusing on theory, we agree that the theoretical aspect of the paper is an important one. Apart from the theoretical exploration of CMI drawbacks, we have also provided theoretical reasoning for adapting our loss **away** from the standard Variational Information Bottleneck Loss (Alemi et al. 2017). These are given in Theorems 1 and 2 and demonstrate that our adapted loss adds another level of regularization that helps to shape the latent space to be more conducive to feature acquisition.
>
> Specifically regarding the use of IB as a regularizer, a key point is that it is not IB in isolation that addresses the drawbacks of CMI, it is our method and acquisition objective as a whole. The two main ways we solve the issues associated with CMI are:
>
> 1. Taking samples from the latent space ($p\_{\theta}(\mathbf{z} | \mathbf{x}\_{O})$) to consider effects of unobserved features.
> 2. Weighting the scores by current predictions ($p\_{\theta, \phi}(y | \mathbf{x}\_{O})$) to distinguish the most likely classes.
>
> However, these novelties alone are not enough; various decisions (such as factorizing the latent space) must be made to make IBFA work effectively. One of these decisions was to apply regularization specifically to the latent space (as opposed to regularization on the whole network such as dropout). Since we are using stochastic encoders, and we want the latent space to contain as much information about the label as possible whilst removing unwanted information about the features, Information Bottleneck is the natural choice of regularizer here; it is what IB was originally designed for (see Tishby et al. 2000). As well as this, as already discussed, by working within the Information Bottleneck framework we can add further regularization by moving away from the standard implementation of IB to our adapted loss (see Theorems 1 and 2). We appreciate the feedback and have made the reasoning behind using IB more clear in Section 5.3 as a result.
>
>
> # Computational Complexity
>
> Please find a discussion about the computational complexities below (the discussion and table can also be found in the Appendix, Table 5).
>
> There are two places to consider runtime: training and inference. The computational complexities of each method with respect to number of features $d$ are:
>
> |Model | Single Training Step | Single Acquisition Step |
> | --- | --- | --- |
> | DIME | $\mathcal{O}(d)$ | $\mathcal{O}(1)$ |
> | GDFS | $\mathcal{O}(d)$ | $\mathcal{O}(1)$ |
> | EDDI | $\mathcal{O}(1)$ | $\mathcal{O}(d)$ |
> | Opportunistic RL | $\mathcal{O}(d)$ | $\mathcal{O}(1)$ |
> | VAE | $\mathcal{O}(1)$ | $\mathcal{O}(d)$ |
> | | | |
> | IBFA | $\mathcal{O}(1)$ | $\mathcal{O}(\vert y\vert)$ |
>
> RL, DIME and GDFS train by simulating acquisition, so each step scales linearly with the number of features. Generative models (also IBFA) are constant to train since they only train to predict well. However, during inference RL, DIME and GDFS only require one forward pass of their policy network, whereas EDDI and VAE must individually score every feature. IBFA instead takes gradients with respect to the predicted class outputs, this is done with backpropagation so scales in the same way as the forward pass. Therefore, the runtime is linear in the number of classes, which is typically far fewer than the number of features.
>
> Regarding memory, during training RL methods need to store past experiences, which is linear in the buffer size. IBFA uses multiple samples to train and carry out inference, so is linear in the number of samples in each case (we discuss how many samples are needed next). EDDI and VAE have a similar memory requirement at inference but are constant during training (since they only use one sample during training).
>
> The main takeaway is that IBFA scales better than half the methods at training time, better than the other half during acquisition (assuming fewer labels than features), and never the worst. IBFA uses more memory during inference than RL, DIME and GDFS depending on how many latent samples are used. We have added the memory requirement as a limitation in our conclusion.

---

> ### Author Response · Authors · 2024-11-22
> **Official Response to Reviewer BL3b Part 2**
>
> # Sensitivity to Number of Samples and Latent Components
>
> Full sensitivity analyses on the number of samples for the synthetic tasks are given in the Appendix, Figures 11 and 12. The summarized results are also in the main text in Table 2. The results match the expectation that 1) more training samples improves acquisition at inference time by regularizing the latent space to be more conducive to acquisition (Theorems 1 and 2), and 2) that more acquisition samples are needed to sample the full diversity of the latent space at inference time. The performance plateaus at around 100 samples, therefore we used the optimal numbers that achieve good performance whilst remaining computationally efficient (100 train and 200 acquisition samples), we found that this produced strong results across all datasets. The recommendation would be to use as many samples as the computational resources allow, but as we see performance plateaus after 100 samples.
>
> Regarding the number of latent factors per feature, this was one of the hyperparameters we tuned, testing three values 4, 6 and 8. To give an idea of the sensitivity we have provided the validation score (the area under the acquisition curve) for each hyperparameter configuration for a variety of the real world datasets below. Note that it is not only the number of latent components that changes between hyperparameter configurations.
>
> | No. Latent Components | Bank Marketing | California Housing | MiniBooNE | TCGA |
> | --- | --- | --- | --- | --- |
> | 4 (Config 1) | $0.904 \pm 0.001$ | $0.652 \pm 0.003$ | $0.946 \pm 0.000$ | $0.834 \pm 0.004$ |
> | 4 (Config 2) | $0.904 \pm 0.000$ | $0.654 \pm 0.004$ | $0.946 \pm 0.000$ | $0.840 \pm 0.003$ |
> | 4 (Config 3) | $0.902 \pm 0.002$ | $0.651 \pm 0.004$ | $0.946 \pm 0.000$ | $0.834 \pm 0.005$ |
> | 4 (Config 5) | $0.897 \pm 0.000$ | $0.637 \pm 0.002$ | $0.942 \pm 0.000$ | $0.829 \pm 0.002$ |
> | 4 (Config 7) | $0.902 \pm 0.002$ | $0.659 \pm 0.003$ | $0.947 \pm 0.000$ | $0.837 \pm 0.001$ |
> | | | | |
> | 6 (Config 4) | $0.907 \pm 0.001$ | $0.648 \pm 0.002$ | $0.947 \pm 0.000$ | $0.847 \pm 0.000$ |
> | 6 (Config 8) | $0.907 \pm 0.001$ | $0.657 \pm 0.001$ | $0.947 \pm 0.000$ | $0.841 \pm 0.002$ |
> | | | | |
> | 8 (Config 6) | $0.897 \pm 0.004$ | $0.640 \pm 0.002$ | $0.946 \pm 0.001$ | $0.843 \pm 0.003$ |
> | 8 (Config 9) | $0.895 \pm 0.001$ | $0.634 \pm 0.000$ | $0.942 \pm 0.000$ | $0.828 \pm 0.003$ |
>
> The validation performance is not particularly sensitive to the number of latent components (for the values tested). In Table 13 we provide the selected hyperparameter configuration for each dataset, and we see there is no significant preference for the number of latent components. As with most hyperparameters, the best solution is to tune it with a suitable validation set and validation metric, we chose the area under the acquisition curve.
>
>
>
>
> # Extension to Regression
>
> We view this as an interesting avenue for future research. The methods we are exploring are:
>
> 1. Setting up our predictor network to have two heads. One that predicts the regression target directly, which is used to make predictions. For the second prediction head we also discretize the label into bins with approximately equal frequency, so that this head is solving a classification problem. Therefore we can use the regression head to predict the regression label, and the classification head to carry out the feature acquisitions. Reframing regression as classification is not a new idea, it is often used to help stabilize training (see Stewart et al 2023. for example).
>
> 2. Using the same objective and extending the expectation over $y$ to continuous values. The acquisition objective in Equation 1 is given by
>
> $R(\mathbf{x}\_O, i) = \sum\_{c \in [C]}p\_{\theta, \phi}(Y=c | \mathbf{x}\_{O})\int p\_{\theta}(\mathbf{z} | \mathbf{x}\_{O})r(c, \mathbf{z}, i) d\mathbf{z}$
>
> We can change the sum over discrete labels to an integral over possible regression targets:
>
> $R(\mathbf{x}\_O, i) = \int p\_{\theta, \phi}(y | \mathbf{x}\_{O}) p\_{\theta}(\mathbf{z} | \mathbf{x}\_{O})r(y, \mathbf{z}, i) d\mathbf{z}dy$
>
> This integral can be carried out using Monte-Carlo estimation. The first method would be faster by not having to take large Monte-Carlo estimates, but approximation errors appear when discretizing the label. On the other hand, there are approximation errors that arise from Monte-Carlo estimates of this integral, so it is an interesting, non-trivial direction for further research.

---

> ### Author Response · Authors · 2024-11-22
> **Official Response to Reviewer BL3b Part 3**
>
> # Factorized Latent Space
>
> The factorized latent space means that each latent component is only encoded by one feature. This allows us (by choice) to design an acquisition objective so that we can score latent components, and then this allows us to trivially score features.
>
> The implication of this modelling choice is that measuring one feature has no effect on the latent encodings of any other features. When in reality if there are correlated features they should affect the latent encoding of the other. For example, if features 1 and 2 are highly correlated and feature 1 is measured, the model’s uncertainty of feature 2 is unchanged (even though a-priori this should be reduced). Note: the acquisition objective still accounts for the observed features, since our objective uses the whole latent vector in $r(c, \mathbf{z}, i)$ so IBFA still makes acquisitions based on current observations.
>
> Ultimately, we had to make a tradeoff between the trivial linking of latent components to features and using a more general encoding architecture. Empirically IBFA performed better than the baselines, justifying this decision.
>
> Regarding alternatives, it is non-trivial to design a method that relaxes this architectural choice and maintains the benefits of IBFA. For instance, a fully connected encoder accounts for correlations between features but it is not possible to link scores in the latent space to single features. For this reason we view this as interesting future research. One **practical approach** is to use a feature selection method before applying IBFA (since we do not make claims that one model has to constitute the entire AFA pipeline), this will filter out correlated features and has the added benefit of reducing the number of features in the AFA problem.
>
>
>
> # References
>
>
> 1. Alexander A. Alemi, Ian Fischer, Joshua V. Dillon, and Kevin Murphy. Deep Variational Information Bottleneck. In International Conference on Learning Representations, 2017.
>
> 2. Tishby, N., Pereira, F.C. and Bialek, W., 2000. The information bottleneck method. arXiv preprint physics/0004057.
>
> 3. Stewart, L., Bach, F., Berthet, Q. and Vert, J.P., 2023, April. Regression as classification: Influence of task formulation on neural network features. In International Conference on Artificial Intelligence and Statistics (pp. 11563-11582). PMLR.

---

> > ### Author Response · Authors · 2024-11-29
> > **Is there anything outstanding to discuss?**
> >
> > Dear Reviewer BL3b,
> >
> > We are just checking in to see if we have addressed all your questions, or if there is anything outstanding we can address in the remaining discussion period. Many thanks again for the time and effort writing the review and providing feedback.

---

### Author Response · Authors · 2024-11-22
**Updated Manuscript**

To all reviewers, area chairs, senior area chairs and program chairs,

We're very grateful for the time spent reviewing our work and the valuable feedback. Following the feedback we have updated our manuscript. The changes are:

- Adding memory requirements to the limitations discussion in the conclusion (Reviewer BL3b)
- Added parameter subscripts where necessary to more clearly distinguish between true probability densities and densities approximated using our model (Reviewer PiTL)
- Revised and restructured Section 5.3 on our loss function, so that it is easier to follow and provides clearer theoretical justification for our loss (Reviewers BL3b, PiTL, B3o7)
- Added GSMRL as an additional baseline to be tested in Appendix M (Reviewers B3o7, FV6m)
- Added a discussion about interpreting the latent space in Appendix L (Reviewer B3o7)

---

### Meta-Review · Area_Chair_Nypu · 2024-12-21

**Metareview:**

This paper proposes a novel active feature acquisition (AFA) framework that deviates from standard CMI and RL-based methods. The approach leverages a latent variable model with a regularization term inspired by the Information Bottleneck (IB). The authors claim improvements in acquisition efficiency and provide theoretical analysis alongside empirical results showing strong performance on benchmarks.

**(b) Strengths:**
1. **Novelty:** Introduces a unique AFA framework distinct from traditional approaches.
2. **Empirical Results:** Demonstrates superior performance over baselines across multiple benchmarks.
3. **Relevance:** Tackles a practical and impactful problem in feature acquisition.

**(c) Weaknesses:**
1. **Theoretical Concerns:** While the authors addressed earlier flaws in their proofs, the revised Theorem 1 adds limited insight, and key assumptions about the latent variable $Z$ and conditional independence could be better justified.
2. **Focus on IB:** The role of the IB framework is relatively minor and primarily serves as a regularizer, which could be clarified further.
3. **Limited Scope:** The absence of stopping criteria and feature cost considerations limits the method’s applicability in broader contexts.

The paper demonstrates strong empirical performance and introduces a novel AFA framework with practical relevance. However, concerns regarding the theoretical contributions, including limited insight from revised theorems and some unclear assumptions, raise questions about the robustness of the analysis. Additionally, the focus on IB could be better contextualized, and the clarity of presentation improved. While promising, the paper would benefit from substantial revisions to address these points.

**Additional Comments On Reviewer Discussion:**

During the reviewer discussion, significant concerns persisted regarding the paper's theoretical contributions. While the authors addressed some earlier issues in their proofs, Reviewer PiTL noted that the revised Theorem 1 provided limited insight, essentially reducing to a restatement of the CMI definition, and emphasized that key assumptions about the latent variable 𝑍 Z and conditional independence remained insufficiently justified.

Additionally, the Information Bottleneck (IB) component was viewed primarily as a regularizer rather than a central aspect of the method. The authors are encouraged to better highlight the novelty of sampling in the latent space and provide a more robust justification for this approach.

---

### Decision · Program_Chairs · 2025-01-22

Reject